# SinoLC-1: the first 1-meter resolution national-scale land-cover map of China created with the deep learning framework and open-access data

Zhuohong Li[1], Wei He[1], Mofan Cheng[2], Jingxin Hu[2], Guangyi Yang[2], and Hongyan Zhang[3]

[1]State Key Laboratory of Information Engineering in Surveying, Mapping and Remote Sensing, Wuhan University, Wuhan, 430079, PR China
[2]School of Electronic Information, Wuhan University, Wuhan 430079, PR China
[3]School of Computer Science, China University of Geosciences, Wuhan, 430074, PR China

*Correspondence to*: Hongyan Zhang (zhanghongyan@cug.edu.cn)

**Abstract.** In China, the demand for a more precise perception of the national land surface has become most urgent given the pace of development and urbanization. Constructing a very-high-resolution (VHR) land-cover dataset for China with national coverage, however, is a non-trivial task and thus, an active area of research impeded by the challenges of image acquisition, manual annotation, and computational complexity. To fill this gap, the first 1-meter resolution national-scale land-cover map of China, SinoLC-1, was established using a deep learning-based framework and open-access data including global land-cover (GLC) products, open street map (OSM), and Google Earth imagery. Reliable training labels were generated by combining three 10-meter GLC products and OSM data. These training labels and 1-meter resolution images derived from Google Earth were used to train the proposed framework. This framework resolved the label noise stemming from a resolution mismatch between images and labels by combining a resolution-preserving backbone, a weakly supervised module, and a self-supervised loss function, to refine the VHR land-cover results automatically without any manual annotation requirement. Based on large storage and computing servers, processing the 73.25 TB dataset to obtain the SinoLC-1 covering entire China, ~9,600,000 km$^2$, took about 10 months. The SinoLC-1 product was validated using a visually interpreted validation set including over 100,000 random samples and a statistical validation set collected from the official land survey report provided by the Chinese government. The validation results showed SinoLC-1 achieved an overall accuracy of 73.61% and a kappa coefficient of 0.6595. Validations for every provincial region further indicated the accuracy of this dataset across whole China. Furthermore, the statistical validation results indicated the SinoLC-1 conformed to the official survey reports with an overall misestimation rate of 6.4%. In addition, SinoLC-1 was compared with five other widely used GLC products. These results indicated SinoLC-1 had the highest spatial resolution and the finest landscape details. In conclusion, as the first 1-meter resolution national-scale land-cover map of China, SinoLC-1 delivered accuracy and provided primal support for related research and applications throughout China. The SinoLC-1 land-cover product is freely accessible at https://doi.org/10.5281/zenodo.7707461 (Li et al., 2023).

# 1 Introduction

As a basic earth observation application, land-cover mapping enables investigating human and nonhuman activities that shape the national landscape (Lin & Ho, 2003). Researchers and decision-makers use the insights from the land-cover maps to assist communities and governments achieve Sustainable Development Goals (Wang et al., 2022). The past few decades have witnessed tremendous advancements in the spatial resolution of land-cover mapping products because remote-sensing images with finer spatial resolution can be acquired more easily (Roy et al., 2021). Very-high-resolution (VHR) imagery in particular, typically finer than 3 m/pixel, reveals land-cover objects at an ever finer granularity providing a clearer, more detailed picture of the situation on the ground (Feng & Li, 2020). The VHR land-cover datasets are becoming increasingly ubiquitous in numerous large-scale research and application domains, such as agriculture (Griffiths et al., 2019), urbanization (Luo & Ji, 2022), and ecology (Y. Yang et al., 2020). As the largest agricultural country and the second-largest economy in the world, China experienced rapid development and urbanization in the past decades (Chang & Brada, 2006; Guan et al., 2018), and much land-cover research about China has been conducted. However, the VHR land-cover map with national coverage is still unavailable in China, hindering effective policy formulation and efficient resource allocation. In this context, the investigation into the fine-grained national-scale land-cover map for China is a necessary guiding principle for comprehensively understanding the environment, development, and future trend of the country.

Over the past 40 years, numerous satellite missions have been launched to improve the knowledge of Earth's resources and monitor natural phenomena. With the continuous updating of airborne and space-borne platforms, the spatial resolution of the available remote-sensing images has undergone rapid increments of change (Tong et al., 2020; Li et al., 2022). Moreover, the studies for the land-cover mapping methods have achieved great progress. Based on the context, the spatial resolutions of the published land-cover products have been through the trends of coarse to fine (Cao & Huang, 2022). Nevertheless, due to the low orbit of the VHR image-captured platforms, the corresponding VHR land-cover products generally have a smaller coverage that is insufficient to cover entire China (Wang et al., 2021). Furthermore, even if the national-scale VHR imagery can be obtained by combining different image sources, the immense data volumes, laborious annotations, and computational costs are still the main obstacles for national-scale VHR land-cover mapping. Thus, currently, available land-cover datasets for China lack either a fine spatial resolution or nationwide coverage. In terms of coverage scale and spatial resolution, the relational existing land-cover datasets can be grouped into four general types: global-scale low-resolution (LR), global-scale moderate-/high-resolution (MR/HR), national-scale MR/HR, and region-scale VHR land-cover products.

(1) Global-scale LR land-cover products:

From the 1980s to the 2010s, global remote-sensing imagery with LR (finer than 1000 m/px) can be captured by satellites including Satellite pour l'Observation de la Terre 4 (SPOT 4), Advanced Very High Resolution Radiometer (AVHRR), Moderate Resolution Imaging Spectroradiometer (MODIS), and Environmental Satellite. Subsequently, many representative LR global products have emerged, for example, the European Commission's Joint Research Centre (JRC) published a 1-kilometer-resolution global land-cover (GLC) product in 2007, which was classified based on the imagery

from SPOT 4 (Bartholomé & Belward, 2007). The JRC and the United States Geological Survey (USGS) produced a 1-kilometer-resolution GLC product based on the monthly AVHRR normalized difference vegetation index composites (Loveland et al., 2010). Moreover, the USGS and the National Aeronautics and Space Administration produced a 500-meter-resolution GLC product in 2009, called MOD12Q1, which was based on MODIS imagery and classified through the decision tree algorithm (Friedl et al., 2010).

(2) Global-scale MR/HR land-cover products:

From the 2010s to the 2020s, owing to the open-access imagery of Landsat and Sentinel missions with moderate (~30 m) and high (~10 m) resolution, the research of the global-scale MR/HR land-cover mapping has blossomed. For the MR land-cover products, Gong et al. (2013) proposed the first 30-meter GLC product based on Landsat data, called FROM_GLC, with an overall accuracy of 65%. Soon afterward, based on the Landsat data and the imagery of the Huanjing-1 satellite, Chen et al. (2015) produced a 30-meter GLC product, called GlobeLand30, with an accuracy of 80%. Lately, based on Landsat time series imagery, Zhang et al. (2021) proposed GLC_FCS30, which is a 30-meter GLC product with an accuracy of 83%. Numerous GLC products with high resolution were also published recently. Based on Sentinel-2A imagery, Gong et al. (2019) produced the first 10-meter GLC map with an accuracy of 73%. Based on Sentinel-1 and 2 data, ESA provided an annually updated 10-meter GLC map since 2020, with a reported accuracy of 74% (Van De Kerchove et al., 2021). Similarly, based on Sentinel-2 imagery, Environmental Systems Research Institute (ESRI), Inc. and Impact Observatory, Inc. proposed a 10-meter GLC product in 2021, which reported an accuracy of 85% (Karra et al., 2021).

(3) National-scale MR/HR land-cover products:

Similarly, based on the open-access MR/HR imagery, numerous national-scale land-cover products are continuously produced. For example, with the Landsat imagery, the USGS cyclically updates the 30-m National Land Cover Database (NLCD) covering the United States (Wickham et al., 2021). With the Sentinel imagery, the United Kingdom Centre for Ecology & Hydrology (UKCEH) periodically publishes the national-scale 10-m land-cover map of the United Kingdom (Morton et al., 2021). For China, researchers adopted diverse methods to produce high-quality national-scale land-cover maps. By manually interpreting the Landsat images, Liu et al. (2014) produced a national-scale 30-m resolution land-cover product covering entire China, which revealed the land-cover patterns of China from the 1980s to 2015 at an interval of 5 years. Furthermore, based on the more frequent Landsat images and Google Earth Engine, Yang & Huang (2021) produced the first 30-m annual land-cover dataset in China and analyzes the national-scale long-term land-cover change from 1990 to 2019, which provided important support for multi-temporal land-cover research in China. Recently, Liu et al. (2023) took the training pairs with the mismatched resolution, which includes the 30-m GLC product (noisy labels) and the 10-m Sentinel images at the year of 2020, to train a deep learning-based method and produced a national-scale 10-m land-cover map of China.

(4) Region-scale VHR land-cover products:

In the 2020s, with the easily available VHR imagery, establishing VHR land-cover datasets for fine object interpretation and deep learning-based research became a research hotspot (Xia et al., 2023). The current VHR land-cover datasets are generally regional scale (typically covering a few cities/provinces and smaller than a national scale) because of the limitation of the coverage and temporal resolutions of VHR imagery. For example, Wang et al. (2021) utilized imagery from airborne cameras and Google Earth to create a 0.3-meter-resolution regional-scale dataset, covering 536.15 km$^2$ areas (including Nanjing, Changzhou, and Wuhan in China). Huang et al. (2020) proposed a 2.1-meter-resolution regional-scale land-cover dataset, called Hi-ULCM, covering 42 major cities in China. Hi-ULCM was produced based on Ziyuan-3 (ZY-3) satellite imagery and reported an overall accuracy of 86%. Moreover, Du et al. (2020) produced a 2.4-meter-resolution land-cover product, called PKU-USED, covering 81 China major cities. PKU-USED was based on the VHR imagery of ZY-3, Gaofen-6 (GF-6), and Google Earth.

Different production schemes are used for these four types of land-cover products. For the LR, MR, and HR land-cover products, the image sources (i.e., MODIS, Landsat, and Sentinel) are commonly free access and contain massive spectral information but relatively low spatial context than VHR imagery. Therefore, pixel-based machine learning algorithms, for example, support vector machine, decision tree, and random forest (RF), are usually adopted to produce acceptable results (Defourny et al., 2007; Friedl et al., 2010; Gong et al., 2019). Nevertheless, the production of VHR land-cover products usually faces two main problems. First, VHR imagery is commonly captured from commercial and military satellites with high acquisition costs (Coltri et al., 2013; Pengra et al., 2015). Second, VHR imagery commonly contains a few bands, for example, the spaceborne 2.1-meter ZY-3 and 2-meter GF-6 imagery only contain four bands of red, green, blue, and near-infrared. With limited spectral information and massive spatial details, pixel-based methods generally report low accuracy in the VHR land-cover mapping task (Ce Zhang et al., 2018). Based on the second problem, the Object-Based Image Analysis (OBIA) technique is widely taken to produce VHR land-cover products. The OBIA-based methods depend on handcraft features to classify land objects and improve product accuracy (Jalan, 2012; Du et al., 2020). However, the feature selection of OBIA-based methods requires human intervention, which inevitably limits their application in large-scale product productions (Pilant et al., 2020; Huang et al., 2020).

Recently, with the blossoming of deep learning techniques, many studies have conducted deep learning-based models for VHR land-cover mapping. For example, the 1-meter National Agriculture Imagery Program imagery was taken to train a deep learning framework and produced the 15-class land-cover map for the entire state of Maryland, United States (Li et al., 2022). Moreover, by using limited spectral information from optical imagery, numerous studies have shown that deep learning methods are suitable and capable of obtaining satisfactory results in a variety of regional-scale VHR applications such as land-use mapping (Srivastava et al., 2019), construction site mapping (Cao & Huang, 2022), greenhouse mapping (Ma et al., 2021), and change detection (Zhang et al., 2020; Li et al., 2021). However, existing deep learning methods rely on well-labeled data, which are time-consuming and laborious to annotate. This limitation has created a large obstacle preventing the production of a national-scale VHR land-cover map (Cao & Huang, 2022; Li et al., 2022).

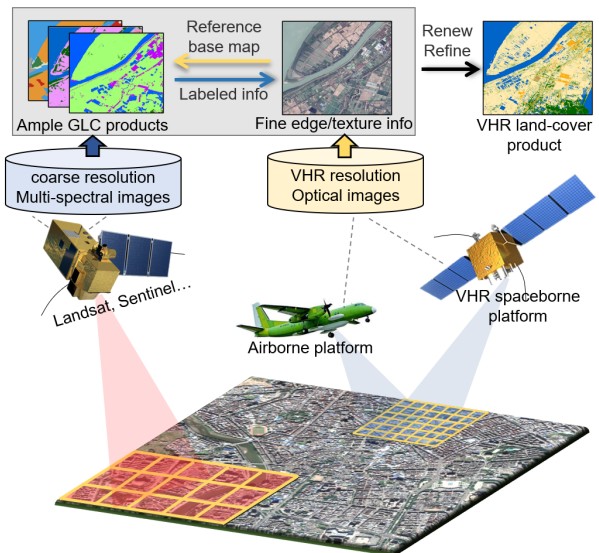

**Figure 1. Demonstration of using the fine edge and texture information from VHR images to renew and refine the current ample coarse-resolution GLC products. The VHR remote sensing images in the figure are from © Google Earth 2021.**

To overcome these limitations, in this paper, a deep learning-based framework is presented to create the first 1-meter land-cover map for entire China, called SinoLC-1, by using freely available 1-meter Google Earth imagery, open-access 10-meter GLC products, and Open Street Map (OSM) as input data. Figure 1 shows by combining the amply available GLC products containing adequate land-cover information and the VHR images containing fine edge and texture information, the VHR land-cover map can be automatically refined through the proposed framework. In detail, the multisource 10-meter land-

cover products and the OSM are first integrated to generate coarse training labels. About 30% of the land surface in China is selected to generate training pairs containing aligned VHR images and coarse labels. Training pairs are used to train the proposed low-to-high network (L2HNet), which is a large-scale VHR land-cover mapping network proposed in our previous work (Li et al., 2022). Considering the label noise caused by the mismatched resolution between the VHR images and the coarse labels, the L2HNet integrates a resolution-preserving backbone, a weakly supervised module, and a self-supervised loss

function to excavate the texture information from the VHR images and utilize the supervision information from the coarse labels. In practice, three large computing servers are used to conduct the training and mapping process. Finally, processing the whole 73.25TB data to produce the 1-meter land-cover map covering ~9,600,000 km$^2$ area of China takes about 10 months. Moreover, SinoLC-1 is produced without using any commercial data and without any requirement for manual annotations, which means the production maintains low capital expenditure and low labor cost. To the best of our knowledge, the produced

SinoLC1- is the first 1-meter-resolution and currently the highest-resolution land-cover product that covers entire China.

The remainder of this paper is arranged as follows. The dataset used is introduced in Sect. 2. The proposed framework including the processes of training data collection, land-cover classification, and assessment is illustrated in Sect. 3. The produced land-cover product is demonstrated, the validation results are analyzed, and the product limitations are discussed in Sect. 4. Access to the data is provided in Sect. 5. Finally, conclusions are given in Sect. 6.

## 2 Datasets

### 2.1 Open-access remote-sensing images at 1-meter resolution

The VHR optical images were collected from the open-access Google Earth imagery, which has a resolution of 1.07 meters. Google Earth, a well-known tool widely used in many popular image processing and GIS software, provides freely available VHR images with large-scale coverage. By integrating the images captured from different satellites (e.g., Worldview, Quickbird, IKONOS, GeoEye1, Pleiades, SuperView-1, and Kompsat3A), Google Earth imagery enables covering a very large range including entire China (Zhao et al., 2014). We have two main reasons for adopting Google Earth as the image source of VHR national-scale land-cover mapping. First, most of the VHR imagery is commonly captured from commercial and military satellites, and purchasing the imagery covering entire China is extraordinarily expensive (Rahman et al., 2010; Coltri et al., 2013; Pengra et al., 2015). Second, Google Earth imagery generally has mature sifting and preprocessing procedures to produce cloudless, high-quality imagery (Pulighe et al., 2016). Based on this image source, the misclassification of land objects caused by the image quality, cloud, and cloud shadow can be minimized. Many researchers have also reported the feasibility and possibility of using Google Earth imagery to conduct VHR large-scale land-cover mapping (Malarvizhi et al., 2016; Guo et al., 2016; Li et al., 2020).

To construct the image database for producing SinoLC-1, the imagery of the "December 2021" version was collected according to every provincial administrative region border of China and cropped into the size of 6000 × 6000 pixels as the basic storage tile. The total storage size of imagery with the band of red, green, and blue was about 73.25 TB, covering ~9,600,000 $km^2$ land surface area of China. The use of Google Earth imagery and the country boundary are demonstrated in Figure 2 (a).

### 2.2 Global land-cover data at 10-meter resolution

Annotating the VHR labeled samples for national-scale VHR land-cover mapping is a challenging, laborious process. In general land-cover mapping studies, most of the published land-cover products were produced based on well-labeled training samples, which inevitably hinders their productivity and application coverage (Cao & Huang, 2022). In this paper, multiple open-access GLC products at 10-meter resolution were integrated to obtain reliable labeled samples, and we combined weakly and self-supervised strategies during the network training to utilize them as a reasonable supervision source.

Concretely, the land-cover labeled data were collected from three open-access 10-meter GLC products, namely, FROM_GLC10 (Gong et al., 2019), ESRI world cover (Karra et al., 2021), and ESA_WorldCover v100 (Van De Kerchove et al., 2021). FROM_GLC10 was produced by using Sentinel-2A imagery, which reported an overall accuracy of 73% on a global scale. ESRI world cover (abbreviated as ESRI_GLC10) was produced based on Sentinel-2 imagery and reported an overall accuracy of 85%. ESA_WorldCover v100 (abbreviated as ESA_GLC10) was produced by using Sentinel-1 and Sentinel-2 data and reported an overall accuracy of 74%. Table 1 shows the land-cover relationships between these products and the proposed SinoLC-1.

Table 2 shows the definition, value, and color of each land-cover type of the SinoLC-1. The SinoLC-1 contains 11 land-cover classes and includes the unique class of "Traffic route" compared with other products. Subfigure (1–3) of Figure 2 (c) shows the demonstration samples of the three 10-meter GLC products located in Wuhan City.

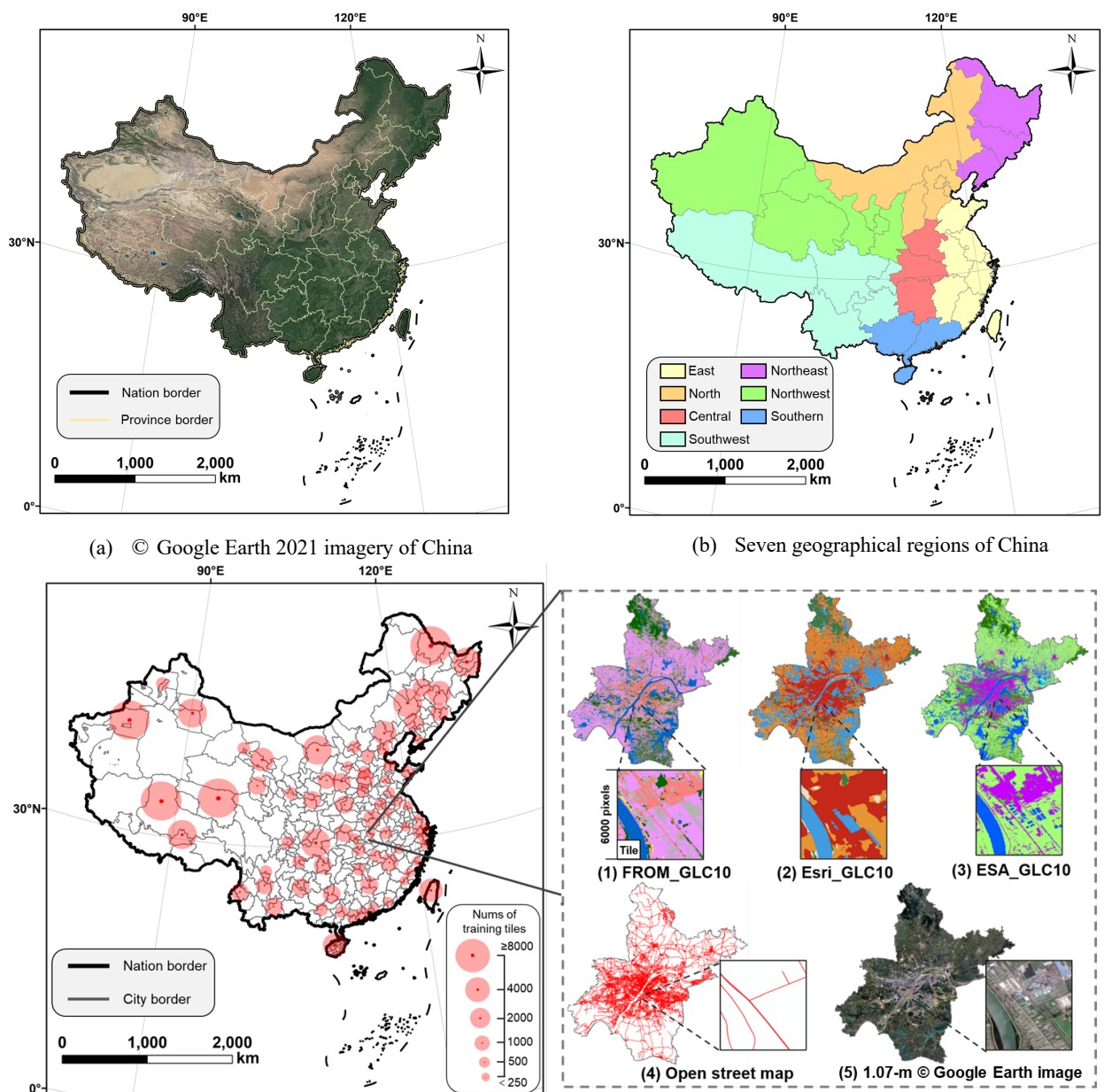

(a)  © Google Earth 2021 imagery of China

(b)  Seven geographical regions of China

(c)  Left: Distribution and volume of the training sample. Right: Demonstration of the using GLC products, OSM data, and 1.07-meter imagery from © Google Earth 2021

**Figure 2. Demonstration of the region division, training sample selection, and use of five datasets.**

**Table 1. Category relations between the FROM_GLC10, ESA_GLC10, ESRI_GLC10, and the proposed SinoLC-1.**

| | FROM_GLC10 | ESRI_GLC10 | ESA_GLC10 | SinoLC-1 |
|---|---|---|---|---|
| Affiliation | THU, China | ESRI & IO, USA | ESA, Europe | WHU, China |
| Resolution | ~10 meters | ~10 meters | ~10 meters | 1.07 meter |
| Coverage | Global | Global | Global | National (China) |
| Land-cover type & Color | Forest | Trees | Trees | Tree cover |
| | Shrubland | Scrub | Shrubland | Shrubland |
| | Grassland | Grass | Grassland | Grassland |
| | Cropland | Crops | Cropland | Cropland |
| | Impervious area | Built area | Built-up | Building |
| | | | | Traffic route |
| | Bare land | Bare | Barren/sparse veg. | Barren and sparse veg. |
| | Snow and ice | Snow and ice | Snow and ice | Snow and ice |
| | Tundra | | | |
| | Water body | Water | Open water | Water |
| | Wetland | Flooded vegetation | Herbaceous wetland | Wetland |
| | | | Mangroves | |
| | | | Moss and lichen | Moss and lichen |
| Notes: | THU=Tsinghua University; Esri=Esri, Inc.; IO=IO, Inc.; WHU=Wuhan University; | | | |

**Table 2. The definition, value, and color of each land-cover type of the SinoLC-1**

| Land-cover type | Definition | Value | Color | |
|---|---|---|---|---|
| Tree cover | Areas covered by trees generally have larger crowns and are higher than 5 meters. It can be sparse arbors or clustered forests which include evergreen forests, mixed forests, artificial forests, bamboo groves, etc. | 2 | (0, 100, 0) | |
| Shrubland | Areas covered by clusters of shrubs with a height below 5 meters. | 3 | (255, 190, 35) | |
| Grassland | Areas covered by low herbaceous plants. It generally includes natural grasslands with a fractional vegetation coverage greater than 5, rangeland with tree canopy density less than 0.3 or shrub canopy density less than 0.4, urban's vacant land dominated by grass, and other artificial grasslands. | 4 | (233, 255, 190) | |
| Cropland | The arable land and human planted crops not at tree height including upland crops (e.g., wheat, corn, potatoes, and cotton) and irrigated crops (e.g., paddy filed, lotus root, and water spinach). | 5 | (255, 235, 175) | |
| Building | Human-made structures and homogenous impervious surfaces including industrial, residential, commercial areas, and construction sites. It is generally located in urban and rural areas with high human activities. | 6 | (255, 170, 0) | |
| Traffic route | Areas constructed according to certain technical standards and equipped with necessary transportation facilities, including railways, highways, urban/rural roads, and pipelines. | 1 | (255, 0, 0) | |
| Barren and sparse vegetation | Areas covered by sparse vegetation or bare land covered by sand, gravel, or rocks, including mountains without dense vegetation and snow cover, deserts, grasslands degraded by drought, and wasteland in urban/rural areas with sparse or no vegetation. | 7 | (180, 180, 180) | |
| Snow and ice | Areas covered by large-scale permanent snow or ice, including glaciers and permanent snowpack in mountain areas or high latitudes. | 8 | (240, 240, 240) | |
| Water | Areas covered by water for a long period, including oceans, naturally formed water bodies (e.g., lakes, rivers, and runoff), and artificially formed water bodies (e.g., reservoirs, canals, water conservancy facilities, ponds, and aquaculture farms). | 9 | (0, 100, 200) | |

| | | | |
|---|---|---|---|
| Wetland | Areas with perennial or seasonal water accumulation and vegetation growth. It includes forest/shrub/grass swamps, peatlands, mudflats, mangroves, and coastal/inland tidal flats. | 10 | (0, 150, 160) |
| Moss and lichen | Surfaces or rocks attached by moss or tiny lichen plants. | 12 | (250, 230, 160) |

## 2.3 Open Street Map data

Traffic routes and transportation networks provide important information for understanding the development, urbanization, and population of a country (Osses et al., 2022). In VHR land-cover mapping research, the traffic route is a fundamental land-cover type to reveal urban patterns and reflect regional traffic (Boguszewski et al., 2020; Xia et al., 2023; Hu et al., 2023). Given that the traffic route can be clearly identified from the 1-meter resolution imagery, the land-cover type of "Traffic route" was also considered in the proposed SinoLC-1 land-cover product. To obtain reliable traffic route labeled information, the labeled data were collected from the OSM database in vector format. As one of the most popular volunteered geographic information data sources, the road pattern labeled information provided by the OSM is stable and reliable, which is often used as a supplement data in the land-cover or land-use mapping task (Zhu et al., 2022; Zhong et al., 2020; Audebert et al., 2017). To take the OSM data as a supervision source during network training, the vector OSM data were transformed into the raster format at the same resolution as the GLC products. Thus, they can be utilized as pixel-level labels to guide the training process. Subfigure (4) of Figure 2 (c) shows the samples of traffic routes obtained from the OSM located in Wuhan City, Hubei Province.

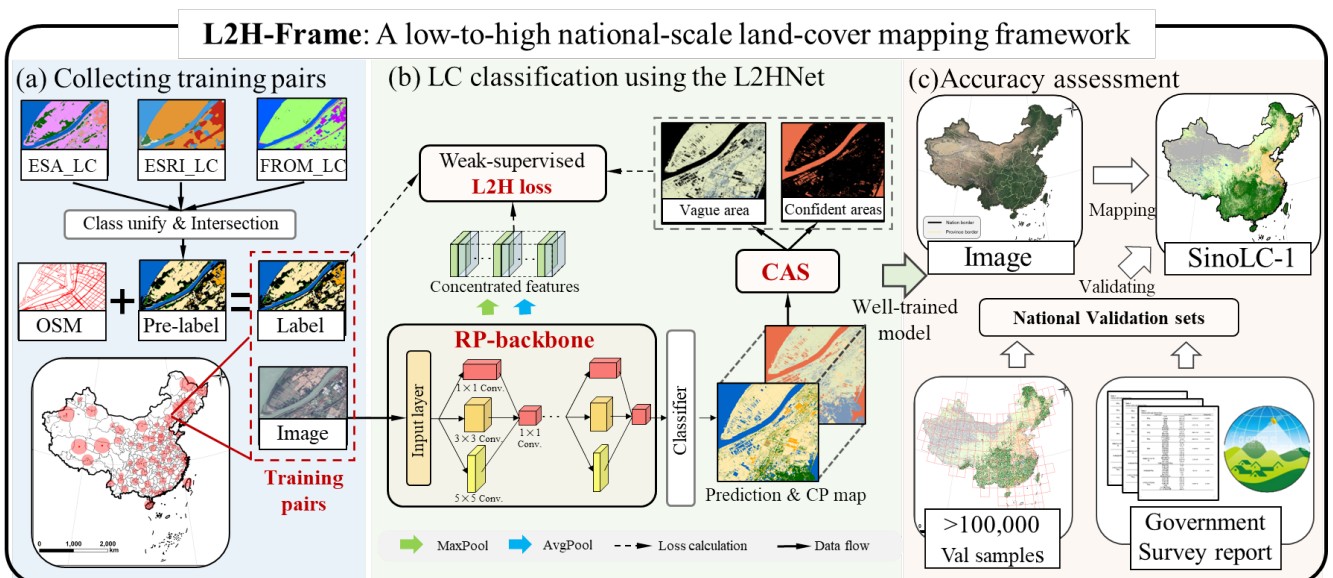

**Figure 3. The overall workflow of the L2H-Frame. The framework includes three main parts: (a) Collecting training pairs, (b) Land-cover classification using the L2HNet, and (c) Accuracy assessment. The VHR remote sensing images in the figure are from © Google Earth 2021.**

## 3 Methods

In this section, the proposed L2H-Frame, which is an efficient deep learning-based framework for national-scale VHR land-cover mapping, is introduced. Based on a series of weaky and self-supervised strategies, the L2H-Frame only takes open-access data sources as training data to produce the 1-meter resolution land-cover map of China, which allows the framework to maintain low capital expenditure cost in image acquisition and low labor cost in training label annotation. As the overall framework depicted in Figure 3, the L2H-Frame consists of three main steps: (a) Collecting nationwide training pairs, (b) Land-cover classification using the L2HNet, and (c) Accuracy assessments. In the following subsection, these main steps are introduced sequentially.

## 3.1 Collecting nationwide training pairs

To collect reliable training pairs for the national-scale VHR land-cover mapping process, 98 municipal-level areas were selected from the 34 provincial administrative regions of China. In every selected municipal-level area, the data were divided into numerous non-overlapped tiles with the size of 6000 $\times$ 6000 pixels. In each tile, the training pairs were constructed by five types of data, which included three 10-meter GLC products, the OSM data, and the 1.07-meter-resolution Google Earth images. Figure 2 (c) demonstrates the sample of the using data, location, and contained volume of tiles for all the selected training areas. Moreover, by considering the immense span of China's territory and the variable landforms, according to the geographic location, climate, economic development, and land-cover pattern (Lin, 2002; Ning et al., 2022), the land surface of China was divided into seven geographical regions for separate training. Figure 2 (b) shows the locations and borders of the seven geographical regions: east, northeast, north, northwest, central, southern, and southwest.

According to the classification system of mainstream large-scale land-cover products and the landscape style of China, the classification system of SinoLC-1 was defined as the following 11 land-cover classes: "Tree cover", "Shrubland", "Grassland", "Cropland", "Building", "Traffic route", "Barren and sparse vegetation", "Snow and ice", "Water", "Wetland", and "Moss and lichen". The detailed definitions of each type are shown in

Specifically, to obtain reliable land-cover information and generate the training labels from three GLC products, the classification systems of ESA_GLC10, ESRI_GLC10, and FROM_GLC10 were unified according to Table 1, and then the unified results were intersected to generate the pre-labels. In the pre-labels, the pixels/areas, where their land-cover types were the same in the three GLC products, would be preserved as the stable labeled areas; otherwise, the pixels/areas would be set as unlabeled type and maintained void value. In particular, because the land-cover type of "Moss and lichen" is a unique type of the ESA_GLC10 product, in the generation of pre-labels, the areas covered by the "Moss and lichen" type were directly inherited from the ESA_GLC10 product. Moreover, to generate stable labeled samples for the traffic route, the vector road pattern information collected from the OSM was transformed into raster format with the same resolution as the pre-labels, and then the transferred samples of road pattern were overlayed to the pre-labels to generate the final training labels. Figure 4 (a)

shows the proportion of the training area in each geographical region, and Figure 4 (b) shows the land-cover distribution of the training labels in each geographical region.

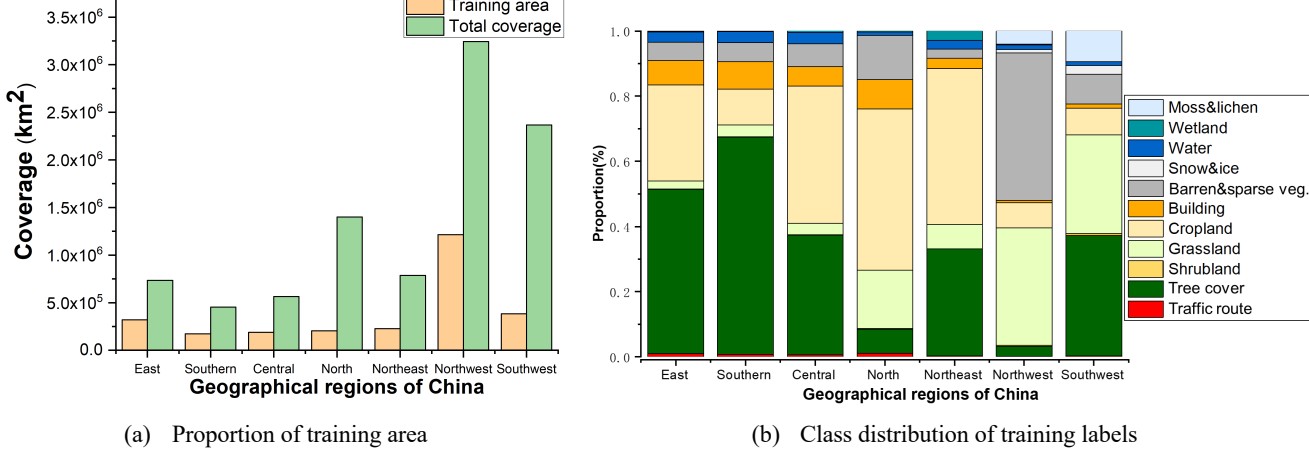

(a)  Proportion of training area                    (b)  Class distribution of training labels

**Figure 4. Statistical information of the selected training labels in seven geographical regions.**

## 3.2 Land-cover classification using the low-to-high network

### 3.2.1 Training of the low-to-high network

To process the resolution-mismatched training pairs and realize automatic national-scale VHR land-cover mapping for China, the low-to-high network (L2HNet) was applied, which has been proposed in our previous work (Li et al., 2022). Aiming at robustly extracting multiscale features and taking the coarse labels as a more reasonable supervision source during the training, as shown in Figure 3 (b), the L2HNet combined a resolution-preserving (RP) backbone, a weakly supervised-based confident area selection (CAS) model, and an unsupervised-based low-to-high (L2H) loss.

To extract features robustly from the VHR images, the images first passed through an input layer (i.e., a 64-channel 3×3 convolutional layer) to obtain dense feature maps. Then, the RP backbone consisting of five blocks, where each block contained multiscale (i.e., 1×1, 3×3, and 5×5) convolution layers with the channel setting of "64:32:16," extracted the multiscale information from the dense feature maps by highly preserving their spatial resolution. Unlike the common deep learning-based networks that deeply down-sample the features with encoder-decoder structures (e.g., UNet [Ronneberger et al., 2015] and DeepLabv3+ [Liang-Chieh et al., 2018]), in each block of the L2HNet, the channel number of different scale convolutional layers were inversely proportional to their receptive fields. Therefore, the multiscale layers can scan the feature maps with proper receptive fields to preserve the feature resolution rather than over down sampling them in case of losing feature details. Lastly, based on a classifier constructed by a SoftMax function and a 1×1 convolutional layer, the extracted features were classified into the prediction results and the corresponding confidence probability (CP) map.

To take the coarse training label as a more reasonable supervision source, the L2H loss was designed as a two-part composition with weakly and self-supervised strategies. For the first part, a weakly supervised-based CAS module was designed to select the trustworthy parts from the coarse labels and ignore the noisy samples according to the CP map of the predictions. Then, the confident area set (represented as **CA**), which had high CP in the predictions, was selected to calculate the cross entropy (CE) loss with the coarse labels, and the vague area set (represented as **VA**), which had low confidence, was ignored during the CE loss calculation. Formally, for a training patch with the size of $W \times H$, $\mathbf{Y'}$, $\mathbf{\hat{Y}}$, and $\mathbf{\hat{G}}$ represent the coarse training labels, the prediction results, and the selected mask generated by the CAS module, respectively. The modified CE loss can be written as follows:

$$\mathcal{L}_{CE}(\mathbf{Y'}, \mathbf{\hat{Y}}, \mathbf{\hat{G}}) = \frac{-\sum_{i=0}^{W}\sum_{j=0}^{H}\left[\hat{g}_{ij}\sum_{l=1}^{L} y'^{(l)}_{ij}\log\left(\hat{y}^{(l)}_{ij}\right)\right]}{\mathrm{card}\,(\mathbf{CA})}, \tag{1}$$

where $y'^{(l)}_{ij}$ and $\hat{y}^{(l)}_{ij}$ denote class $l$ of the label $\mathbf{Y'}$ and the prediction $\mathbf{Y'}$ in coordinates $(i, j)$, respectively. Element $\hat{g}_{ij}$ of the selected mask $\mathbf{\hat{G}}$ is a binary scalar to represent if the coordinate $(i, j)$ is selected into the **CA** set.

For the second part, by considering the feature similarity of the same land-cover classes, the unsupervised dynamic vague area (DVA) loss was designed to constrain the within-class variance (Otsu, 1979) dynamically between the well-predicted **CA** set and unsupervised **VA** set in the feature space. Formally, the 2-norm of the inter-area mean difference was used, represented as $\sigma_{l,b}^2$, to describe the land-cover class $l \in [1, L]$ variance in the $b \in [1, B]$ feature layer. Moreover, the DVA loss is the accumulation of $\sigma_{l,b}^2$ in every land-cover class and feature layer, whose specific form is as follows:

$$\mathcal{L}_{DVA} = \gamma \sum_{b=1}^{B}\sum_{l=1}^{L} \sigma_{l,b}^2, \tag{2}$$

where $\gamma$ is a scale factor and set as 0.05 according to our previous work (Li et al., 2022). By combining Eqs. (1) and (2), the L2H loss can be described as follows:

$$\mathcal{L}_{L2H} = \mathcal{L}_{CE}(\mathbf{Y'}, \mathbf{\hat{Y}}, \mathbf{\hat{G}}) + \mathcal{L}_{DVA}, \tag{3}$$

Furthermore, according to the location of seven geographical regions and the training sample distributions shown in Figure 2 (b) and (c), seven L2HNets were trained separately for every region to adapt the variable landforms and different land-cover patterns in the immense span of China's territory. During the training of L2HNet, each training tile (the aligned VHR image and training label with the size of 6000 × 6000 pixels) was randomly cropped into 500 patches, where each patch had a size of 256 × 256 pixels, to utilize the training data fully while ensuring training efficiency.

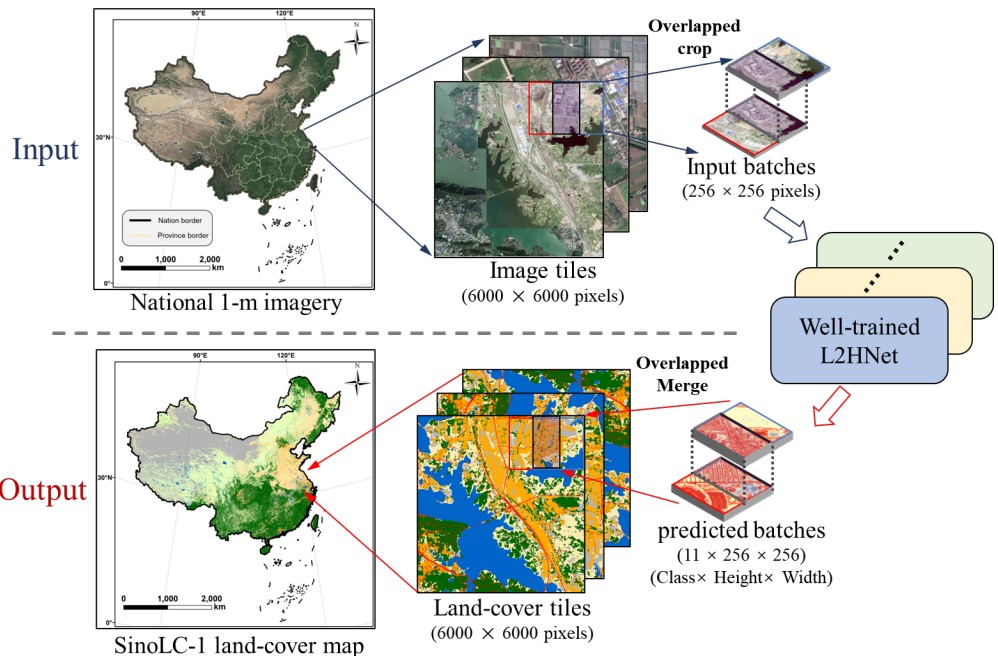

**Figure 5. Demonstration of the mapping and merging for producing SinoLC-1. The VHR remote sensing images in the figure are from © Google Earth 2021.**

## 3.2.2 Seamless mapping and merging process

To acquire the seamless national-scale land-cover map, a seamless mapping and merging strategy was employed to process the massive data covering China successively. Specifically, as shown in Figure 5, the whole process included four steps. First, the nationwide 1-meter resolution imagery was sorted out according to the borders of each provincial administrative region. In each region, the regionwide coverage image was sequentially cropped into numerous non-overlapped image tiles with each size of 6000 × 6000 pixels. Second, to obtain the image batches that can be sent to the well-trained networks, each image tile was sequentially cropped into numerous 256 × 256 patches with 128 overlapped pixels. Based on the training process introduced in Sect. 3.2.1, seven L2HNets were separately trained with the training pairs collected from seven geographical regions of China. Third, according to the geographical region of the input image source, the image batches were sent to the corresponding well-trained L2HNet, and the predicted batches of the land-cover mapping results were obtained. The input batches had 128 overlapped pixels, so the adjacent predicted batches, which represent a predicted probability matrix with the sizes of 11 × 256 × 256 (Class × Height × Width), were seamlessly merged into the land-cover tiles by calculating average probabilities of the overlapped areas and taking the arguments of the maxima (argmax) among all the classes. By conducting the seamless mapping and merging process, the influence of edge cracks between the cropped predicted batches is reduced. Finally, for each provincial administrative region, every merged land-cover tile was sequentially spliced into the intact land-cover map.

Based on the procedure, three large computing servers including 8 NVIDIA GeForce RTX 3090 GPUs and a large storage server were employed to conduct the mapping and merging of the SinoLC-1 in parallel. Processing the whole imagery with a total storage size of about 73.25 TB to obtain the SinoLC-1 land-cover product covering ~9,600,000 km$^2$ area of China took about 10 months.

## 3.3 Accuracy assessment

Assessing the accuracy of land-cover products is an essential step in describing their quality before they are used in related applications (Olofsson et al., 2013). To validate the accuracy of the proposed SinoLC-1 at pixel and statistical levels comprehensively, and to analyze the omission and commission errors in detail, a nationwide pixel-level validation set was built by randomly sampling and visually interpreting over 100,000 points for entire China, and a statistical-level validation set for every provincial administrative region in China was derived by collecting the official land resource survey data from the Natural Resources and Planning Bureau of the Chinese government.

### 3.3.1 Generating pixel-level validation sample set across China

As a widely used assessment method for land-cover products, many studies including the 30-meter annual land-cover dataset of China (Yang & Huang 2021) and the impervious surface map of China (Gong, et al., 2019) divided the entire China into numerous grids with the same size and randomly sampled the points in each grid for generating the validation sets. In this paper, China was divided into 171 grids with each size of 3° × 3°, and 800 points in each grid were randomly sampled to generate the national validation sample set for assessing the accuracy of SinoLC-1. After removing the sample points located in the far ocean and outside the nation's borders, 106,852 points remained, and then these sample points were manually annotated by combining the visual interpretation results of VHR imagery captured from Google Earth and HR imagery captured from Sentinel-2 mission to identify their land-cover types. Figure 6 shows the sample grids, legend, and VHR samples of the national validation set, and Figure 7 shows the class proportion comparison between the sample set and the SinoLC-1 product. The land-cover proportion of selected sample points in the validation set is relatively similar to the SinoLC-1 dataset, further indicating that the ~100,000 sample points have reasonable class distribution. Based on the national validation sample set, the quantitative metrics including the user's accuracy (U.A.) (measuring the commission error), producer's accuracy (P.A.) (measuring the omission error), overall accuracy (O.A.), and kappa coefficient can be calculated for assessing the performance of SinoLC-1 comprehensively.

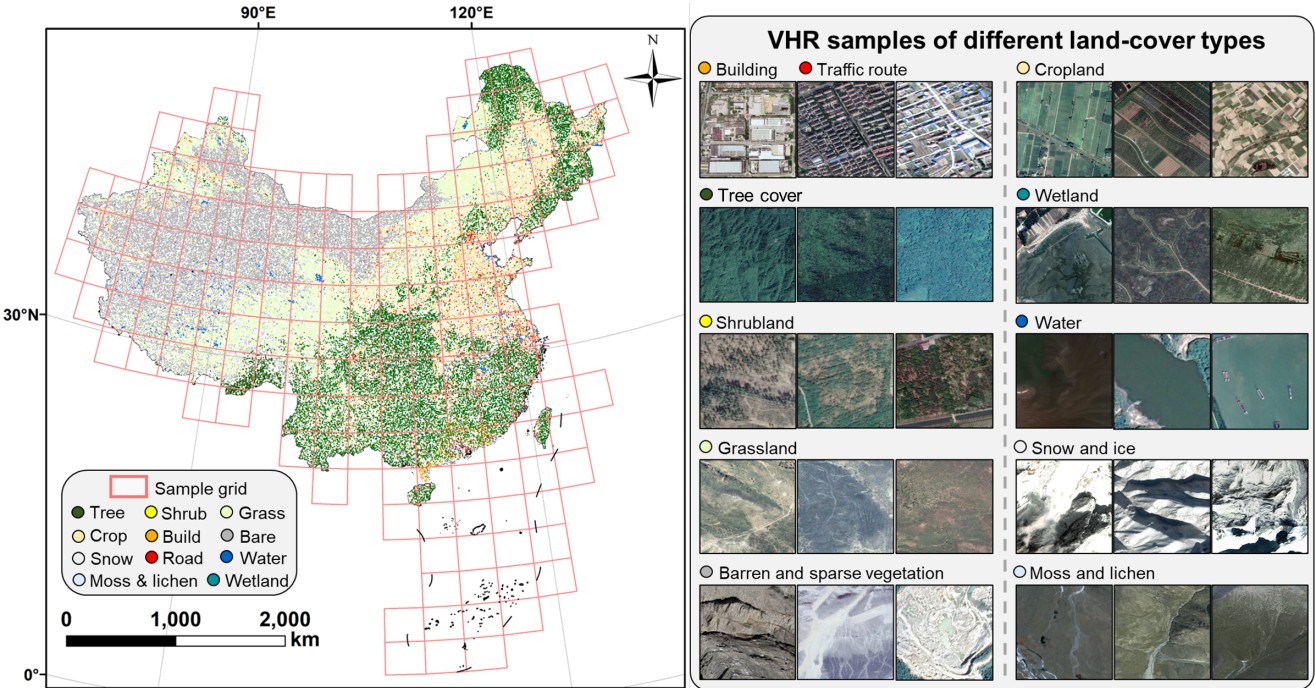

**Figure 6. Demonstration of the sample grid, VHR samples, and the national validation sample set. Left: the spatial distributions of the sample set (the legend is written in shorter forms). Right: the VHR samples of different land-cover types collected from 1.07-m resolution © Google Earth imagery all around China.**

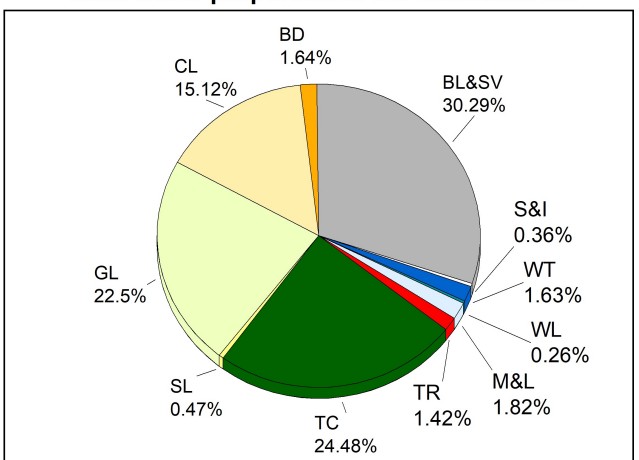

(a) Class proportion of the national validation sample set.

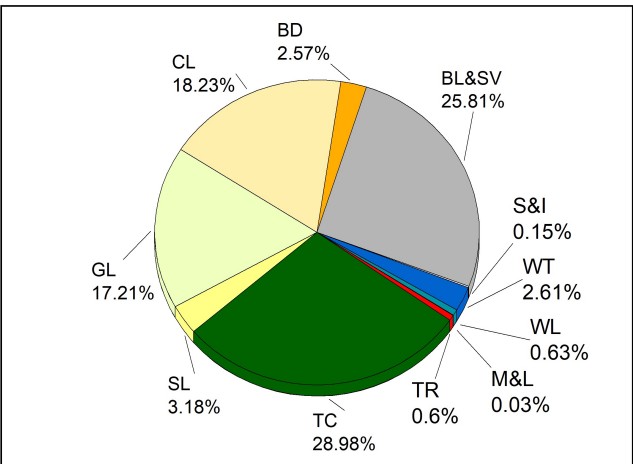

(b) Class proportion of the SinoLC-1 land-cover product

**Figure 7. The land-cover proportion of the national validation sample set and the produced SinoLC-1 land-cover product.**

### 3.3.2 Collecting statistical-level validation set from government survey reports

To assess the statistical-level performance of SinoLC-1 in every provincial administrative region of China, the statistical validation set was collected from the Third National Land Resource Survey Project (abbreviated as 3rd NLRS) from the Ministry of Natural Resources of the People's Republic of China and the Natural Resources and Planning Bureau of every provincial administrative region in China. The NLRS projects were launched since 1984 to monitor urban expansion and land resources comprehensively through remote sensing technology (Zhang & Zhang, 2007; Liu et al., 2015). From Oct. 2017 to Dec. 2020, the 3rd NLRS project adopted remote-sensing images with a resolution better than 1 m to accumulate survey data for the entire China. Advanced technologies such as mobile Internet, cloud computing, and unmanned aerial vehicle were also widely used during the survey. Overall, 295 million survey spot data were collected, and the state of national land use and land cover had been thoroughly investigated. Therefore, the survey report collected from the government institutes can be used as an authoritative reference source to validate the quality of the produced SinoLC-1 at the statistical level.

By considering the classification standard of the 3rd NLRS, the land-cover type relationship between the SinoLC-1 and the 3rd NLRS was built, as shown in Table 3. In the corresponding relationship, the 3rd NLRS data commonly have finer land-cover types, e.g., for the general type "Cropland," six sub-types are in the 3rd NLRS data. However, some of the land-cover types in the 3rd NLRS data were still described in a more generalized way. For example, the 3rd NLRS only contains three subtypes (natural, artificial, and other grasslands) to describe the landscapes that are covered by spare and low vegetation, which correspond to the type of "Grassland" and "Barren and spare vegetation" in SinoLC-1. As shown in Table 4, the statistical validation set was collected from 31 provincial administrative regions, where three special administrative zones (Hongkong, Marco, and Taiwan) are not available in the 3rd NLRS project. In general, the statistical validation set enabled comparing the statistical results of SinoLC-1 with the official survey data collected from the 3rd NLRS projects, and thus, assessing the overall performance of SinoLC-1.

**Table 3. Corresponding land-cover type relationship between the SinoLC-1 products and the 3rd national land survey.**

| SinoLC-1 category | 3rd NLRS land-cover type | SinoLC-1 category | 3rd NLRS land-cover type |
|---|---|---|---|
| Tree cover | Arbor woodland | Building | Urban land |
| | Bamboo groves | | Administrative towns |
| | Other woodland | | Village land |
| Shrubland | Shrubland | | Airport land |
| Grassland Barren and sparse vegetation | Natural grassland | | Wharf land |
| | Artificial grassland | | Pipeline transportation |
| | Other grasslands | | Scenic Spot |
| | Mining land | Wetland | Forest swamp |
| Cropland | Paddy field | | Shrub swamp |
| | Irrigated land | | Swampy grassland |
| | Dry cropland | | Coastal tidal flat |
| | Orchard | | Inland tidal flat |
| | Tea plantation | | Marshland |
| | Rubber plantation | Water | River |
| | Other plantations | | Lake |
| Traffic route | Railway | | Reservoir |
| | Rail transit | | Pond |
| | Highway | | Ditch |
| | Rural road | | Hydraulic construction |
| Snow and ice | Glaciers and snow | Moss and lichen | Tundra |

**Table 4. Statistical validation set collected from the third national land resource survey projects.**

| Geo. region | Province/ City | Statistical results of different land-cover types (km$^2$) | | | | | | | | | |
|---|---|---|---|---|---|---|---|---|---|---|---|
| | | TR | TC | SL | GL+BL&SV | CL | BD | S&I | WT | WL | M&L |
| South | Hainan | 524 | 10799 | 943 | 173 | 17047 | 2468 | 0 | 1831 | 1157 | 57 |
| | Guangxi | 3272 | 124831 | 36122 | 2767 | 49779 | 9857 | 0 | 7490 | 1178 | 94 |
| | Guangdong | 3000 | 106522 | 1404 | 2390 | 32267 | 17757 | 0 | 13423 | 1683 | 106 |
| East | Fujian | 2000 | 87427 | 686 | 753 | 18503 | 7109 | 0 | 3731 | 1874 | 12 |
| | Anhui | 2824 | 40055 | 860 | 483 | 59196 | 17588 | 0 | 17285 | 477 | 0 |
| | Zhejiang | 2268 | 58616 | 2319 | 3 | 20507 | 11559 | 0 | 7025 | 1655 | 1 |
| | Shanghai | 275 | 818 | 1 | 0 | 1772 | 2944 | 0 | 1913 | 727 | 0 |
| | Jiangsu | 3362 | 7787 | 84 | 942 | 43293 | 21103 | 0 | 25426 | 4264 | 0 |
| | Shandong | 3997 | 25383 | 670 | 2379 | 77242 | 28206 | 0 | 13254 | 2463 | 0 |
| Central | Hubei | 3047 | 83936 | 8865 | 898 | 53243 | 14172 | 0 | 19837 | 615 | 0 |
| | Hunan | 3425 | 121363 | 5804 | 18520 | 45150 | 16336 | 0 | 12585 | 2362 | 0 |
| | Henan | 3560 | 37362 | 6601 | 2579 | 79419 | 24495 | 0 | 14445 | 393 | 0 |
| North | Shanxi | 2420 | 43611 | 17346 | 31064 | 45105 | 10185 | 0 | 1731 | 546 | 0 |
| | Hebei | 3666 | 44371 | 19883 | 19492 | 70400 | 21094 | 0 | 5711 | 1428 | 0 |
| | Beijing | 401 | 5977 | 3701 | 146 | 2509 | 3176 | 0 | 618 | 32 | 0 |
| | Inner Mongolia | 21228 | 167115 | 76564 | 543772 | 115508 | 14975 | 0 | 10645 | 38094 | 0 |
| | Tianjin | 453 | 1852 | 0 | 153 | 3296 | 3319 | 0 | 2373 | 327 | 0 |
| Northeast | Liaoning | 2654 | 52080 | 8077 | 4886 | 57100 | 13302 | 0 | 6916 | 2864 | 0 |
| | Jilin | 272 | 15733 | 53 | 86 | 9303 | 1125 | 0 | 1001 | 82 | 0 |
| | Heilongjiang | 5043 | 214459 | 1773 | 11864 | 172578 | 11671 | 0 | 16864 | 35010 | 0 |
| Northwest | Shaanxi | 2804 | 106245 | 18515 | 22109 | 41483 | 9204 | 0 | 2733 | 487 | 0 |
| | Gansu | 1320 | 11968 | 4488 | 149072 | 93632 | 15840 | 0 | 5984 | 10736 | 0 |
| | Xinjiang | 5172 | 40832 | 81293 | 519885 | 81087 | 14163 | 22242 | 30842 | 15245 | 0 |
| | Ningxia | 942 | 9537 | 0 | 20312 | 11984 | 2973 | 0 | 1688 | 249 | 0 |
| | Qinghai | 3125 | 9096 | 36940 | 394727 | 6265 | 4909 | 4233 | 20233 | 51012 | 0 |
| Southwest | Guizhou | 3174 | 79346 | 32755 | 1888 | 34726 | 7751 | 0 | 2554 | 71 | 0 |
| | Chongqing | 1433 | 38067 | 8823 | 237 | 21508 | 6426 | 0 | 2717 | 150 | 0 |
| | Xizang (Tibet) | 1596 | 98180 | 80782 | 800653 | 4540 | 1642 | 20715 | 38589 | 43025 | 0 |
| | Yunnan | 5219 | 220773 | 28917 | 13238 | 79676 | 10773 | 431 | 5654 | 398 | 0 |
| | Sichuan | 4492 | 183471 | 70724 | 96884 | 64302 | 18496 | 459 | 10073 | 12309 | 0 |
| Note: | | TR=Traffic route; TC=Tree cover; SL=Shrubland; GL+BL&SV=the total of 'Grassland' and 'Barren and sparse vegetation'; CL=Cropland; BD=Building; S&I=Snow and ice; WT=Water; WL=Wetland; M&L=Moss and lichen. | | | | | | | | | |

## 4 Results and discussions

### 4.1 SinoLC-1: a 1-meter resolution national-scale land-cover map for China

First, the 1-meter resolution national-scale land-cover map for China (SinoLC-1) and the legend for the containing 11 land-cover types are illustrated in Figure 8. The type of "Tree cover" is mainly located in the southern part and the northeast border of China; the croplands are mainly distributed in the north and northeast China plains; the northwest and southwest parts of China are mainly covered by the types of "Grassland" and "Barren and sparse vegetation". In general, based on previous research and land-cover survey reports of China (Yue et al., 2007; Song & Deng, 2017), the overall visual result of SinoLC-1 accurately reflects the land-cover distribution of China and conforms to the actual land-cover pattern of China.

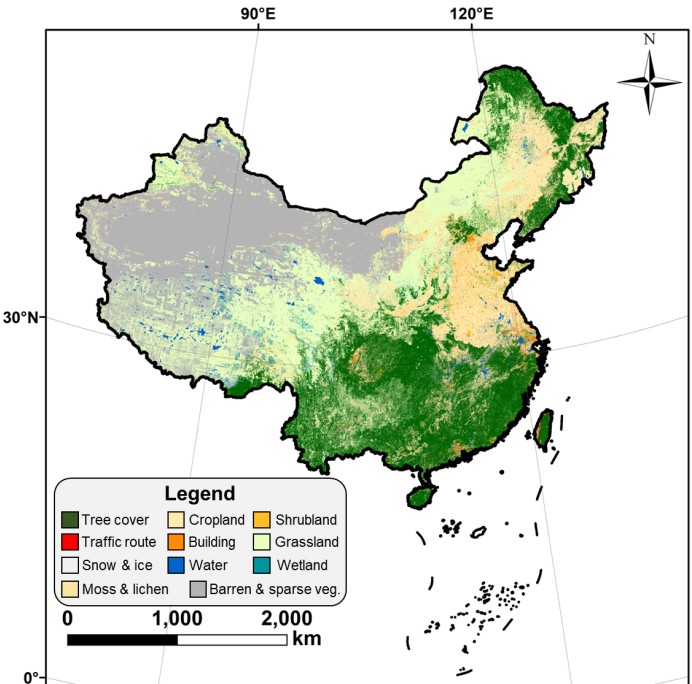

**Figure 8. Demonstration of SinoLC-1: a 1-meter-resolution national-scale land-cover map of China.**

Second, to visualize the results of SinoLC-1 in detail, the 30-meter digital elevation model (DEM) data collected from the Shuttle Radar Topography Mission (SRTM) were illustrated, and three typical regions were selected to demonstrate the performance of the SinoLC-1 product. As shown in Figure 9, the three typical regions include the following: (1) northeastern China, where the northeastern plain (an important grain production base of China) and the Greater Khingan Range, known as the largest virgin forest in China, are located; (2) eastern China, where the northern plain (another important grain production bases of China) and the Yangtze River delta (an important economic zone in China) are located; and (3) southern China, where the Pearl River Delta, known as the largest urban agglomeration with the largest population in the world, is located. In detail, as shown in Figure 10, the sample areas of Heilongjiang, Jilin, and Liaoning Provinces in northeastern China show the boundaries between forest, grassland, and cropland are clearly predicted. As shown in Figure 11 and Figure 12, the sample

areas of eastern China including Shandong, Jiangsu, and Jiangxi Provinces and southern China including Guangxi, Guangdong,

and Hainan Provinces show that the village and city patterns of rural and urban areas are accurately reflected in the SinoLC-1 product. Overall, by combining all the visual results and analysis, the SinoLC-1 performs well in various landscapes (e.g., forest, cropland, rural, and urban) and shows acceptable results at the national and regional scales.

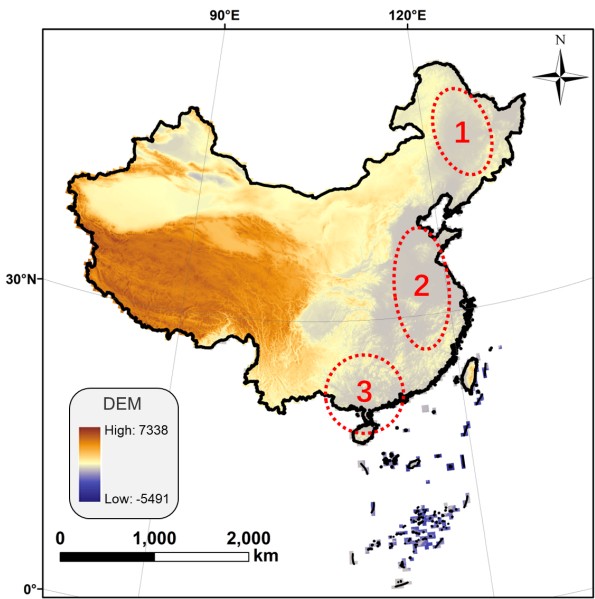

**Figure 9. Illustration of the 30-meter DEM data (from SRTM) and the locations of three demonstration areas.**

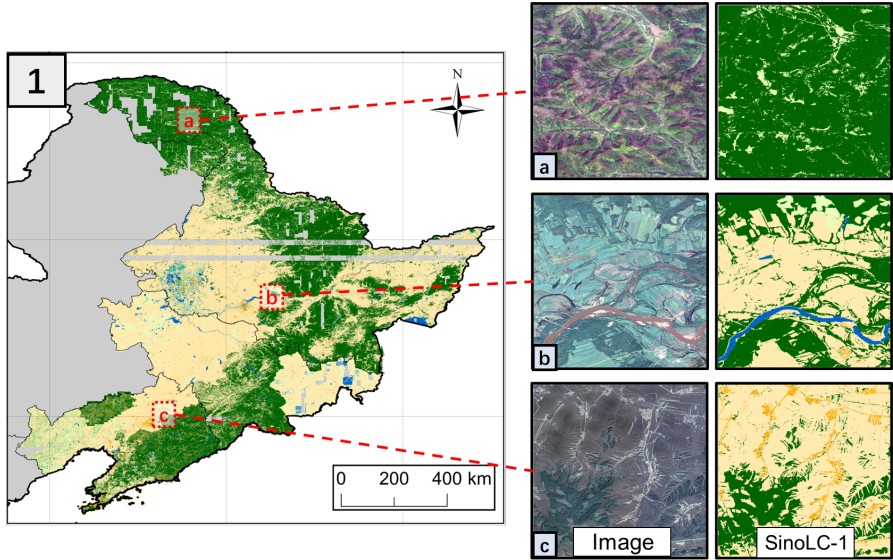

**Figure 10. Demonstration of northeastern China including the sample areas of Heilongjiang, Jilin, and Liaoning. The VHR remote sensing images in the figure are from © Google Earth 2021.**

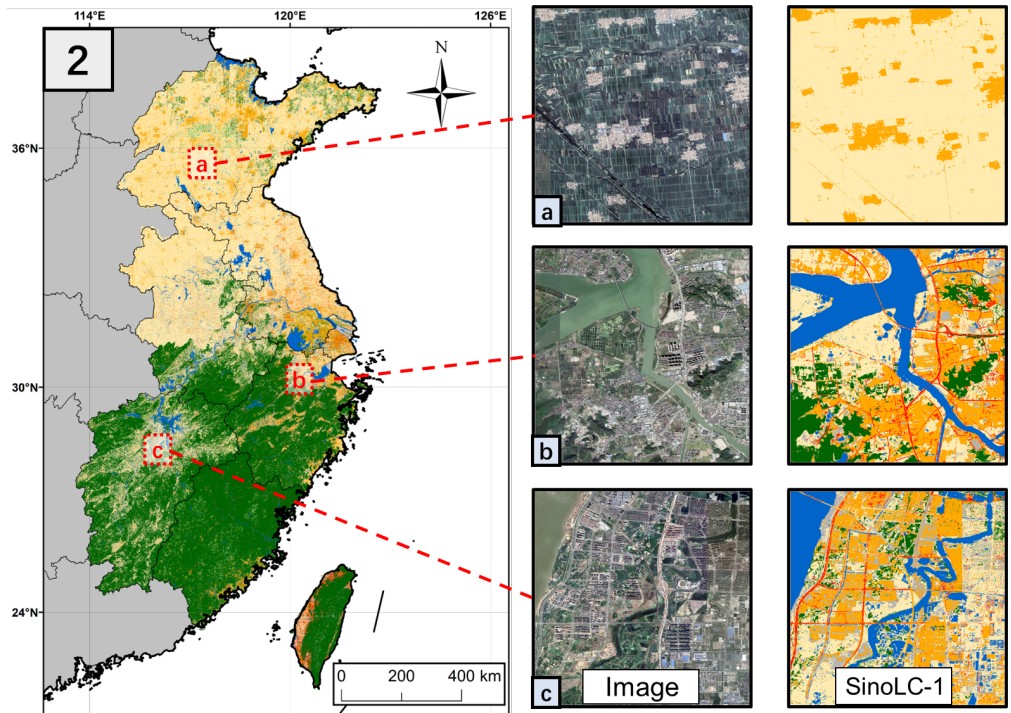

**Figure 11. Demonstration of Eastern China including the sample areas of Shandong, Jiangsu, and Jiangxi. The VHR remote sensing images in the figure are from © Google Earth 2021.**

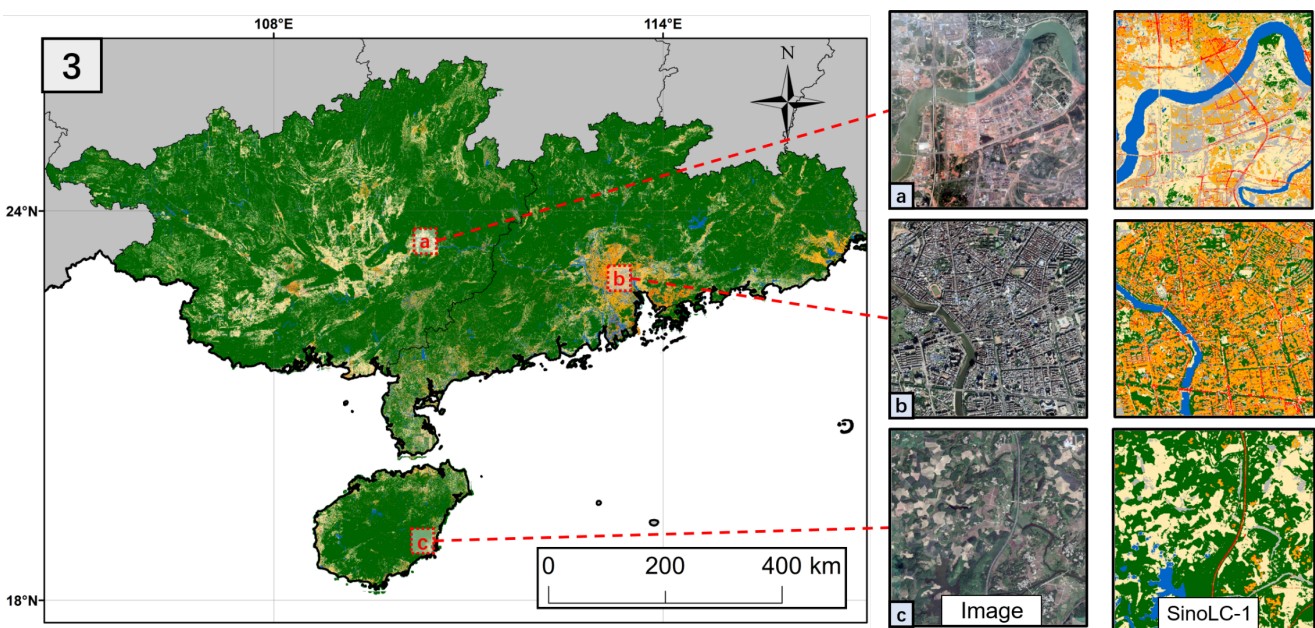

**Figure 12. Demonstration of Southern China including the sample areas of Guangxi, Guangdong, and Hainan. The VHR remote sensing images in the figure are from © Google Earth 2021.**

## 4.2 Qualitative comparison with other land-cover products

To assess the SinoLC-1 land-cover product qualitatively, the produced SinoLC-1 and five widely used large-scale land-cover products were visually compared. The comparison land-cover products included ESA_GLC10 (Van De Kerchove et al., 2021), FROM_GLC10 (Gong et al., 2019), ESRI_GLC10 (Karra et al., 2021), GLC_FCS30 (X Zhang et al., 2021), and GlobeLand30 (Chen et al., 2015). The information for these comparison products is listed in Table 5. As shown in Figure 13 and Figure 14, five typical regions covering various landscapes and different land-cover patterns were selected to compare the performance of SinoLC-1 with the five land-cover products.

**Table 5. Information for the comparative land-cover products.**

| Name | Resolution | Version & Timeline | Number of land-cover type |
|---|---|---|---|
| ESA_GLC10 | 10m | v2020 | 11 |
| FROM_GLC10 | 10m | v2017 | 10 |
| ESRI_GLC10 | 10m | v2020 | 10 |
| GLC_FCS30 | 30m | v2020 | 16 |
| GlobeLand30 | 30m | v2020 | 10 |

First, Figure 13 illustrates a large-scale comparison in Changzhou City, Jiangsu Province, where the region contains balance and various land-cover types. From the qualitative comparison, ESRI_GLC10 in Figure 13 (e) and GlobeLand30 in Figure 13 (g) have blurred land-cover results according to the VHR image in Figure 13 (a), where the detailed land object located in the urban areas (i.e., the tree cover, building, and cropland) are confused. Moreover, the SinoLC-1, ESA_GLC10, FROM_GLC10, and GLC_FCS30 show relatively accurate spatial distributions of the land-cover types. Among them, GLC_FCS30 shows limited performance in tree cover and slender land objects (i.e., traffic routes, rivers, and runoff). FROM_GLC10 shows accurate performance for water bodies (e.g., the pools, canals, and rivers) but has limited performance in the type of tree cover. ESA_GLC10 shows relatively better results among other comparative products, but it still shows insufficient visualization in water bodies. Compared with these GLC products, the SinoLC-1 comprehensively shows better performance where the fine land-cover details including slender rivers, runoff, small pools, vegetation, and building are well predicted. Furthermore, because the land-cover type of "Traffic route" is also included in the SinoLC-1 products, the road networks can better reflect the traffic pattern and city layout of the region.

Second, Figure 14 illustrates four other typical regions, which were sampled from four provincial administrative regions including Shanghai, Jiangxi, Guangdong, and Hainan. Similarly, ESRI_GLC10 and GlobeLand30 show limited performances and lose the land-cover details. By comparing the urban areas shown in Figure 14 (a) and (b) (i.e., the demonstration areas of Shanghai and Jiangxi), the SinoLC-1 indicates more accurate land-cover details, where some of the slender roads that cannot be observed in the 10-meter-resolution land-cover products are well predicted in the 1-meter-resolution SinoLC-1 products.

The comparison suggests the 1-meter SinoLC-1 can be a better land-cover product in indicating the finer urban pattern and providing more accurate information to the users. By comparing the agricultural areas (e.g., fish ponds and paddy fields) in Figure 14 (c) and (d) (i.e., the demonstration areas of Guangdong and Hainan), ESRI_GLC10 and GlobeLand30 overestimate the water bodies and misguide the real land-cover situation, where many independent fish ponds and paddy fields are incorrectly mapped as a large water-cover area. On the contrary, ESA_GLC10 and GLC_FCS30 underestimate the water bodies, where most of the ponds are not indicated in their mapping results. SinoLC-1 and FROM_GLC10 indicate the most accurate land-cover situations, where all single ponds are mapped. However, due to the limitation of the spatial resolution, FROM_GLC10 still loses partial land-cover details located around ponds and fields (e.g., traffic route and tree cover).

Third, Figure 15 demonstrates three special landscapes that are challenging to distinguish in VHR optical images and even HR multispectral images. The three landscapes include (a) Marshland (i.e., muddy areas with dense water and grass that have been soaked in stagnant water) captured from the Daqing Longfeng Wetland Nature Reserve, Heilongjiang Province, which is the largest urban wetland in China, (b) Forest swamp (i.e., the landscape dominated by trees or shrubs formed under humid soil, stagnant water, or shallow water layers) captured from Chongming island, Shanghai City, which is known as the world's largest estuarine alluvial island wetland, and (c) Watercourse (the route through which river water flows, usually referring to navigable waterways) captured from the Beijing-Hangzhou Grand Canal. As shown in Figure 15 (a), the SinoLC-1 reveals most of the marshland in the area and distinguishes the surrounding water and grasslands. Among the three 10-m land-cover products generated from the Sentinel image, the ESA_GLC10 accurately reflects the marshland in the area, but the FROM_GLC10 and ESRI_GLC10 miss the majority of wetland type. As shown in Figure 15 (b), it is observed that the VHR optical image shows more clear spatial detail than the 10-meter Sentinel-2 image. From the perspective of the land-cover map, the SinoLC-1 shows the forest swamp (i.e., land cover type of wetland in the legend), rivers, and tree cover content in the area. The ESRI_GLC10 shows an accurate result on the forest swamp landscape. The ESA_GLC10 overestimates the tree cover type, and the FROM_GLC10 overestimates the cropland. As shown in Figure 15 (c), the SinoLC-1 accurately reflects the watercourse, and due to the fine spatial resolution, the bridges on the watercourse are also clearly displayed. Among the three 10-m land-cover products generated from the Sentinel image, the ESRI_GLC10 and FROM_GLC10 have acceptable classification results on the watercourse. However, the FROM_GLC10 only shows the central part of the watercourse and underestimates the width. For ESA_GLC10, the watercourse was incorrectly classified into the land-cover type of "Barren and sparse vegetation".

Overall, by comparing the SinoLC-1 product with five widely used land-cover products in many typical regions, the produced SinoLC-1 shows three main advantages: (1) With higher spatial resolution, the SinoLC-1 can reflect finer land objects and indicates more precise land details. (2) With more diverse and reliable training samples, the SinoLC-1 shows more accurate spatial distributions in land-cover types. (3) With the additional land-cover type "Traffic route," the SinoLC-1 can better outline the traffic network and city layout in dense urban areas.

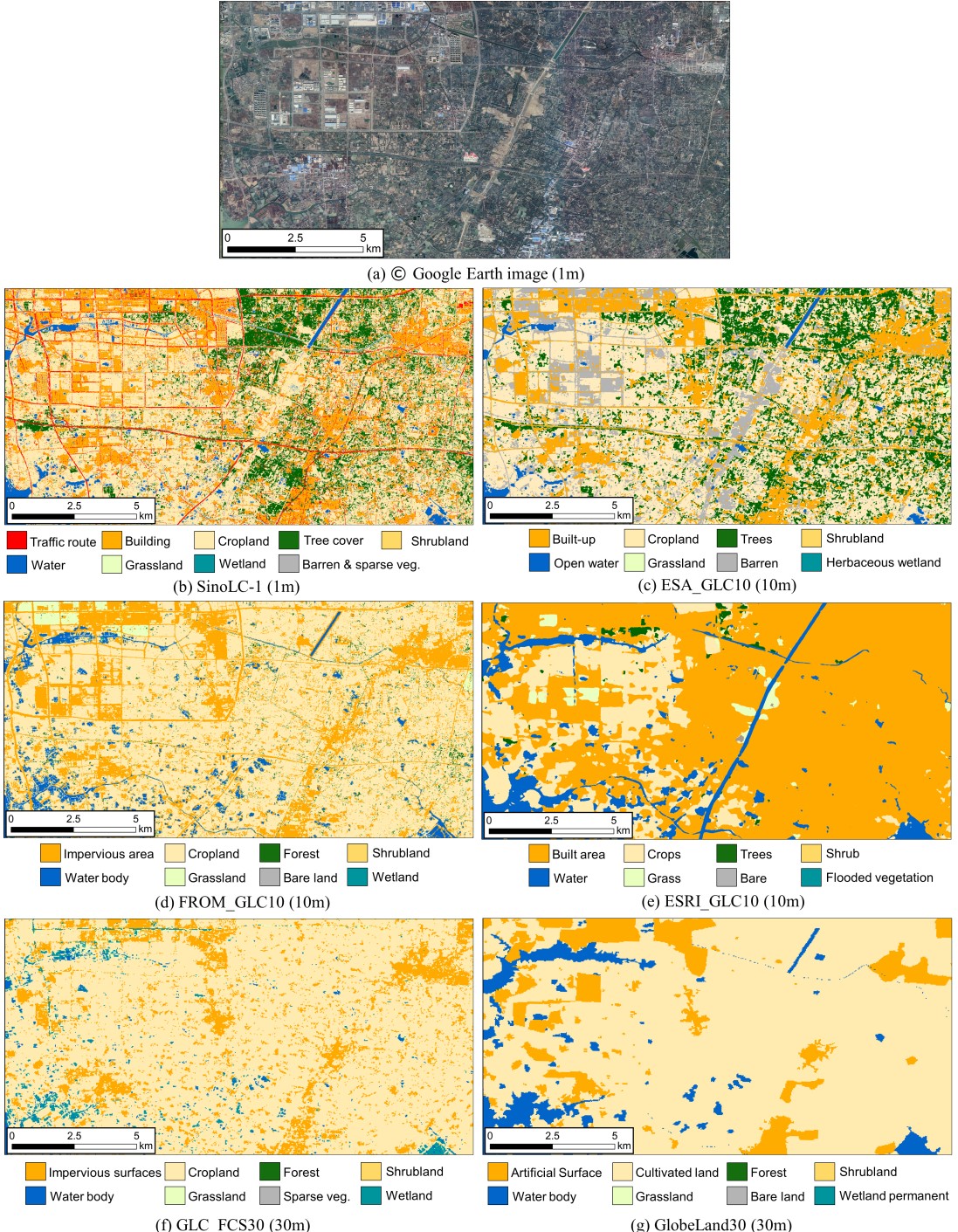

(a) © Google Earth image (1m)

**Legend (b) SinoLC-1:** Traffic route, Building, Cropland, Tree cover, Shrubland, Water, Grassland, Wetland, Barren & sparse veg.

(b) SinoLC-1 (1m)

**Legend (c) ESA_GLC10:** Built-up, Cropland, Trees, Shrubland, Open water, Grassland, Barren, Herbaceous wetland

(c) ESA_GLC10 (10m)

**Legend (d) FROM_GLC10:** Impervious area, Cropland, Forest, Shrubland, Water body, Grassland, Bare land, Wetland

(d) FROM_GLC10 (10m)

**Legend (e) ESRI_GLC10:** Built area, Crops, Trees, Shrub, Water, Grass, Bare, Flooded vegetation

(e) ESRI_GLC10 (10m)

**Legend (f) GLC_FCS30:** Impervious surfaces, Cropland, Forest, Shrubland, Water body, Grassland, Sparse veg., Wetland

(f) GLC_FCS30 (30m)

**Legend (g) GlobeLand30:** Artificial Surface, Cultivated land, Forest, Shrubland, Water body, Grassland, Bare land, Wetland permanent

(g) GlobeLand30 (30m)

**Figure 13. Demonstration of the visual comparison for Changzhou City, Jiangsu Province. The VHR remote sensing image in the figure is from © Google Earth 2021.**

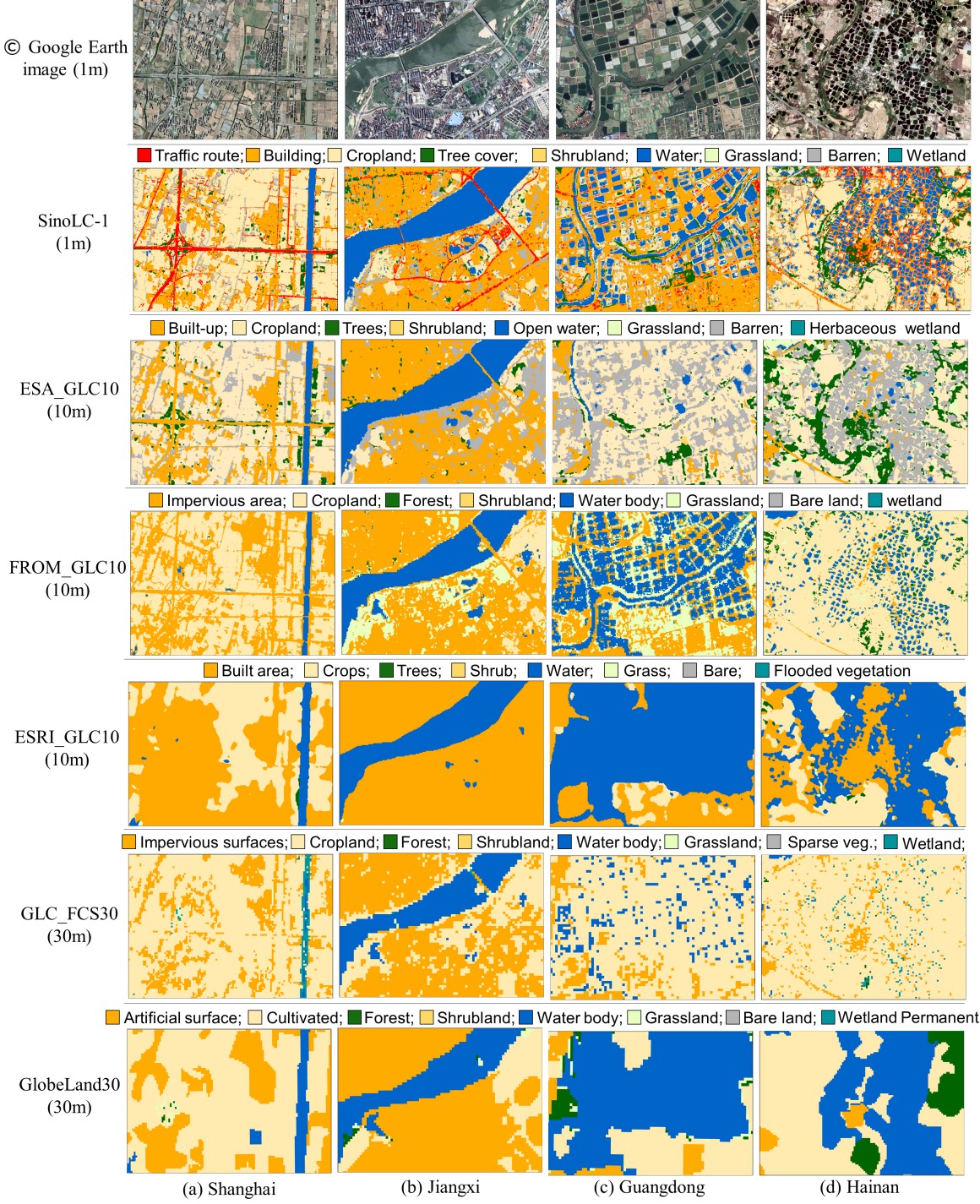

Figure 14. Demonstrations of the visual comparison for four typical regions. The VHR remote sensing images in the figure are from © Google Earth 2021.

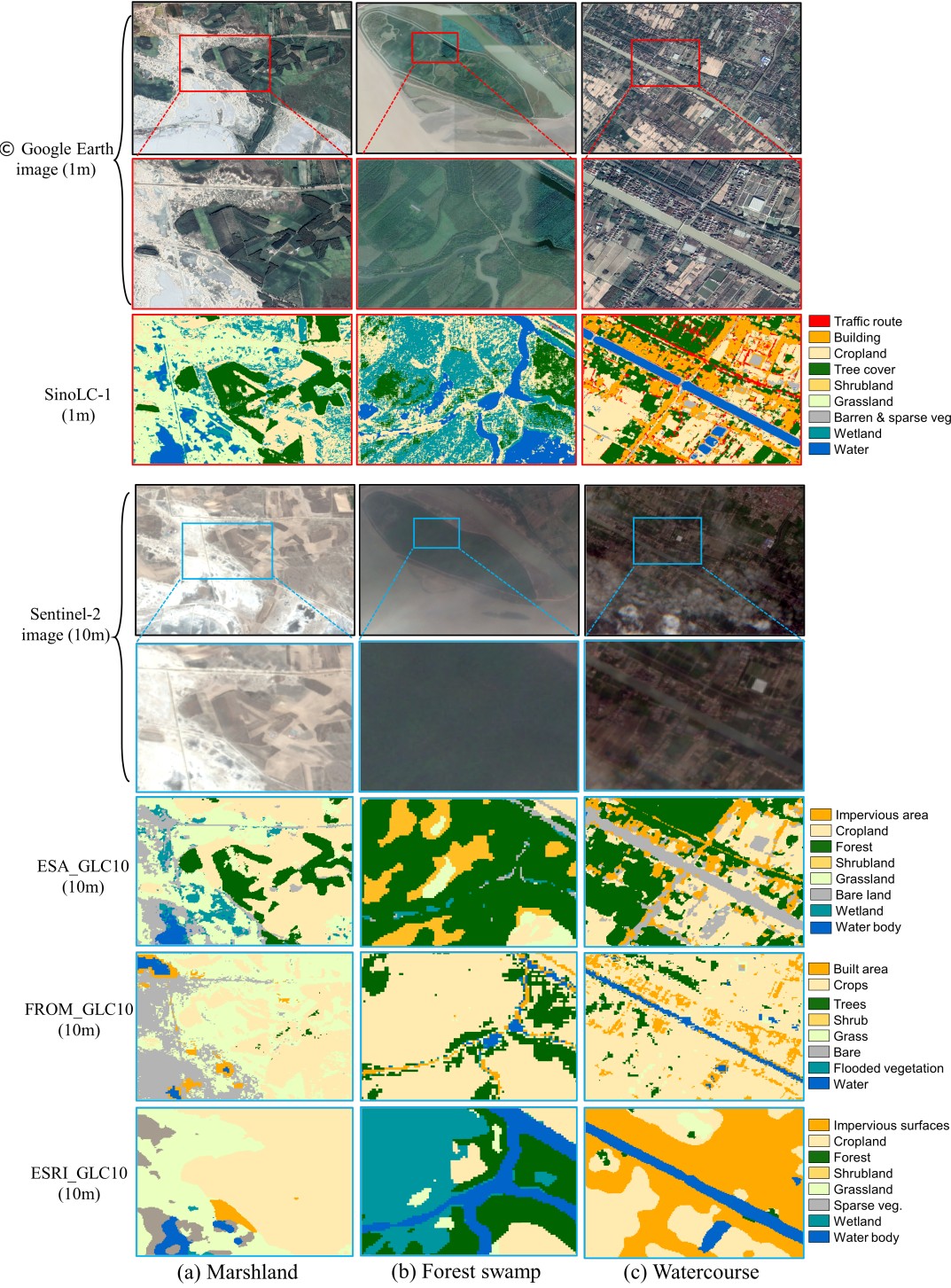

**Figure 15. Demonstrations of the visual comparison for challenging land-cover types which include (a) Marshland, (b) Forest swamp, and (c) Watercourse. The VHR remote sensing images in the figure are from © Google Earth 2021.**

## 4.3 Quantitative analysis and accuracy assessment

### 4.3.1 Pixel-level validation

Based on the national validation sample set introduced in Sect. 3.3.1, over 100,000 sample points were visually interpreted to validate the accuracy of the SinoLC-1 land-cover product quantitatively. First, as a widely used method of assessing the accuracy of land-cover maps (Foody & Mathur, 2004; Gómez et al., 2016; Olofsson et al., 2014), the overall confusion matrix is shown in Table 6, and the confusion proportions for each land-cover type is demonstrated in Figure 16. With the confusion matrix, the O.A. and kappa coefficients were calculated to measure the overall performance of the SinoLC-1 product. Then, the U.A. and P.A. were calculated to measure the commission and omission errors of the product. Furthermore, as shown in Table 7, the number of samples, coverage area, O.A., and kappa coefficients of every provincial administration region were listed to demonstrate the accuracy of SinoLC-1 in different regions. It is important to notice that the "Number of samples" of Table 7 represents the sample counts of the generated validation set where some points may locate in the void value of the VHR images and SinoLC-1 product. The spatial distribution of the O.A. of every provincial administration region and the statistical accuracy of every geographical region are shown in Figure 17.

The confusion matrix in Table 6 shows the SinoLC-1 land-cover product achieves an O.A. of 73.61% and a kappa coefficient of 0.6595. Due to the void value of images and land-cover results in some regions, 106,344 validation sample points were ultimately counted in the confusion matrix. In terms of P.A., the land-cover type of "Water" has the highest accuracy (86.1%), followed by "Tree cover", "Barren and spare vegetation", "Grassland", "Cropland", and "Building"; however, the land-cover type of "Shrubland", "Wetland", "Moss and lichen", "Snow and ice", and "Traffic route" have relative low accuracies. By combining the class proportion of the validation sample set shown in Figure 7 and the confusion matrix shown in Table 6 and Figure 16, the quantitative results of the basic land-cover types (i.e., the types of "Tree cover", "Grassland", "Cropland", "Barren and sparse vegetation", and "Water"), which have easily distinguishable features and occupy a large area in China, report higher accuracies and have a small proportion of misclassification. By contrast, the land-cover types (i.e., the types of "Traffic route", "Moss and lichen", and "Snow and ice"), which occupy a small area, obtain relatively low accuracies and have a large proportion of misclassification.

The confusion proportion in Figure 16 shows three points. First, partial traffic routes are incorrectly classified into a few common land-cover types (e.g., "Tree cover", "Cropland", and "Grassland") because the models incorrectly predict the road width; thus, other land objects distributed on both sides of the roads cause commission errors. Second, most of the types including "Tree cover", "Shrubland", "Grassland", "Cropland", "Building", "Barren and spare vegetation", "Wetland", and "Water" are well predicted and only contain a small proportion of the commission errors. Third, the land-cover types of "Snow and ice" and "Moss and lichen" are commonly distributed in the northwest region of China, so the confusing land-cover types are mainly the types of "Grassland" and "Barren and sparse vegetation", which are the most confusable and occupy a large proportion of northwestern China.

The O.A. and kappa coefficients of every provincial administrative region in Table 7 and Figure 17 show the following findings. First, by comparing the spatial distribution of O.A in China, most of the provinces have an O.A. of over 70%, where eight provinces (Hainan, Taiwan, Jiangxi, Fujian, Yunnan, Chongqing, Xinjiang, and Heilongjiang) achieve over 80%. Hebei and Beijing have relatively low O.A. (in the range of 50%–60%). Second, by comparing every geographical region shown in
Figure 17 (b), southern and northeastern China have the highest O.A. among other regions (about 78%) because the land-cover type of "Tree cover" occupies a very large proportion and the land-cover patterns in southern and northeastern China are relatively simple. Northern China including Beijing, Tianjin, Hebei, Shanxi, and Inner Mongolia have the lowest O.A. (lower than 70%). For Inner Mongolia, the wide longitude span of the region and the diverse landscapes caused the misclassification of the region. For Beijing, most of the misclassified samples are (1) the confusion between "Tree cover" and "Grassland"; (2)
the confusion between "Building" and "Traffic route". For Tianjin, most of the misclassification is the confusion among "Cropland", "Building", and "Traffic route". For Hebei, most of the misclassified samples are (1) the confusion between "Tree cover" and "Grassland"; (2) the confusion between "Cropland" and "Grassland". For Shanxi, most of the misclassified samples are (1) the confusion among "Tree cover", "Grassland", and "Cropland"; (2) the confusion between "Building" and "Traffic route"; (3) the confusion between "Cropland" and "Barren & sparse vegetation". Moreover, except for Northern China, the
rest of the geographical regions have accuracies of over 70%.

**Table 6. Confusion matrix for the SinoLC-1 land-cover product according to the national validation sample sets.**

| Classification | TR | TC | SL | GL | CL | BD | BL&SV | S&I | WT | WL | M&L | Total | P.A. (%) |
|---|---|---|---|---|---|---|---|---|---|---|---|---|---|
| Traffic route | 447 | 173 | 5 | 209 | 184 | 228 | 240 | 0 | 28 | 0 | 0 | 1514 | 29.52 |
| Tree cover | 37 | 20708 | 14 | 2713 | 1899 | 124 | 134 | 0 | 352 | 5 | 52 | 26038 | 79.53 |
| Shrubland | 0 | 25 | 270 | 74 | 27 | 2 | 102 | 0 | 1 | 0 | 0 | 501 | 53.89 |
| Grassland | 9 | 1332 | 35 | 17256 | 1837 | 119 | 2848 | 0 | 75 | 11 | 401 | 23923 | 72.13 |
| Cropland | 53 | 1310 | 45 | 1976 | 11424 | 275 | 857 | 0 | 119 | 16 | 0 | 16075 | 71.07 |
| Built-up | 57 | 83 | 3 | 72 | 274 | 1128 | 122 | 0 | 8 | 0 | 0 | 1747 | 64.57 |
| Barren &Sparse veg. | 50 | 209 | 23 | 5643 | 1031 | 418 | 24546 | 3 | 93 | 1 | 194 | 32211 | 76.20 |
| Snow & ice | 0 | 2 | 0 | 94 | 7 | 0 | 51 | 135 | 2 | 0 | 92 | 383 | 35.25 |
| Water | 2 | 21 | 0 | 39 | 105 | 12 | 59 | 0 | 1493 | 1 | 2 | 1734 | 86.10 |
| Wetland | 0 | 37 | 11 | 46 | 28 | 3 | 7 | 0 | 14 | 135 | 0 | 281 | 48.04 |
| Moss & lichen | 0 | 22 | 2 | 698 | 18 | 2 | 455 | 2 | 5 | 0 | 733 | 1937 | 37.84 |
| Total | 655 | 23922 | 408 | 28820 | 16834 | 2311 | 29421 | 140 | 2190 | 169 | 1474 | 106344 | |
| U.A. (%) | 68.24 | 86.56 | 66.18 | 59.88 | 67.86 | 48.81 | 83.43 | 96.43 | 68.17 | 79.88 | 49.73 | | |
| O.A. (%) | | | | | | 73.61 | | | | | | | |
| Kappa | | | | | | 0.6595 | | | | | | | |

Note:     TR=Traffic route; TC=Tree cover; SL=Shrubland; GL=Grassland; CL=Cropland; BD=Building; BL&SV=Barren and sparse vegetation; S&I=Snow and ice; WT=Water; WL=Wetland; M&L=Moss and lichen.

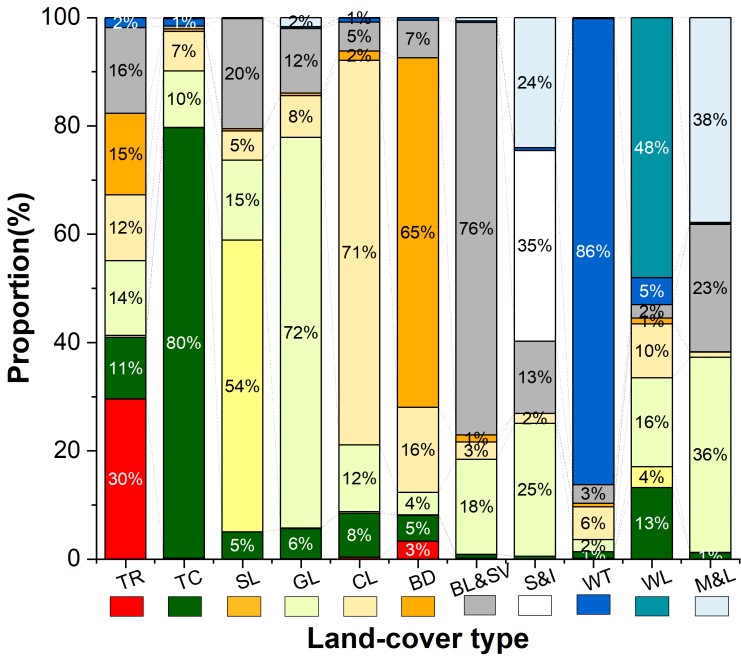

**Figure 16. Confusion proportions for each land-cover type in the SinoLC-1 validation scheme.**

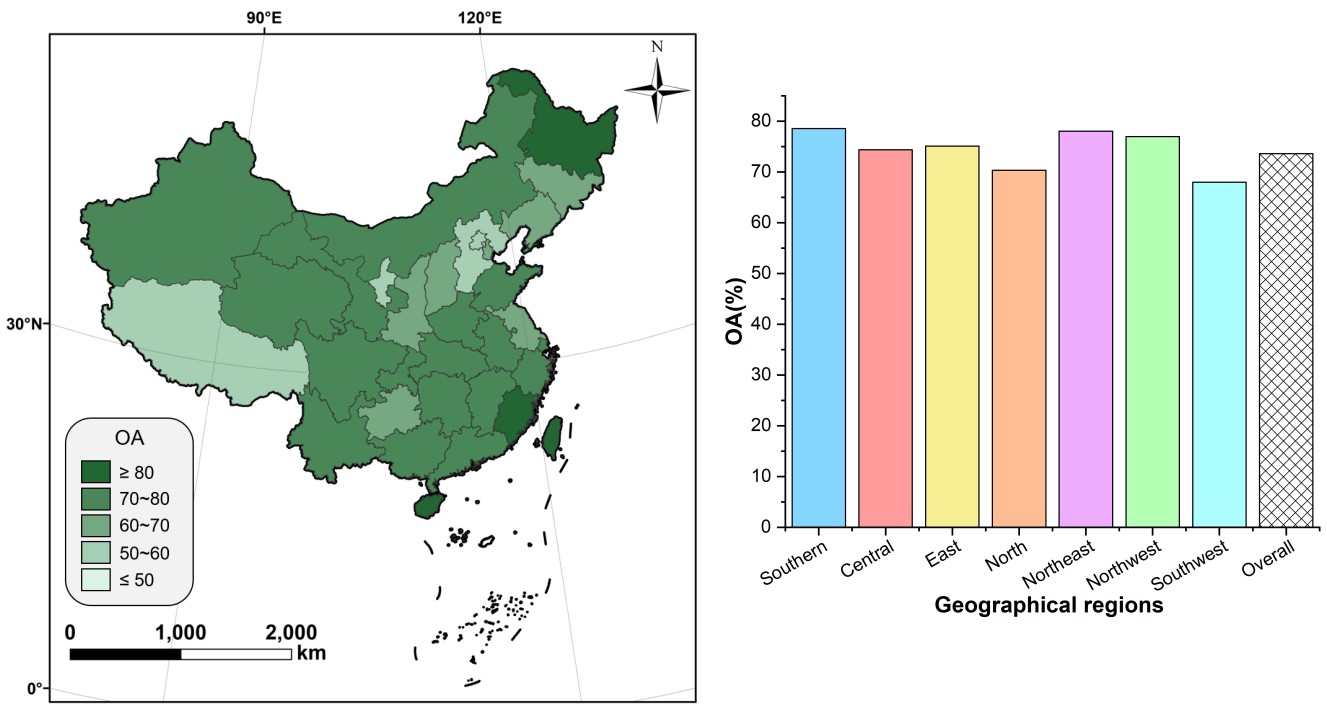

(a) Spatial distribution of O.A. for every province     (b) Statistical O.A. for every geographical region of China

**Figure 17. Spatial distribution and the statistical results of overall accuracy all around China.**

**Table 7. Number of samples, coverage area, O.A., and Kappa coefficient of provincial administrative regions in China.**

| Geographical region | Provincial region | Number of samples | Provincial proportion to China's coverage (%) | O.A. (%) | Kappa coefficient |
|---|---|---|---|---|---|
| South | Hainan | 314 | 0.37 | 82.41 | 0.6404 |
| | Guangxi | 2260 | 2.50 | 81.83 | 0.6346 |
| | Guangdong | 1737 | 1.89 | 73.60 | 0.5923 |
| East | Fujian | 1222 | 1.31 | 83.39 | 0.5202 |
| | Anhui | 1548 | 1.48 | 72.64 | 0.6827 |
| | Zhejiang | 1091 | 1.11 | 76.59 | 0.7022 |
| | Shanghai | 81 | 0.07 | 60.78 | 0.6541 |
| | Jiangsu | 1068 | 1.13 | 66.41 | 0.5904 |
| | Taiwan | 380 | 0.38 | 85.28 | 0.6382 |
| | Jiangxi | 1713 | 1.76 | 80.04 | 0.6555 |
| | Shandong | 1767 | 1.64 | 74.19 | 0.6366 |
| Central | Hubei | 1989 | 1.96 | 73.92 | 0.6538 |
| | Hunan | 2162 | 2.23 | 76.03 | 0.6444 |
| | Henan | 1755 | 1.75 | 72.75 | 0.6573 |
| North | Shanxi | 1700 | 1.65 | 65.81 | 0.6318 |
| | Hebei | 2227 | 1.99 | 58.10 | 0.5463 |
| | Beijing | 211 | 0.17 | 55.55 | 0.5431 |
| | Inner Mongolia | 14297 | 12.47 | 73.00 | 0.7457 |
| | Tianjin | 111 | 0.13 | 63.68 | 0.5961 |
| Northeast | Liaoning | 1723 | 1.56 | 65.94 | 0.6267 |
| | Jilin | 2357 | 0.29 | 65.98 | 0.5771 |
| | Heilongjiang | 6117 | 4.98 | 86.04 | 0.8921 |
| Northwest | Shaanxi | 2282 | 2.17 | 62.08 | 0.5927 |
| | Gansu | 4879 | 4.49 | 77.58 | 0.7878 |
| | Xinjiang | 19448 | 17.54 | 79.64 | 0.5799 |
| | Ningxia | 587 | 0.70 | 61.15 | 0.5688 |
| | Qinghai | 7728 | 7.61 | 75.36 | 0.6817 |
| Southwest | Guizhou | 1780 | 1.86 | 67.25 | 0.5969 |
| | Chongqing | 869 | 0.87 | 79.54 | 0.5016 |
| | Xizang (Tibet) | 12681 | 12.68 | 61.06 | 0.5487 |
| | Yunnan | 3787 | 4.15 | 72.53 | 0.6191 |
| | Sichuan | 4981 | 5.12 | 80.24 | 0.8290 |

## 4.3.2 Quantitative comparison based on open-access validation sets

To compare the SinoLC-1 land-cover product with the other land-cover products quantitatively, we conducted a complete validation to the SinoLC-1 and the other five land-cover products based on two open-access validation datasets (Zhao et al., 2014; Liu et al., 2019). These validation datasets were created based on multiple data sources and manual verification, reporting a stable quality and high independence. Their spatial distribution and classification system are shown in Figure 18.

Based on two open-access validation datasets, we calculated the confusion matrix of SinoLC-1 and further validated the O.A., and kappa coefficient of the SinoLC-1. The O.A. of the SinoLC-1 validated on the validation sets created by Liu et al. and Zhao et al. are 78.80% and 64.69%, respectively. The Kappa coefficients are 0.7394 and 0.5588, respectively. To illustrate more detailed assessment results, Figure 19 shows the corresponding confusion proportions for each considered land-cover type of the SinoLC-1 validated on two datasets. Furthermore, Figure 20 shows the validation results of five comparative land-cover products. Comparing the validation results of two datasets, all products have a higher O.A. on the validation set created by Liu et al., where the SinoLC-1 ranks second with an O.A. of 78.81% (lower than the 30-meter GLC_FCS30). With the validation set created by Zhao et al, all products have an O.A. of around 60%, while the SinoLC-1 ranks second with an O.A. of 64.69% (lower than the 10-meter ESA_GLC10).

Overall, by quantitatively comparing the SinoLC-1 product with five widely used land-cover products on two open-access validation datasets, the produced SinoLC-1 shows acceptable confusion proportion among all considered land-cover types and has competitive accuracy among the other land-cover products across China.

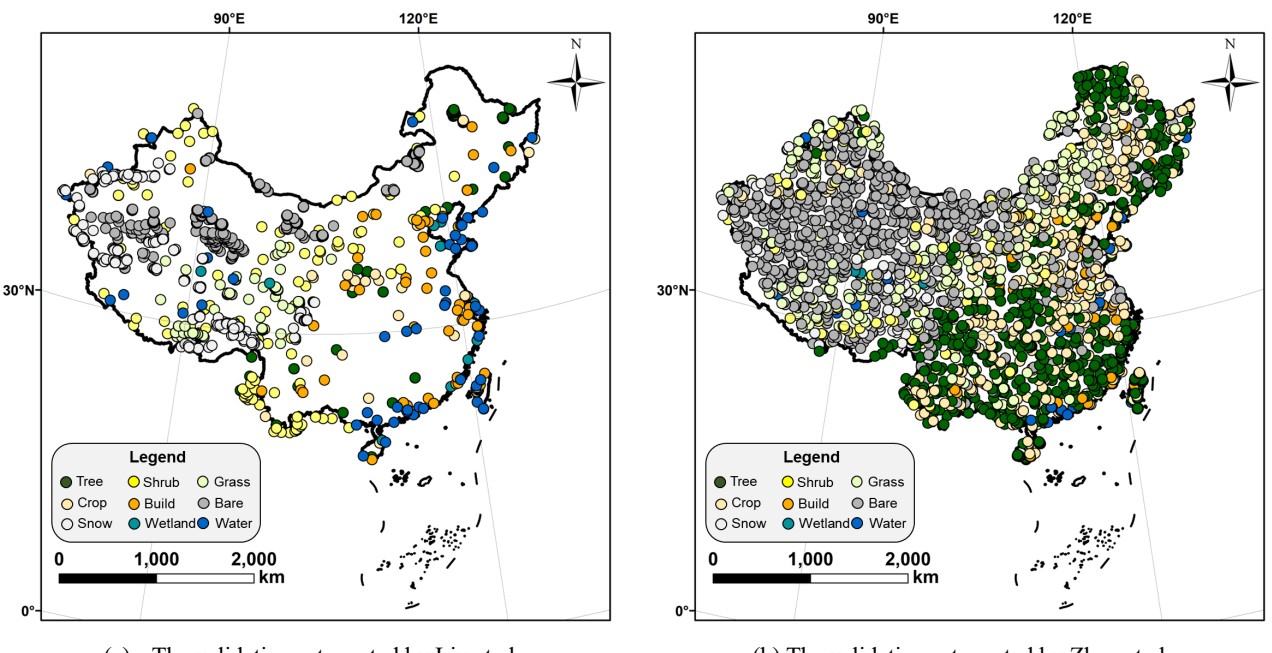

(a) The validation set created by Liu et al.          (b) The validation set created by Zhao et al.

**Figure 18. Spatial distribution and classification system of two open-access validation sets.**

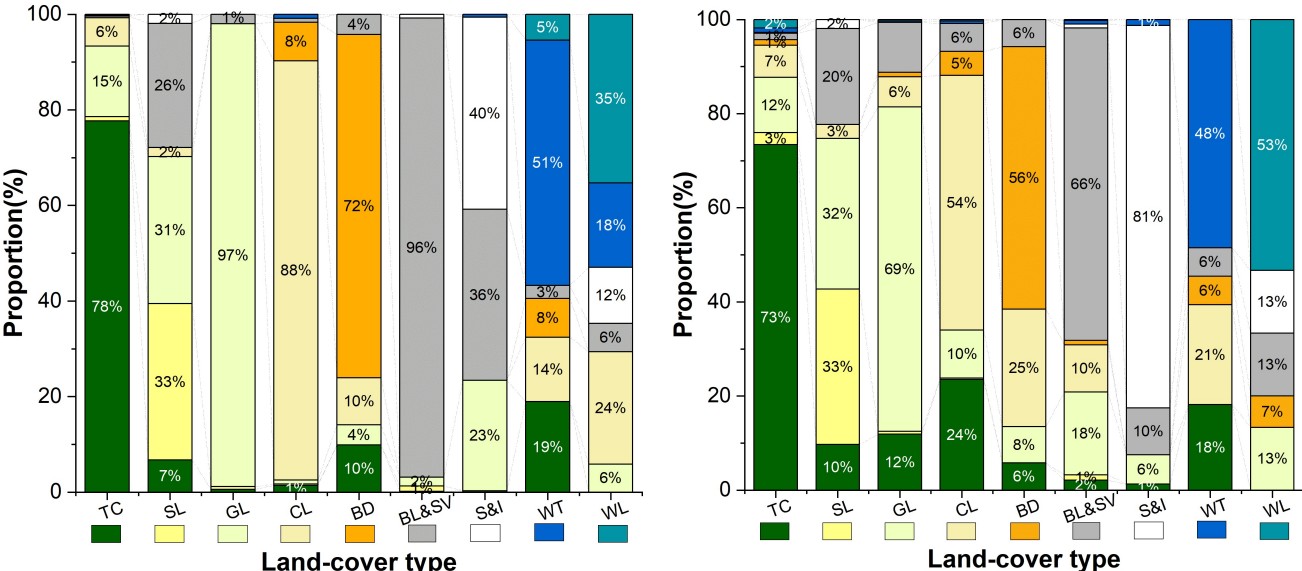

(a) Confusion proportions of the SinoLC-1 validated with the set created by Liu et al.

(b) Confusion proportions of the SinoLC-1 validated with the set created by Zhao et al.

**Figure 19. Confusion proportions of the SionLC-1 with two open-access validation datasets.**

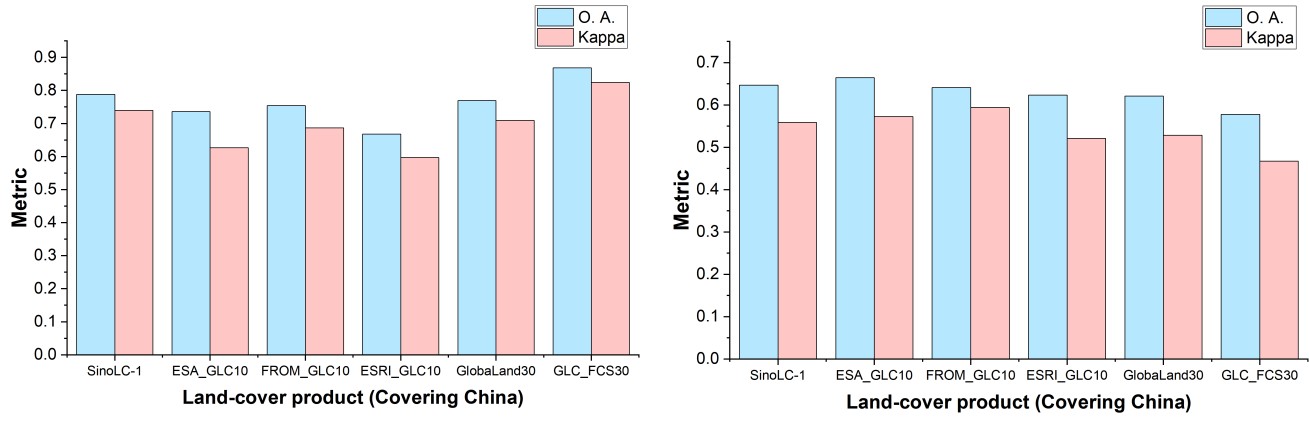

(a) The validation results based on the dataset created by Liu et al.

(b) The validation results based on the dataset created by Zhao et al.

**Figure 20. The quantitative validation and comparison of the SinoLC-1 and the other five land-cover products**

### 4.3.3 Statistical-level validation

Based on the statistical validation set described in Sect. 3.3.2, the official land resource survey data of 31 provincial administrative regions were collected to validate the statistical-level performance of SinoLC-1, as shown in Table 3 and Table 4. Figure 21 compares the statistical results of all considered land-cover types between the SinoLC-1 and 3rd NLRS data in every considered provincial administrative region where the overestimation (positive value) and underestimation (negative value) of SinoLC-1 are reflected. Furthermore, the statistical analysis among the provincial- and geographical-level regions is shown in Figure 23.

The statistical comparisons in Figure 21 reveal the statistical results of most regions are relatively consistent with the 3rd NLRS data. Overall, in southern and central China, the misestimation of land-cover types is mainly distributed in "Tree cover" and "Cropland". In eastern China, the over forecast of the cropland is the main confusion for the SinoLC-1 product, which is evident in Shandong, Anhui, and Jiangsu provinces. In northern China, the statistical comparisons indicate similar conclusions to the pixel-level validation discussed in Sect. 4.3.1. The landscapes vary and easily lead to incorrect predictions due to the wide longitude span of the regions. The misestimation of land-cover types in northern China is mainly the underestimation of shrubland and the over forecast of grassland, barren and sparse vegetation, and cropland. In northeastern China, the results of all provincial administrative regions show acceptable performance, which is highly consistent with the survey data, because the landscapes of northeastern China are relatively similar (mainly composed of tree cover and cropland) and not easily confused. In northwestern and southwestern China, as the main distribute land-cover types, the misestimation of "grassland" and "barren and sparse vegetation" still exists in some provinces.

To demonstrate the spatial distribution of the misestimation rate for each land-cover type across China, we illustrated the misestimation maps for every land-cover type in Figure 22. From the results, the misestimation of some land-cover types shows a strong distribution pattern. For example, the misestimation of "Shrubland" is mainly distributed in the north and southwest of China. The misestimations of "Grassland" and "Barren and sparse vegetation" are concentrated in the north, northwest, and southwest of China. The misestimations of "Cropland" and "Building" are distributed on the coasts of eastern and southern China. The main misestimation land-cover types distributed in western China (i.e., Qinghai-Tibet Plateau and Xinjiang) are "Wetland" with a misestimation rate of 7.6%–9.5%, "Snow and ice" with a misestimation rate of 0.5%–1.8%, and "Moss and lichen" with a misestimation rate of 0.2%–0.3%. Besides, the SinoLC-1 of Hainan and Chongqing Provinces has a high overestimation of "Tree cover" and an underestimation of "Cropland". By considering the survey data, statistical comparison, and model training processing shown in Table 4, Figure 21, and Figure 3, Hainan and Chongqing Provinces have a high proportion of "Tree cover" in practice, and the labels generated for model training retain massive samples of "Tree cover" in these two areas, which led to the model overfitting and overestimating the types of "Tree cover".

To evaluate and analyze the overall misestimation area of every land-cover type, first, a box plot was used to describe the error distribution of every land-cover type in 31 provincial administrative regions. Figure 23 (a) shows the misestimation rate of most types remains low, which indicates SinoLC-1 is a statistically acceptable land-cover product across the nation.

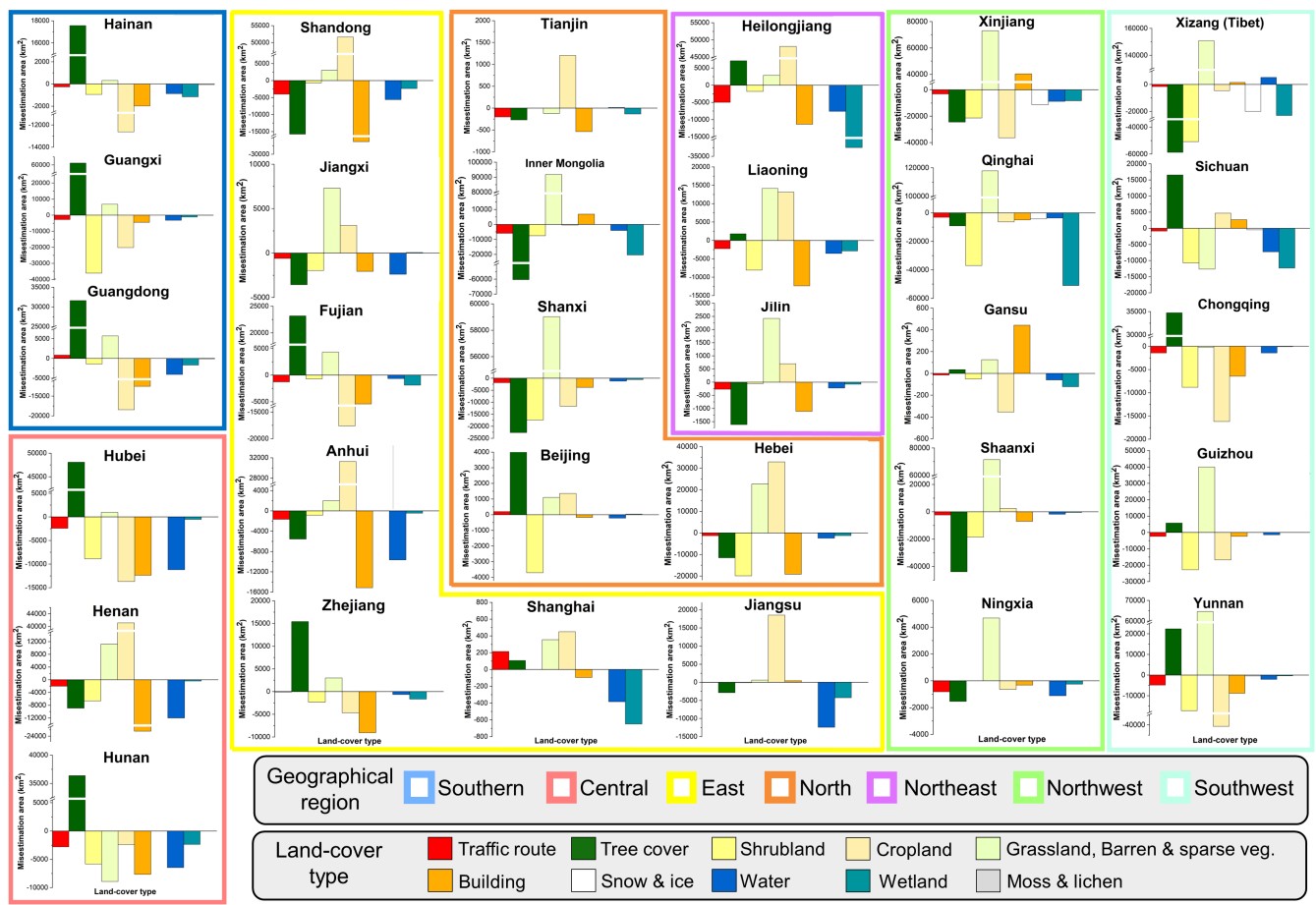

**Figure 21. Statistical comparison between SinoLC-1 and 3$^{rd}$ NLRS data for 31 provinces in China. The provinces in different geographical region are represented by dissimilar wireframe colors. In every subplot, the abscissa axis represents the land-cover types, and the vertical axis represents the misestimation area.**

Nevertheless, some outliers and large misestimation areas are observed in the type of "Grassland" and "Barren and sparse vegetation," and this misestimation is mainly in the northwest and southwest parts of China where such land-cover types occupy a very large proportion of these regions and are easily overestimated. Second, a multicolumn chart was used to demonstrate the misestimation rate in the seven geographical regions, which was calculated by using the misestimation area for each land-cover type to divide the total area of the region. Figure 23 (b) shows based on the various main landscapes of seven geographical regions, these regions exhibit different dominant misestimation land-cover types, and the misestimation rates of seven regions are all under 20% (most of them are under 15%). Third, we demonstrate a histogram of the national misestimation rate shown in Figure 23 (c) to visualize the statistical assessment of every land-cover type contained in SinoLC-1.

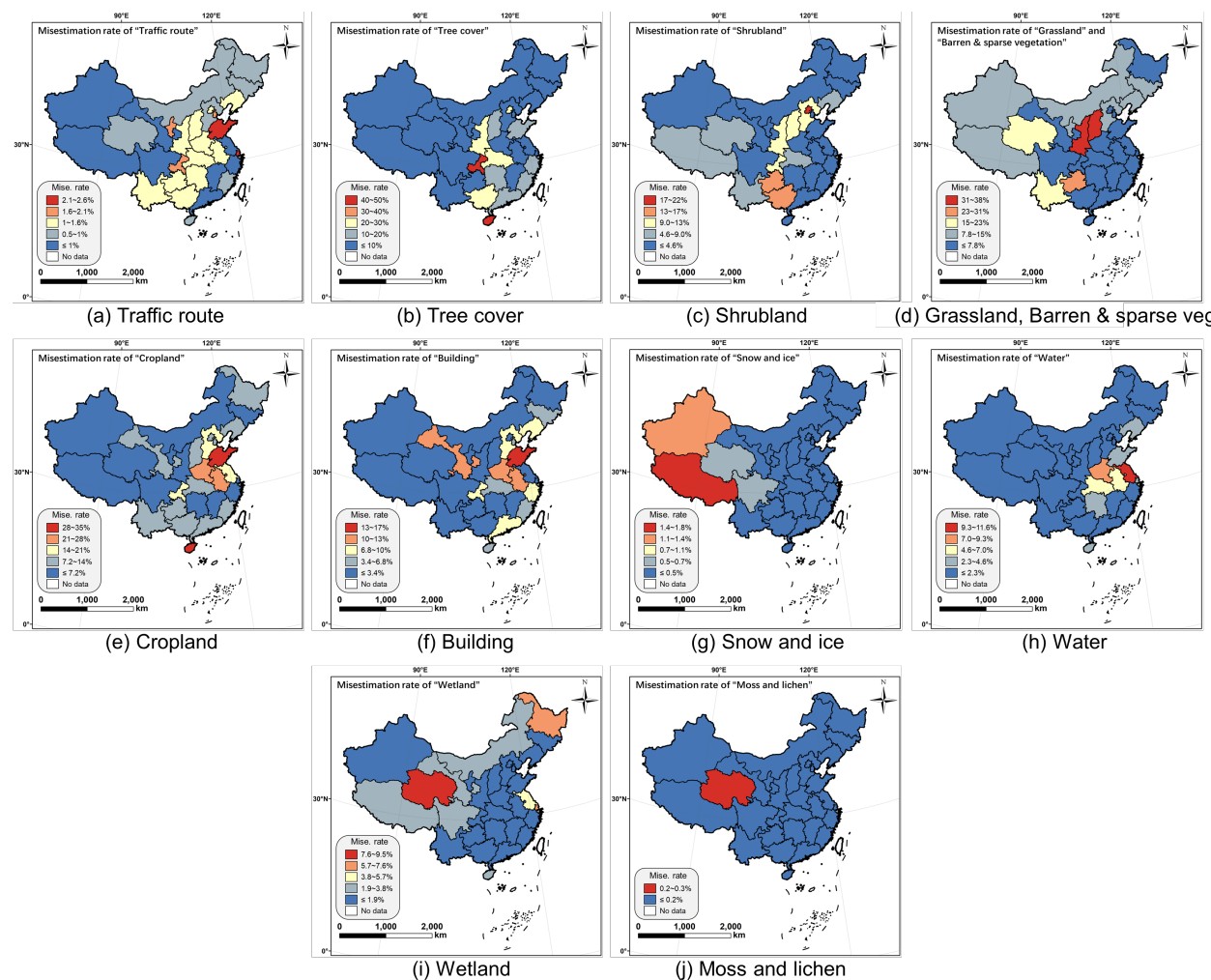

**Figure 22. The misestimation rate of SinoLC-1 for 31 provinces in China. In every subplot, the statistical comparison between SinoLC-1 and 3rd NLRS data in every land-cover type is illustrated.**

Moreover, to measure the overall statistical performance of SinoLC-1, we calculated the Frequency Weighted Misestimation Rate (FWMR) of SinoLC-1 to measure the overall proximity of SinoLC-1 to the official survey reports. Formally, FWMR is calculated by multiplying the misestimation rate of each land-cover type by their proportion shown in Figure 7 (b) and summing them up. The FWMR can be written as:

$$FWMR = \sum_{c=1}^{11} p_c m_c,$$

(4)

where $c$ represents the land-cover types counting from 1 to 11 (from "Traffic route" to "Moss and lichen"), $p_c$ represents the class proportion of $c$ land-cover type, and $m_c$ represents the misestimation rate of $c$ land-cover type. According to the

results shown in Figure 23 (c), the national misestimation rates of all land-cover types are under 11%, and the overall FWMR is 6.4%.

Overall, according to the official land resource survey data collected from the 3rd NLRS project, the reliability of the SinoLC-1 from the statistical aspect was further validated. The 3rd NLRS data were published by the provincial administrative governments, so the comparisons of every land-cover type in 31 provincial administrative regions first indicate the SinoLC-1 product is highly consistent with the official survey data in most of the provinces. Second, the overall performance of the SinoLC-1 at 31 provincial administrative regions and seven geographical regions was examined. The results indicate the misestimation rate of the SinoLC-1 is acceptable in general with an overall FWMR of 6.4%, and the main misestimation land-cover types are "Grassland" and "Barren and sparse vegetation" in northwest and southwest China.

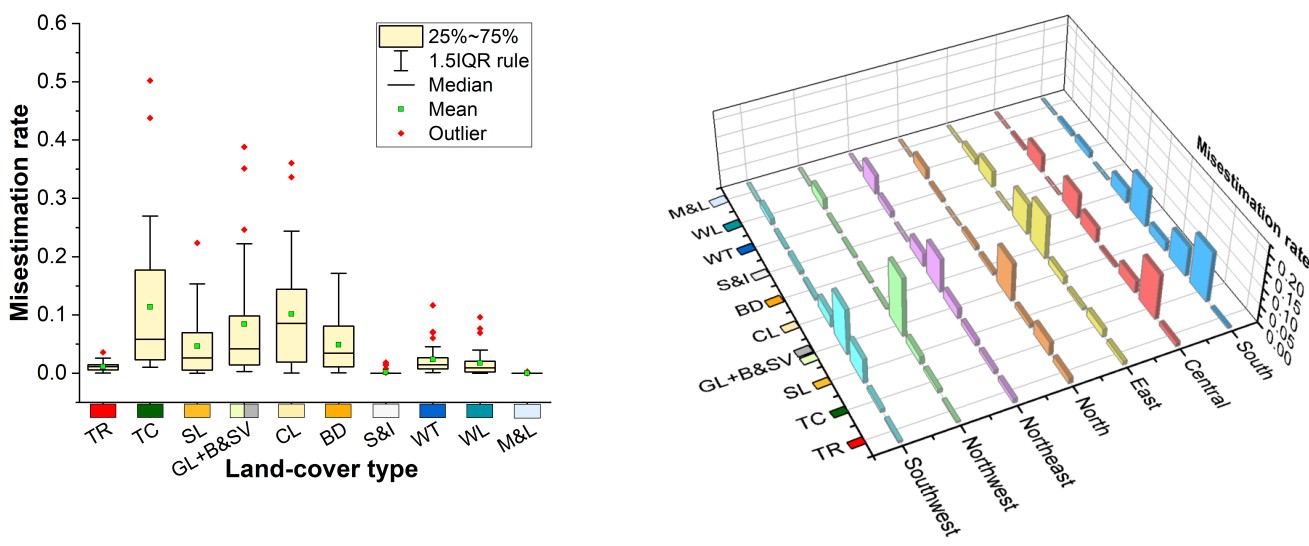

(a) Overall misestimation rate of every land-cover type through 31 provinces in China

(b) Overall misestimation rate of every land-cover type through seven geographical regions

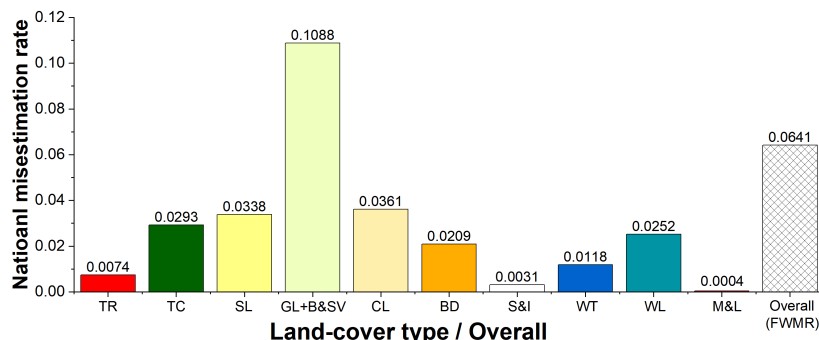

(c) National misestimation rate of every land-cover type across China

**Figure 23. Overall misestimation distributions in every land-cover type across China.**

## 4.4 Uncertainty and limitations of the SinoLC-1 land-cover product

SinoLC-1 enables VHR land-cover monitoring over China by using a deep learning-based mapping framework with multisource open-access data. During the production of SinoLC-1, no manual annotation to create VHR-labeled data was required, and no commercial VHR image source was used. The general process maintained low capital expenditure and low labor costs. However, as the trade-off situation between the spatial and temporal resolution of the remote-sensing images, one of the major limitations to the production of SinoLC-1 was the uneven temporal coverage of Google Earth images. The Google

Earth images were collected from different platforms at different time points to generate seamless images with large-scale coverage. Although Google Earth is a low-cost source to acquire nationwide coverage VHR images, the uneven temporal coverage of the images can affect the uniformity of the land-cover products.

Figure 24 shows the spatial distribution of the image capture time and the number of image tiles captured in different years. Most of the images were acquired around the year 2021, and the early captured images were mainly distributed in the

northern land frontier and the northwest part of China. According to the DEM data shown in Figure 9 and other published GLC products, the outdated images were generally in the west of China and are covered by plateau landforms (typically "Grassland" and "Barren and sparse vegetation" land-cover types). Furthermore, based on the 30-meter annual land-cover datasets provided by Yang & Huang (2021), as shown in Figure 25, we generated the annual land-cover change heatmap from 2011 to 2021 (the main time-distributions of the using VHR image) and the province-scale land-cover change map to

demonstrate the change rate in every provincial region. From Figure 25 (a), the annual change heatmaps show the land-cover change from 2011 to 2021 was relatively sparse. From  Figure 25 (b), the spatial distribution of the change areas shows that the most significant land-cover changes from 2011 to 2010 are located in the provinces of the south (e.g., Hainan, Guangdong, Guangxi, etc.), north (e.g., Inner Mongolia, Shanxi, Hebei, etc.), northeast (i.e., Jilin), and northwest (e.g., Xinjiang and Gansu). By combining the image capture time shown in Figure 24, the outdated VHR images are most probably to cause uncertainty

in the mapping results for the northern part of Inner Mongolia and Gansu (i.e., the northern border of China, with the change rate of 1%－3% from 2011 to 2021) and the southern part of Xinjiang (i.e., the Tarim Basin, with the change rate of 1%－3% from 2011 to 2021).

This distribution indicates the areas containing mass outdated images generally had less land-cover change over the years (e.g., Tibet and Qinghai provinces of Southwest China, with a change rate lower than 1%), which limited the uneven effect on

the produced results. Furthermore, during the production of SinoLC-1, the land-cover information mostly came from the three 10-meter GLC products where two of them (ESA_WorldCover v100 and ESRI land cover) represented a more recent (i.e., the year of 2020) land-cover information, and the VHR optical images mainly provided the fine edge and texture information of the land surface. Therefore, although the uneven temporal of the VHR images can still cause uncertainty in the SinoLC-1 land-cover products, owning to the training strategy that reasonably utilized the texture information of images and land-cover

information of the labels, the influence was minimized.

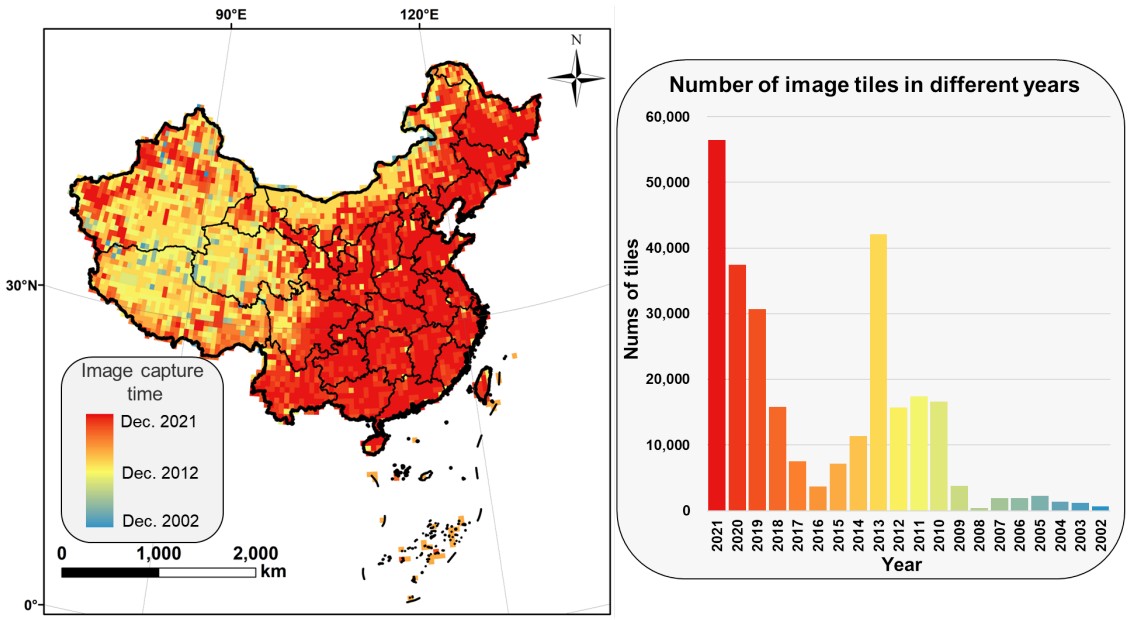

**Figure 24.** Demonstration of the image capture time and the number of image tiles in different years.

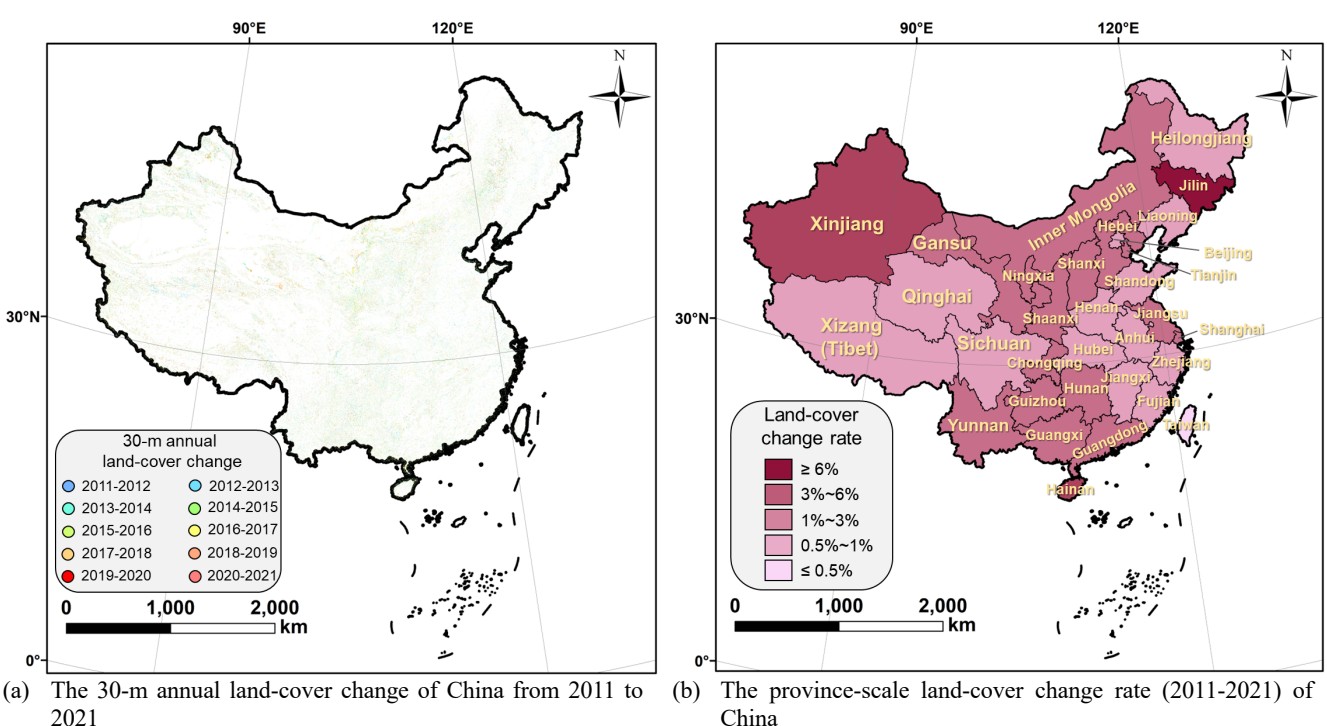

(a) The 30-m annual land-cover change of China from 2011 to 2021

(b) The province-scale land-cover change rate (2011-2021) of China

**Figure 25.** Spatial distribution of 30-m land-cover change in China from 2011 to 2021.

**Table 8. The province-scale land-cover change area/rate (2011-2021) of China**

| Geographical region | Provincial region | Provincial proportion to China's coverage (%) | Change area (km$^2$) | Change rate (%) |
|---|---|---|---|---|
| South | Hainan | 0.37 | 714.06 | 2.04 |
| | Guangxi | 2.50 | 3207.55 | 1.36 |
| | Guangdong | 1.89 | 2107.36 | 1.18 |
| East | Fujian | 1.31 | 779.53 | 0.64 |
| | Anhui | 1.48 | 820.93 | 0.59 |
| | Zhejiang | 1.11 | 719.86 | 0.69 |
| | Shanghai | 0.07 | 111.32 | 1.32 |
| | Jiangsu | 1.13 | 1697.93 | 1.60 |
| | Taiwan | 0.38 | 145.90 | 0.41 |
| | Jiangxi | 1.76 | 1488.89 | 0.89 |
| | Shandong | 1.64 | 1416.42 | 0.92 |
| Central | Hubei | 1.96 | 1852.50 | 1.00 |
| | Hunan | 2.23 | 2300.15 | 1.02 |
| | Henan | 1.75 | 1172.96 | 0.69 |
| North | Shanxi | 1.65 | 2631.97 | 1.73 |
| | Hebei | 1.99 | 2186.14 | 1.18 |
| | Beijing | 0.17 | 126.53 | 0.76 |
| | Inner Mongolia | 12.47 | 13144.22 | 1.33 |
| | Tianjin | 0.13 | 207.55 | 1.76 |
| Northeast | Liaoning | 1.56 | 878.47 | 0.59 |
| | Jilin | 0.29 | 1739.63 | 0.93 |
| | Heilongjiang | 4.98 | 2849.54 | 0.61 |
| Northwest | Shaanxi | 2.17 | 2631.97 | 1.29 |
| | Gansu | 4.49 | 6175.12 | 1.45 |
| | Xinjiang | 17.54 | 90325.45 | 5.43 |
| | Ningxia | 0.70 | 1173.43 | 1.77 |
| | Qinghai | 7.61 | 5695.08 | 0.79 |
| Southwest | Guizhou | 1.86 | 2702.60 | 1.67 |
| | Chongqing | 0.87 | 1045.01 | 1.32 |
| | Xizang (Tibet) | 12.68 | 8792.25 | 0.81 |
| | Yunnan | 4.15 | 4743.78 | 1.30 |
| | Sichuan | 5.12 | 3818.27 | 0.83 |

**5 Data availability**

The SinoLC-1 land-cover product generated in this paper and corresponding user guidelines are available at
https://doi.org/10.5281/zenodo.7707461 (Li et al., 2023). The product is grouped by city tiles in the GeoTIFF format, which
are packaged in provincial administrative region folders and stored as ".zip" files. Each city tile is named "G_P_C.tif," where
"G" explains the geographical region (south, central, east, north, northeast, northwest, and northeast of China) information,
"P" explains the provincial administrative region information, and "C" explains the city name. For example, the 1-meter land-
cover map for Wuhan City, Hubei Province is named "Central_Hubei_Wuhan.tif". Furthermore, each tile contains a land-
cover label band ranging from 0 to 255, where the corresponding relationship between the value and the land-cover types is
shown in Table 2 of Sect. 2.

**6 Conclusions**

A VHR (i.e., 1.07-meter resolution) national-scale land-cover product for China, called SinoLC-1, was produced by using
a low-cost deep learning-based L2H-Frame and multisource free access data derived from three 10-meter GLC products, OSM,
and Google Earth imagery. In the L2H-Frame, the reliable land-cover and traffic route labeled information was collected to
generate the training labels, and the VHR texture features were extracted from the 1-meter images by using the RP backbone.
The resolution mismatch between the VHR prediction results and the coarse training labels was resolved using the CAS module
and the L2H loss function with their weakly and self-supervised strategies.

The produced SinoLC1 dataset is the first 1-meter resolution and currently the highest resolution land-cover product that
covers all of China. Comprehensive comparisons with five other widely used products revealed the SinoLC-1 product with the
highest spatial resolution yielded the most accurate land-cover edges, indicating the finest landscape details. Moreover, with
an additional "Traffic route" land-cover type, the SinoLC-1 product portrayed the details of dense city and urban patterns more
precisely compared with other products. Quantitative assessments found the validation results derived from over 100,000
samples indicate SinoLC-1 achieved an O.A. of 73.61% and a kappa coefficient of 0.6595 across China. The validation results
of every geographical region indicated an acceptable accuracy distribution all around China. Furthermore, the statistical
validation results indicated SinoLC-1 conforms to the official survey reports with an overall misestimation rate of 6.4%
according to the government data. Overall, assessments and analysis in this paper suggested the SinoLC-1 land-cover product
accurately provided clear land-cover information and could become a vital support for downstream applications.

**Author contributions:**

Zhuohong Li and Hongyan Zhang designed the method. Zhuohong Li and Wei He programmed the framework codes.
Zhuohong Li, Mofan Cheng, Jingxin Hu, and Guangyi Yang collected and annotated the validation sets. Zhuohong Li wrote
the original draft. Hongyan Zhang and Wei He reviewed the draft.

**Competing interests:**

The authors declare that they have no conflict of interest.

**Disclaimer:**

Publisher's note: Copernicus Publications remains neutral with regard to jurisdictional claims in published maps and institutional affiliations.

**Acknowledgements:**

The authors gratefully acknowledge the free access of ESA_WorldCover v100 land-cover products provided by the
625 European Space Agency, the ESRI land-cover products provided by ESRI, Inc. and IO, Inc., the FROM_GLC products provided by Tsinghua University, the traffic route information provided by the OSM, and the VHR Google Earth images provided by the Google Inc.

**Financial support:**

This work was supported in part by the National Key Research and Development Program of China under Grant
2022YFB3903605, in part by the National Natural Science Foundation of China under Grant 42071322, and in part by the Natural Science Foundation of Hubei Province No. 2020CFA053.

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
