# Peer review of "SinoLC-1: the first 1-meter resolution national-scale land-cover map of China created with the deep learning framework and open-access data"

_Earth System Science Data, 2023_

## Author Comment (AC1)

**Referee #2**

We thank the reviewer for a thoughtful and thorough review of our manuscript (ESSD-2023-87: SinoLC-1: the first 1-meter resolution national-scale land-cover map of China created with the deep learning framework and open-access data). The suggestions and comments are listed in **bold** type. The modified words or materials are marked as blue color in the revised manuscript. The item-by-item responses to all comments are listed below.

**General comments:**

**The authors of this manuscript took such a tremendous effort to classify land cover of China in a very high (1m) resolution. However, the uncertainty of training datasets, the reproducibility of methods and the independence of validation were not clear.**

**Response:**

We appreciate your considerable comments and suggestions which help to clarify the scientific significance of SinoLC-1 land-cover dataset and expand its applicability. We have carefully considered all of the comments and suggestions listed below and tried our best to improve the manuscript focusing on clarifying the certainty of the training set, the reproducibility of the method, and the independence of validation.
* * *
**Suggestions and comments:**

**(1) This manuscript utilized 3 global-scale land cover products as training samples, but the mapping accuracy of them in China is uncertain especially considering that a small number of observations in China were included to generate these maps. Also, the uncertainty of the SinoLC-1 in the Southwest, Northwest and North regions due to unmatched training data and outdated VHR images need to be considered.**

**Response:**

We appreciate the reviewer for providing relevant and constructive comments and suggestions. To be clearer and in accordance with your concerns, we made major revisions and added materials as follows:

Firstly, to analyze the uncertainty of three global land-cover products, which were used to generate the SinoLC-1, more rigorously, we added two widely used open-access validation datasets to assess the accuracy of five global-scale products (including three utilized 10-m products and other two 30-m products) across China. According to your concerns in **Comments 2 and 9**, we have fully evaluated their user accuracy, overall accuracy, and kappa coefficient for each land-cover type in China, which are presented in Table R2-5, Figure R2-17, and Figure R2-18. We also analyzed their potential impact on the production process of SinoLC-1 comprehensively. Figure R2-1 shows the supplemented workflow added to comprehensively evaluate the accuracy and uncertainty of the SinoLC-1 and other land-cover products. The detailed material and descriptions are demonstrated in response to **Comments 2 and 9** (pages 2 and 19 of this response letter).

[Figure]

**Figure R2-1. The supplemented workflow to evaluate the accuracy and uncertainty of the SinoLC-1 and other five global land-cover products**

Secondly, to evaluate the uncertainty of the SinoLC-1 in the Southwest, Northwest, and North regions due to unmatched training data and outdated VHR images, we conducted a more complete accuracy validation based on the two open-access datasets in Section 4.3.2 (Statistical-level validation) Section 4.2.2 (Quantitative comparison with other land-cover products) of the revised manuscript and added a statistical-level error analysis of each land-cover type in Section 4.3.2 (Statistical-level validation). Furthermore, following your concerns in **Comment 11**, we have added a statistical table in Table 8 of the revised manuscript (shown in Table R1 of the response letter) to demonstrate the proportion and coverage of the change areas in each provincial region and added a province-scale change map in Figure 22 of the revised manuscript (shown in Figure R2-22 of the response letter) to illustrate the change rate (2011-2021) of China. Figure R2-2 shows the supplemented workflow to evaluate the error and uncertainty distribution of the SinoLC-1. The detailed material and descriptions are demonstrated in response to **Comments 9 and 11** (pages 19 and 24 of this response letter).

[Figure]

**Figure R2-2. The supplemented workflow to evaluate the error and uncertainty distribution of SinoLC-1**

**(2) Validation uncertainty. The authors manually annotated 106,852 points by visual interpretation results of VHR or HR imagery as validation datasets (Line 296-298). However, the accuracy of visual interpretation might contain considerable uncertainty. For example, ponds/lakes, paddy fields, and wetlands might be mis-interpretated. There are some open-accessed validation datasets (some obtained from field surveys), it would be great if the authors could add more rigorous and transparent validation.**

**Response:**

We are grateful to the reviewer for pointing out this problem. To address it, we first added the VHR samples captured from the 1.07-m Google Earth images for all land-cover types in Figure 6 of the manuscript (Figure R2-3 of the response letter). For each land-cover type, three VHR samples were added to help readers comprehend their characteristics. Secondly, we added two widely used open-access validation datasets (Liu et al., 2019; Zhao et al., 2014) to conduct more rigorous and transparent validation. These validation datasets were created on a basis of multiple data sources and manual verification, reporting a stable quality and high independence. The detailed information of these validation sets is as follows:

(1) Validation set created by Liu et al. DOI: https://doi.org/10.5281/zenodo.3551995.

Liu et al. (2019) created a global land-cover validation set by combining several existing reference datasets, such as the GLCNMO2008 training dataset, VIIRS reference dataset, STEP reference dataset and Global cropland reference data, to guarantee the confidence and objective of the validation samples. Furthermore, high-resolution imagery in Google earth and time-series NDVI, NDSI values of each related point were integrated to obtain the validation datasets.

(2) Validation set created by Zhao et al. DOI: https://doi.org/10.1080/01431161.2014.930202.

Zhao et al. (2014) created a global land-cover validation set with a total of 38,664 sample units by interpreting Landsat images and MODIS EVI time series data, as well as high-resolution images from Google Earth, recording the quality of reference data, and interpreter confidence. Zhao et al. confirmed that the dataset had been carefully improved through several rounds of interpretation and verification by different image interpreters, and checked by one quality controller. Independent test interpretation indicated that the quality control correctness level reached 90% at level 1 land-cove type.

According to the description of the data providers, these validation sets contain two levels of land-cover types, and their spatial distribution and classification system are shown in Figure R2-4, Table R2-1, and Table R2-2.

[Figure]

**Figure R2-3. Demonstration of the sample grid, VHR samples, and the national validation sample set. Left: the spatial distributions of the sample set (the legend is written in shorter forms). Right: the VHR samples of different land-cover types collected from 1.07-m resolution © Google Earth imagery all around China.**

[Figure]

(a)  Validation set created by Liu et al.

(b) Validation set created by Zhao et al.

**Figure R2-4. Demonstration of two open-access validation set.**

**Table R2-1. The classification system of the validation set created by Liu et al.**

| Level 1 type | Level 2 type | Sample count | Total | Proportion (%) |
|---|---|---|---|---|
| Cropland | Rainfed cropland | 44 | 353 | 14.33% |
| | Herbaceous cover | 0 | | |
| | Irrigated cropland | 311 | | |
| Forest | Evergreen broadleaved forest | 123 | 542 | 22.01% |
| | Deciduous broadleaved forest | 303 | | |
| | Mixed leaf forest | 116 | | |
| Shrubland | Shrubland | 78 | 104 | 4.22% |
| | Evergreen shrubland | 26 | | |
| Grassland | Grassland | 360 | 360 | 14.62% |
| Wetlands | Wetlands | 17 | 17 | 0.69% |
| Impervious surfaces | Impervious surfaces | 71 | 71 | 2.88% |
| Bare areas | Sparse vegetation | 285 | 641 | 26.03% |
| | Bare areas | 329 | | |
| | Consolidated bare areas | 3 | | |
| | Unconsolidated bare areas | 24 | | |
| Water body | Water body | 37 | 37 | 1.50% |
| Permanent ice and snow | Permanent ice and snow | 338 | 338 | 13.72% |

**Table R2-2. The classification system of the validation set created by Zhao et al.**

| Level 1 type | Level 2 type | Sample count | Total | Proportion (%) |
|---|---|---|---|---|
| Crop | Rice | 3 | 353 | 16.98% |
| | Greenhouse | 1 | | |
| | Other | 349 | | |
| Forest | Broadleaf | 303 | 512 | 24.63% |
| | Needleleaf | 81 | | |
| | Mixed | 114 | | |
| | Orchard | 14 | | |
| Grass | Managed | 0 | 312 | 15.01% |
| | Nature | 312 | | |
| Shrub | Shrub | 103 | 103 | 4.95% |
| Wetland | Grass | 15 | 15 | 0.72% |
| | Silt | 0 | | |
| Water | Lake | 7 | 33 | 1.59% |
| | Pond | 19 | | |
| | River | 7 | | |
| | Sea | 0 | | |
| Impervious | High albedo | 19 | 52 | 2.50% |
| | Low albedo | 33 | | |
| Bare land | Saline-Alkali | 10 | 619 | 29.77% |
| | Sand | 138 | | |
| | Gravel | 303 | | |
| | Bare-cropland | 89 | | |
| | Dry river/lake bed | 2 | | |
| | other | 77 | | |
| Snow and Ice | Snow | 80 | 80 | 3.85% |
| | Ice | 0 | | |

Based on two open-access validation sets, we calculated the confusion matrix of SinoLC-1 and further validated its producer accuracy (P.A.), user accuracy (U.A.), overall accuracy (O.A.), and kappa coefficient. As shown in Table R2-3 and Table R2-6, the O.A. of the SinoLC-1 validated on the validation sets created by Liu et al. and Zhao et al. are 78.80% and 64.69%, respectively. The Kappa of the SinoLC-1 validated on the validation sets created by Liu et al. and Zhao et al. are 0.7394 and 0.5588, respectively.

Furthermore, to illustrate more detailed assessment results, Figure R2-5 shows the corresponding confusion proportions for each considered land-cover type of the SinoLC-1 validated on two sets. In addition, to assess the SinoLC-1 more rigorously and transparently, we used these validation sets to validate the accuracy of five comparative land-cover datasets, and the quantitative results are shown in Table R5. With the validation set created by Liu et al, all products have a higher O.A. and the SinoLC-1 ranks second with an O.A. of 78.81%. With the validation set created by Zhao et al, all products have an O.A. of around 60%, and the SinoLC-1 ranks second with an O.A. of 64.69%.

According to your consideration in **Comment 9** (recommending us to add numerical statistics results to compare the performance of different land-cover products in China), we made a more detailed comparison and analysis in response to **Comment 9** to compare the SinoLC-1 and the other five products more comprehensively.

**Table R2-3. Confusion matrix for the SinoLC-1 according to the validation set created by Liu et al.**

| Classification | TC | SL | GL | CL | IP | BL&SV | S&I | WT | WL | Total | P.A. (%) |
|---|---|---|---|---|---|---|---|---|---|---|---|
| Tree Cover | 421 | 5 | 80 | 32 | 0 | 2 | 1 | 1 | 0 | 542 | 77.68 |
| Shrubland | 7 | 34 | 32 | 2 | 0 | 27 | 2 | 0 | 0 | 104 | 32.69 |
| Grassland | 2 | 2 | 342 | 0 | 0 | 7 | 0 | 0 | 0 | 353 | 96.88 |
| Cropland | 5 | 1 | 3 | 316 | 29 | 3 | 0 | 3 | 0 | 360 | 87.78 |
| Impervious | 7 | 0 | 3 | 7 | 51 | 3 | 0 | 0 | 0 | 71 | 71.83 |
| Barren &Sparse veg. | 1 | 7 | 12 | 0 | 0 | 616 | 5 | 0 | 0 | 641 | 96.10 |
| Snow and ice | 1 | 0 | 78 | 0 | 0 | 121 | 136 | 2 | 0 | 338 | 40.24 |
| Water | 7 | 0 | 0 | 5 | 3 | 1 | 0 | 19 | 2 | 37 | 51.35 |
| Wetland | 0 | 0 | 1 | 4 | 0 | 1 | 2 | 3 | 6 | 17 | 35.29 |
| Total | 451 | 49 | 551 | 366 | 83 | 781 | 146 | 28 | 8 | 2463 | |
| U.A. (%) | 93.35 | 69.39 | 62.07 | 86.34 | 61.45 | 78.87 | 93.15 | 67.86 | 75.00 | | |
| O.A. (%) | | | | | 78.80 | | | | | | |
| Kappa | | | | | 0.7394 | | | | | | |

Note: TC=Tree cover; SL=Shrubland; GL=Grassland; CL=Cropland; IP=Impervious (Building and traffic route); BL&SV=Barren and sparse vegetation; S&I=Snow and ice; WT=Water; WL=Wetland

**Table R2-4. Confusion matrix for the SinoLC-1 according to the validation set created by Zhao et al.**

| Classification | TC | SL | GL | CL | IP | BL&SV | S&I | WT | WL | Total | P.A. (%) |
|---|---|---|---|---|---|---|---|---|---|---|---|
| Tree Cover | 376 | 13 | 60 | 35 | 6 | 7 | 1 | 5 | 9 | 512 | 73.44 |
| Shrubland | 10 | 34 | 33 | 3 | 0 | 21 | 2 | 0 | 0 | 103 | 33.01 |
| Grassland | 37 | 2 | 215 | 20 | 3 | 33 | 0 | 1 | 1 | 312 | 68.91 |
| Cropland | 83 | 1 | 36 | 191 | 18 | 21 | 0 | 2 | 1 | 353 | 54.11 |
| Impervious | 3 | 0 | 4 | 13 | 29 | 3 | 0 | 0 | 0 | 52 | 55.77 |
| Barren &Sparse veg. | 13 | 7 | 109 | 62 | 6 | 411 | 5 | 5 | 1 | 619 | 66.40 |
| Snow and ice | 1 | 0 | 5 | 0 | 0 | 8 | 65 | 1 | 0 | 80 | 81.25 |
| Water | 6 | 0 | 0 | 7 | 2 | 2 | 0 | 16 | 0 | 33 | 48.48 |
| Wetland | 0 | 0 | 2 | 0 | 1 | 2 | 2 | 0 | 8 | 15 | 53.33 |
| Total | 529 | 57 | 464 | 331 | 65 | 508 | 75 | 30 | 20 | 2079 | |
| U.A. (%) | 71.08 | 59.65 | 46.34 | 57.70 | 44.62 | 80.91 | 86.67 | 53.33 | 40.00 | | |
| O.A. (%) | | | | | | 64.69 | | | | | |
| Kappa | | | | | | 0.5588 | | | | | |

Note: TC=Tree cover; SL=Shrubland; GL=Grassland; CL=Cropland; IP=Impervious (Building and traffic route); BL&SV=Barren and sparse vegetation; S&I=Snow and ice; WT=Water; WL=Wetland

(a) Confusion proportions for land-cover type of the SinoLC-1 validated with the set created by Liu et al.

(b) Confusion proportions for land-cover type of the SinoLC-1 validated with the set created by Zhao et al.

**Figure R2-5. Confusion proportions of the validation results.**

**Table R2-5. Quantitative comparison between the SionLC-1 and other five land-cover products.**

| Metric / Dataset | Validation set of Zhao et al. | | Validation set of Liu et al. | |
|---|---|---|---|---|
| | O. A. | Kappa | O. A. | Kappa |
| SinoLC-1 | 0.6469 | 0.5588 | 0.7881 | 0.7394 |
| ESA_GLC10 | 0.6646 | 0.5722 | 0.7356 | 0.6269 |
| FROM_GLC10 | 0.6411 | 0.5942 | 0.7538 | 0.6871 |
| ESRI_GLC10 | 0.6232 | 0.5210 | 0.6675 | 0.5972 |
| GlobaLand30 | 0.6209 | 0.5285 | 0.7694 | 0.7090 |
| GLC_FCS30 | 0.5778 | 0.4675 | 0.8684 | 0.8241 |

The cited references of this response are as follows:

Zhao, Y., Gong, P., Yu, L., Hu, L., Li, X., Li, C., Zhang, H., Zheng, Y., Wang, J., Zhao, Y. and Cheng, Q. (2014). Towards a common validation sample set for global land-cover mapping. International Journal of Remote Sensing, 35(13), 4795-4814. https://doi.org/10.1080/01431161.2014.930202

Liu, L., Gao, Y., Zhang, X., Chen, X., & Xie, S. (2019). A Dataset of Global Land Cover Validation Samples (Version v1) [Data set]. Zenodo. https://doi.org/10.5281/zenodo.3551995

**(3) Line 25: "SinoLC-1 conformed closely to the official survey reports", this expression is vague, needs statistical values to support how close.**

**Response:**

Thank you for the suggestion. To be clearer and in accordance with your concerns, we have added a histogram of the national misestimation rate, as shown in Figure 23 (c) of the revised manuscript (Figure R2-6 (a) of the response letter), to visualize the statistical assessment of every land-cover type containing in SinoLC-1. Furthermore, we calculated the Frequency Weighted Misestimation Rate (FWMR) of SinoLC-1 to measure the overall proximity of SinoLC-1 to the official survey reports. Referring to the calculation of Frequency Weighted Intersection over Union (FWIoU) (Long et al., 2015), FWMR is calculated by multiplying the misestimation rate of each land-cover type by their proportions shown in Figure R2-6 (b) and summing them up. Formally, the FWMR can be written as:

$$FWMR = \sum_{c=1}^{11} p_c m_c,$$

where $c$ represents the land-cover types counting from 1 to 11 (from 'traffic route' to 'Moss and

lichen'), $p_c$ represents the class proportion of $c$ land-cover type, and $m_c$ represents the misestimation rate of $c$ land-cover type.

According to the results shown in Figure R2-6 (a), the national misestimation rates of all land-cover types are under 11%, and the overall FWMR is 6.4%. Based on the analysis, we have revised the expression describing the overall proximity of SinoLC-1 to the official survey reports in the Abstract, Section 4.3.2 (Statistical-level validation), and Section 6 (Conclusion) of the manuscript.

[Figure]

[Figure]

(a) National misestimation rate of every land-cover type across China     (b) Class proportion of the SinoLC-1 dataset.

**Figure R2-6. National misestimation rate and class proportion of the SinoLC-1 dataset.**

The cited reference of this response is as follow:

Long, J., Shelhamer, E., & Darrell, T. (2015). Fully convolutional networks for semantic segmentation. In Proceedings of the IEEE conference on computer vision and pattern recognition, 3431-3440.

**(4) Line 275-276: "the predicted batches were seamlessly merged into the land-cover tiles by taking the average predicted values of the overlapped areas", since the land cover is categorical data, it would be more reasonable to take the majority instead of the average.**

**Response:**

Thanks for your constructive feedback. For common majority-voting process, three or more prediction results are required. For the overlapping part of two prediction results, we calculated the average of probability matrix for the overlapping areas, and then for every pixel located in the overlapping areas, we take the class with maximum predicted probabilities among all land-cover classes as the final prediction results. According to your comment, we would like to explain the seamless mapping and merging process more clearly. In this response letter, we supplemented

Figure R2-7 to illustrate the processing process of overlapped areas and Figure R2-8 to show a simple example to explain how the final results are obtained via two overlapped batches.

For each image tile ( $6000\times6000$ pixels) shown in Figure R2-7, adjacent image batches (256 $\times256$ pixels) with 128 pixels overlapped areas are taken as the input of a well-trained model to obtain two prediction matrices $\mathbf{M}_1$ and $\mathbf{M}_2$ , where each matrix has a prediction probability with the sizes of $11\times256\times256$ (Class $\times$ Height $\times$ Width). Subsequently, the average value of the overlapped parts on each class (e.g., tree, building, water, etc.) is calculated to obtain the average matrix $\mathbf{M}_{avg}$. Finally, as shown in Figure R2-8, the maximum value of each pixel in $\mathbf{M}_{avg}$ is taken among each class channel to obtain the final land-cover mapping results. Based on this process, the problem of edge mismatch between adjacent prediction results is alleviated to a certain extent, assisting us to obtain seamless and continuous land-cover maps.

In order to provide a clearer explanation of this process in the revised manuscript, we have supplemented the expression in Section 3.2.2 (Seamless mapping and merging) and modified Figure 5 of the manuscripts (shown in Figure R2-9 of the response letter).

[Figure]

**Figure R2-7. Demonstration of the processing process of overlapped areas**

[Figure]

**Figure R2-8. Demonstration of a simple example to explain how the final results are obtain via two overlapped batches.**

[Figure]

**Figure R2-9.  Demonstration of the mapping and merging for producing SinoLC-1. The VHR remote sensing images in the figure are from © Google Earth 2021**

**(5) Figure 7: the bar showed the sample number instead of the proportion. It would be better to show the proportion of the validation samples of each type account for all sample points (106, 852) and the area proportion of each land-cover type of China in the SinoLC-1 dataset.**

**Response:**

Thank you for the constructive comments which can improve the quality and reasonability of the manuscript. According to your comments, we modified the histogram shown in Figure 7 of the previous manuscript (Figure R2-10 (a) of the response letter) into the pie chart which can better demonstrate the proportion of each land-cover type. Furthermore, as shown in Figure R2-10 (b), we supplemented the pie chart of the land-cover proportion in the SinoLC-1 dataset. Based on the modified Figure 7 of the revised manuscript, the land-cover proportion of selected sample points in the validation set is relatively similar to the SinoLC-1 dataset, further indicating that the ~100,000 sample points have reasonable class distribution.

[Figure]

[Figure]

(a) Class proportion of the national validation sample set.   (b) Class proportion of the SinoLC-1 land-cover dataset.

**Figure R2-10. Land-cover proportion of the national validation sample set and the produced SinoLC-1 land-cover dataset.**

**(6) Figure 8: the legend is missing.**

**Response:**

Thank you for your constructive feedback. We have supplemented the legends to Figures 8 of the revised manuscript (Figure R2-11 of the response letter). Furthermore, to improve the visualization of the qualitative comparison between the SionLC-1 and other land-cover datasets, we also supplemented the legends to all maps shown in Figures 13 and 14 of the manuscript (Figure R2-12 and Figure R2-13 of this respond letter).

[Figure]

**Figure R2-11. Demonstration of the SinoLC-1: a 1-meter-resolution national-scale land-cover map of China.**

[Figure]

**Figure R2-12. Demonstration of the visual comparison for Changzhou City, Jiangsu Province. The VHR remote sensing image in the figure is from © Google Earth 2021.**

[Figure]

Figure R2-13. Demonstrations of the visual comparison for four typical regions. The VHR remote sensing images in the figure are from © Google Earth 2021.

**(7) Line 409-412: The expression is not clear, please clarify which types showed higher accuracies (O.A. and kappa), and which types showed low accuracies.**

**Response:**

Thank you for the comment. We have clarified the exact land-cover types that showed higher and lower accuracies in Section 4.3.1 (Pixel-level sample validation). To describe the analysis results in a more understandable way, the descriptions of the revised manuscript have been revised to '

*By combining the class proportion of the validation sample set shown in Figure 7and the confusion matrix shown in Table 6 and Figure 19, the quantitative results of the basic land-cover types (i.e., the types of tree canopy, grassland, cropland, barren & sparse vegetation, and water), which have easily distinguishable features and occupy a large area in China, report higher accuracies and have a small proportion of misclassification. By contrast, the land-cover types (i.e., the types of traffic route, moss & lichen, and snow & ice), which occupy a small area, obtain relatively low accuracies and have a large proportion of misclassification.*'

**(8) Figure 15: Adding the numerical values of confusion proportions to this figure would provide more quantitative information.**

   **Response:**

   Thank you for the constructive feedback for improving the quantitative information of the figure. We have added the numerical values in Figure 15 of the previous manuscript (Figure R2-14 of this response letter).

[Figure]

**Figure R2-14. Confusion proportions for each land-cover type in the SinoLC-1 validation scheme.**

**(9) 3.2 section belongs to Results, but almost no numerical statistics were shown to support the descriptions.**

**Response:**

We are grateful to the reviewer for pointing out this problem. In the previous manuscript, Section 4.2 (Qualitative comparison with other land-cover products) focused on the qualitative and visual comparison based on one large-scale demonstration area (shown in Figure 13 of the manuscript) and four region-scale areas (shown in Figure 14 of the manuscript). To conduct a more rigorous comparison and quantitative analysis, we added two widely used open-accessed validation datasets (Liu et al., 2019; Zhao et al., 2014) to conduct validation and comparison of the SinoLC-1 and other five products across China. Moreover, we added a subsection of 'Quantitative comparison with other land-cover products' in Section 4.2.2 to make the comparison more scientific and transparent. Detailed information of these two open-access validation sets has been introduced in **Comment 2**. For clearer expression, we mark the validation set created by Liu et al. (2019) as S1 and mark the set created by Zhao et al. (2017) as S2. Figure R2-15 and Figure R2-16 show the spatial distribution of two validation sets among five comparative products in China.

[Figure]

**Figure R2-15. Demonstration of five comparison products and the validation set (S1) created by Liu et al.**

[Figure]

**Figure R2-16. Demonstration of five comparison products and the validation set (S2) created by Zhao et al.**

Based on the two validation sets, we compared the O.A. and Kappa between the SinoLC-1 and the other five products. The comparison results are shown in Table R2-5 and Figure R2-17. From the quantitative comparison, the SinoLC-1 has the second highest O.A. on two validation sets where the SinoLC-1 has a O.A. of 0.6469 with S1 (lower than the 10-meter ESA_GLC10) and has an O.A. of 0.7881 with S2 (lower than the 30-meter GLC_FCS30). Furthermore, we compared the U.A. of every considered type between the SionLC-1 and the other five products in Figure R2-18. From the results shown in Figure R2-18 (a), the SinoLC-1 has the second highest U.A. in types of 'Tree canopy', 'Shrubland', 'Grassland', and 'Wetland' compared to the other five products, and has the U.A. of 'Cropland' and 'Impervious surface' surpassing the average of other five products. From the results shown in Figure R2-18 (b), the SinoLC-1 has the highest U.A. in types of 'Shrubland' and 'Grassland', and has the U.A. of 'Snow and ice' and 'Wetland' surpassing the average of the other five products.

In general, by quantitatively comparing the SinoLC-1 product with five widely used land-cover products on two open-access validation datasets, the produced SinoLC-1 shows acceptable confusion proportion among land-cover types and has competitive accuracy among the other land-cover products across China.

[Figure]

(a)    The validation results based on S1

(b)    The validation results based on S2

**Figure R2-17. The quantitative validation and comparison of the SinoLC-1 and other five products**

[Figure]

(a)    The U.A. comparison based on S1

(b)    The U.A. compassion based on S2

**Figure R2-18. The U.A. comparison of the SinoLC-1 and other five products.**

**(10) Figure 18, the left figure (a) showed the misestimated area, while it would be more comparable if it showed the misestimated rate for each land-cover type.**

**Response:**

We are grateful for the suggestion. We agree that the misestimated rate can include more comparable information than the misestimated area between different land-cover types. To be clearer and in accordance with your concerns, we illustrated the misestimated rate of every land-cover type through 31 provincial regions in Figure R2-19 to better visualize the distribution of original results. In the revised manuscript, we have revised Figure 23 (shown in Figure R2-20 of the response letter) by changing the vertical axis of subfigure (a) from 'misestimation area (km$^2$)' to misestimation rate. Moreover, to visualize the total results of the statistical assessment in China, we have added a histogram of the national misestimation rate shown in Figure 23 (c) of the revised manuscript (Figure R2-20 (c) of the response letter).

In addition, to demonstrate the spatial distribution of the misestimation rate for each land-cover type across China, and to provide more comparable information on the statistical assessment, we have collected the results and added the map of the misestimation rate for every land-cover type in Figure 22 of the revised manuscript (shown in Figure R2-21 of the response letter). From the maps of the misestimation rate, misestimations of some land-cover types show a strong distribution pattern. For example, the misestimation of 'Shrubland' is mainly distributed in the north and southwest of China. The misestimations of 'Grassland' and 'Barren and sparse vegetation' are concentrated in the north, northwest, and southwest of China. The misestimations of 'Cropland' and 'Building' are distributed on the coasts of eastern and southern China. The main misestimation land-cover types distributed in western China (i.e., Qinghai-Tibet Plateau and Xinjiang) are 'Wetland' with a misestimation rate of 7.6%－9.5%, 'Snow and ice' with a misestimation rate of 0.5%－1.8%, and 'Moss and lichen' with a misestimation rate of 0.2%－0.3%.

[Figure]

**Figure R2-19. Misestimation rate of every land-cover type through 31 provinces in China.**

[Figure]

(a)  Overall misestimation rate of every land-cover type through 31 provinces in China

(b)  Overall misestimation rate of every land-cover type through seven geographical regions

(c)  National misestimation rate of every land-cover type across China

**Figure R2-20. Overall misestimation distributions in every land-cover type across China.**

[Figure]

**Figure R2-21. The misestimation rate of SinoLC-1 for 31 provinces in China. In every subplot, the statistical comparison between SinoLC-1 and 3rd NLRS data in every land-cover type is illustrated.**

(11) **Line 480-485: Figure 20 shows significant land-cover changes between 2011 and 2021. It would be better to add a statistical table of the proportion of change areas in each region, which would be helpful to assess the uncertainty in the Southwest, Northwest and North region.**

**Response:**

Thank you for the suggestion that can help visualize the change areas between 2011 to 2021 more clearly and further assist the analysis of uncertainty in the Southwest, Northwest, and North regions. In accordance with your concerns, we have added a statistical table in Table 8 of the revised manuscript (shown in Table R2-6 of the response letter) to demonstrate the proportion and coverage of the change areas in each provincial region.

**Table R2-6. The province-scale land-cover change area/rate (2011-2021) of China**

| Geographical region | Provincial region | Provincial proportion to China's coverage (%) | Change area (km2) | Change rate (%) |
|---|---|---|---|---|
| South | Hainan | 0.37 | 714.06 | 2.04 |
| | Guangxi | 2.50 | 3207.55 | 1.36 |
| | Guangdong | 1.89 | 2107.36 | 1.18 |
| East | Fujian | 1.31 | 779.53 | 0.64 |
| | Anhui | 1.48 | 820.93 | 0.59 |
| | Zhejiang | 1.11 | 719.86 | 0.69 |
| | Shanghai | 0.07 | 111.32 | 1.32 |
| | Jiangsu | 1.13 | 1697.93 | 1.60 |
| | Taiwan | 0.38 | 145.90 | 0.41 |
| | Jiangxi | 1.76 | 1488.89 | 0.89 |
| | Shandong | 1.64 | 1416.42 | 0.92 |
| Central | Hubei | 1.96 | 1852.50 | 1.00 |
| | Hunan | 2.23 | 2300.15 | 1.02 |
| | Henan | 1.75 | 1172.96 | 0.69 |
| North | Shanxi | 1.65 | 2631.97 | 1.73 |
| | Hebei | 1.99 | 2186.14 | 1.18 |
| | Beijing | 0.17 | 126.53 | 0.76 |
| | Inner Mongolia | 12.47 | 13144.22 | 1.33 |
| | Tianjin | 0.13 | 207.55 | 1.76 |
| Northeast | Liaoning | 1.56 | 878.47 | 0.59 |
| | Jilin | 0.29 | 1739.63 | 0.93 |
| | Heilongjiang | 4.98 | 2849.54 | 0.61 |
| Northwest | Shaanxi | 2.17 | 2631.97 | 1.29 |
| | Gansu | 4.49 | 6175.12 | 1.45 |
| | Xinjiang | 17.54 | 90325.45 | 5.43 |
| | Ningxia | 0.70 | 1173.43 | 1.77 |
| | Qinghai | 7.61 | 5695.08 | 0.79 |
| Southwest | Guizhou | 1.86 | 2702.60 | 1.67 |
| | Chongqing | 0.87 | 1045.01 | 1.32 |
| | Xizang (Tibet) | 12.68 | 8792.25 | 0.81 |
| | Yunnan | 4.15 | 4743.78 | 1.30 |
| | Sichuan | 5.12 | 3818.27 | 0.83 |

Furthermore, we added a province-scale change map in Figure 22 of the revised manuscript (shown in Figure R2-22 of the response letter) to illustrate the change rate (2011-2021) in China. In Figure R2-22 (b), the spatial distribution of the change areas shows that the most significant land-cover changes from 2011 to 2010 are located in the provinces of the south (e.g., Hainan, Guangdong, Guangxi, etc.), north (e.g., Inner Mongolia, Shanxi, Hebei, etc.), northeast (i.e., Jilin), and northwest (e.g., Xinjiang and Gansu). By combining the distribution of outdated images shown in Figure R2-23 and the significant change area shown in Figure R2-22 (b), the outdated VHR images are most probably to cause uncertainty in the mapping results for the northern part of Inner Mongolia and Gansu (i.e., the northern border of China, with the change rate of 1%–3% from 2011 to 2021) and the southern part of Xinjiang (i.e., the Tarim Basin, with the change rate of 1%–3% from 2011 to 2021).

This distribution indicates the areas containing mass outdated images generally had less land-cover change over the years (e.g., Tibet and Qinghai provinces of Southwest China, with a change rate lower than 1%), which limited the uneven effect on the produced results.

[Figure]

(a) The 30-m annual land-cover change of China from 2011 to 2021

(b) The province-scale land-cover change rate (2011-2021) of China

**Figure R2-22. Spatial distribution of 30-m land-cover change in China from 2011 to 2021.**

[Figure]

**Figure R2-23. Demonstration of the image capture time and the number of image tiles in different years**

---

## Author Comment (AC2)

**Referee #1**

We thank the reviewer for a thoughtful and thorough review of our manuscript (ESSD-2023-87: SinoLC-1: the first 1-meter resolution national-scale land-cover map of China created with the deep learning framework and open-access data). The suggestions and comments are listed in **bold** type. The modified words or materials are marked as blue color in the revised manuscript. The item-by-item responses to all comments are listed below.

**General comments:**

**The SinoLC-1 product, the initial 1-m resolution land cover data product for China, is introduced in this work. It may be useful for understanding fine-scale biogeophysical issues on the land. Also, the product offers development in big data processing, sample migration, and open-access data application that might be useful for efficient national land resource surveys and the mapping of large-scale very-high-resolution land cover data. Before it may be accepted, this manuscript should yet be improved.**

**Response:**

We are particularly grateful for your careful reading, and for giving us many constructive comments on this work. According to the suggestions and comments, we have carefully considered all of them and tried our best to improve the manuscript.
* * *
**Suggestions and comments:**

**(1) The Introduction, which focuses on data at the global scale, highlights 3 types of land use land cover data that fully or partially cover China. Reviewing national and local-scale land use land cover data in China is advised given that this manuscript focuses on the production of land cover maps at the national scale. The CLCD and CLUD in China and the NLCD in the United States are examples of the several extensively utilized national-scale land use land cover product.**

**Response:**

We appreciate this suggestion from the reviewer and agree that reviewing the various achievements of researchers in national-scale land-cover products enables us to improve the logicality of the introduction part and make the manuscript more complete.

Based on the suggestions, we supplemented the types of 'National-scale moderate-/high-resolution land-cover products' in the Introduction Section and carefully collected the materials of five high-quality national-scale land-cover products. Furthermore, to clearly demonstrate various types of land-cover datasets reviewed in the manuscript, we summarized their information and reference sources in Table R1-1.

The supplement materials include:

- The 30-m resolution National Land Cover Database (NLCD) covering the whole United States, which is cyclically updated by the United States Geological Survey (USGS) with the Landsat imagery (Wickham et al., 2021);

- The 10-m resolution LCM2020 covering the whole United Kingdom, which is periodically published by the United Kingdom Centre for Ecology & Hydrology (UKCEH) with the Sentinel imagery (Morton et al., 2021);

- The 30-m China Land Use Dataset (CLUD) covering the whole China from the 1980s to 2015 at an interval of 5 years, which was produced by the Chinese Academy of Sciences with multitemporal Landsat imagery (Liu et al., 2014);

- The 30-m China Land Cover Dataset (CLCD) covering the whole China from 1990 to 2019 annually, which was produced by Wuhan University with multitemporal Landsat imagery and Google Earth Engine (Yang & Huang 2021).

- The 10-m Cross Resolution Land Cover (CRLR 2020) covering the whole China in the year 2020, which was generated by Wuhan University with Sentinel imagery and deep learning framework (Liu et al., 2023).

**Table R1-1. Different types of land-cover datasets reviewed in the manuscripts. The supplements are colored in blue.**

[revised manuscript text omitted]

The cited references of the national-scale moderate-/high-resolution land-cover datasets are as follows:

Wickham, J., Stehman, S. V, Sorenson, D. G., Gass, L., & Dewitz, J. A. (2021). Thematic accuracy assessment of the NLCD 2016 land cover for the conterminous United States. Remote Sensing of Environment, 257, 112357. https://doi.org/10.1016/j.rse.2021.112357

Morton, R. D., Marston, C. G., O' Neil, A. W., & Rowland, C. S. (2021). Land Cover Map 2020 (10m classified pixels, GB). NERC EDS Environmental Information Data Centre. https://doi.org/10.5285/35c7d0e5-1121-4381-9940-75f7673c98f7

Liu, J., Kuang, W., Zhang, Z., Xu, X., Qin, Y., Ning, J., Zhou, W., Zhang, S., Li, R., & Yan, C. (2014). Spatiotemporal characteristics, patterns, and causes of land-use changes in China since the late 1980s. Journal of Geographical Sciences, 24, 195–210. https://doi.org/10.1007/s11442-014-1082-6

Yang, J., & Huang, X. (2021). The 30m annual land cover dataset and its dynamics in China from 1990 to 2019. Earth System Science Data, 13(8), 3907–3925. https://doi.org/10.5194/essd-13-3907-2021

Liu, Y., Zhong, Y., Ma, A., Zhao, J., & Zhang, L. (2023). Cross-resolution national-scale land-cover mapping based on noisy label learning: A case study of China. International Journal of Applied Earth Observation and Geoinformation, 118, 103265. https://doi.org/10.1016/j.jag.2023.103265

**(2) Why not use bands composition to assist in mapping?**

**Response:**

Thank you for the constructive comment. We have carefully considered this question during the production process of SinoLC-1. Since the VHR Google Earth images contain three basic bands which are difficult to apply common band composition process, we agree that using multi-spectral images (e.g., Sentinel-2) or composition index data (e.g., NDVI, NDWI, etc.) enables to assist the mapping process. Specifically, I would like to respond this comment from three aspects, which are concluded during the practical production of the SinoLC-1:

- (1/3) The additional information provided by multi-spectral images and band composition data.

    In the moderate-/high-resolution land-cover mapping process, multi-spectral images from Landsat or Sentinel mission general provide abundant spectral information which can better distinguish confused land-cover types. This enables land-cover products produced based on these images to contain reasonable classification results on these confused land-cover types. The production process of SinoLC-1 was based on the classification results of these multi-spectral

images, where these products (i.e., ESA_GLC10, FROM_GLC10, and ESRI_GLC10) were used as training labels. By combining the rich labeled information of these training labels and the fine edge and texture information of three band VHR Google images, the classification results based on multi-spectral images with a fine edge and details can be learned and inherited by SinoLC-1 to a certain extent.

As shown in the top row of Figure R1-1, SinoLC-1 has learned and inherited these classification results (e.g., water, vegetation, impervious) of 10-m GLC produced by using multispectral images. As shown in the bottom row of Figure R1-1, we collected the multi-spectral Sentinel-2 image from the same location and used common band composition to generate index data for NDVI and NDWI, the information contained in the index data is basically reflected in the GLC training labels.

[Figure]

**Figure R1-1. Demonstration of different data in a sample area of Shanghai.**

- (2/3) The mismatched resolution between the VHR optical images, multi-spectral images, and band composition index data.

    The 10-m Sentinel-2 imagery, as a suitable multi-spectral auxiliary data to conduct band composition, has a 10 times spatial resolution discrepancy to the using VHR Google Earth images. As shown in the top row of Figure R1-2, the mismatched resolution between the VHR

optical images, multi-spectral images, and band composition index data can cause data offset when they are input to the network for training. Furthermore, the land-cover mapping network used in this manuscript includes a resolution-preserving backbone, a weakly supervised label selection module, and a self-supervised loss function. Although the network performs well in resolving the resolution offset between images and labels by highly preserving the resolution of extracted features, the land-object edge and spatial detail of the mapping results inevitably rely on the input VHR images. Besides, in our previous works, we have validated that taking multi-spectral images with lower resolution as auxiliary data for the VHR land-cover mapping process can bring in abundant spectral information, but reduce the spatial details of the mapping results.

Figure R1-2 shows the masks generated from the index data by setting different thresholds. In addition to the resolution offset between these data, different threshold setting has a significant impact on the mask generated by these band composition index data. In the national-scale land-cover mapping process, the variation of band composition index data in different regions can make it difficult to select appropriate thresholds for mask generation.

[Figure]

(a) An example of NDVI                         (b) An example of NDWI

**Figure R1-2. Demonstration of the index data generated by the band composition**

● (3/3) Discussion on two ways of using band composition to assist the VHR national-scale land-cover mapping.

Based on the above-mentioned analysis, we have listed two suitable ways to use band composition to assist VHR national scale land cover mapping after fully considering the utilized framework and calculation resources. As shown in Figure R1-3, the listed two ways include: (a) Using Sentinel-2 imagery as auxiliary data to the VHR optical imagery and (b) Using band composition index data as post-processing data to the SinoLC-1 dataset.

(a) By combining the 13 multi-spectral bands of Sentinel-2 images with the current VHR images to reconstruct the input image data is a basic method to assist the land-cover mapping process. As shown in Figure R1-3 (a), this combination can maximize the utilization of spectral information provided by the Sentinel-2 image. However, in addition to the abovementioned resolution offset issue between different image sources, it requires us to collect the Sentinel-2 images covering the whole China, and the reconstructed image contains a large number of channels, which relies on more computational and storage capabilities for conducting the national-scale mapping process.

(b) By calculating the band composition index data and using them to assist the post-processing of the mapping results is another suitable method to improve the quality of the SinoLC-1 dataset. As shown in Figure R1-3 (b), with the index data (e.g., NDVI, NDWI, etc.) calculated based on multispectral images, it is possible to correct the misclassification land-cover types and improve the overall quality of the SinoLC-1. The advantage of such a method is that it does not require the reconstruction of training data (especially the images) and model retraining, but only uses appropriate band composition data to post-process the results. Therefore, we are trying our best to utilize the index data generated by the Sentinel-2 image to improve the current results and reevaluate the accuracy. In future work, we continue to reevaluate the improved results and update the dataset.

[Figure]

(a)  Using Sentinel-2 imagery as auxiliary data to the VHR optical imagery

[Figure]

(b)  Using band composition index data as post-processing data to the SinoLC-1 dataset.

**Figure R1-3. Two ways of using band composition to assist the VHR national-scale land-cover mapping.**

**(3) Why are other OSM types not involved in mapping?**

**Response:**

Thanks for the question. Before conducting the national-scale land-cover mapping, we conducted thorough research on the selection of input data. Open Street Map (OSM), as one of the most popular volunteer geographic information data sources, allows everyone in the community to edit the maps. As shown in Figure R1-4, OSM contains three types of data: points of interest, traffic routes, and polygons. Among them, the points of interest are usually labeled coordinate points without a systematic classification system. For example, in the example of Shanghai, the points of interest are labeled with the names of different restaurants, hotels, and coffee shops, which makes them difficult to utilize in land-cover mapping tasks. For polygons, their corner points are manually

labeled, and their attributes contain the basic land use types (such as commercial, industrial, etc.). For traffic routes, they are usually labeled by uploaders who carried with GPS receivers and updated to OSM by walking, cycling, or driving along the road. Based on this, traffic routes usually have more accurate labeling information.

[Figure]

**Figure R1-4. Demonstration of the OSM data in Shanghai City.**

To demonstrate different types of data in the OSM, we selected three sample areas in Figure R1-4 (a). For sample 1, it can be found that OSM's polygons data does not accurately label the boundaries of 'industrial', but instead labels most of the area of a factory. For samples 2 and 3, it can be observed that due to the inaccurate manual-annotation of OSM's polygons data, many basic land-use types such as industrial and residential areas in the same area miss annotation information. Figure R1-4 (c) shows the land-cover product which was used in the training process of the SinoLC-1. Since the land-cover products are interpreted from the remote-sensing images rather than manually annotating, the information provided by land-cover products is more complete.

[Figure]

**The example in Heihe City**

(a) 1-m VHR image

(2) Image with OSM

(c) Land-cover product (ESA WorldCover v100)

(d) Image with OSM (traffic route)

(e) Image with OSM (Polygons)

(f) Image with land-cover product

**Figure R1-5. Demonstration of the OSM data in Heihe City, Heilongjiang Province.**

Furthermore, Figure R1-5 shows another example of OSM data in Heihe City, Heilongjiang Province. Compared to Shanghai City, Heihe City is sparsely populated and has a lower level of urban development. From Figure R1-5 (d-f), the traffic route data of OSM is accurate, but the polygons data is relatively inaccurate (only labeling a rough area). The land-cover or land-use information provided by the polygons data is far less accurate than the land-cover product.

In general, OSM's traffic route data can provide additional information for land cover mapping tasks. In numerous land-cover mapping studies, OSM's traffic route data is also widely used (Audebert et al 2017; Guzder-Williams et al, 2022; Zhu et al, 2022), because they provide accurate road-labeled information and reflect urban patterns. For the national-scale land-cover mapping process, the examples shown in Figure R1-5 reveal that the data quality of OSM's polygons has significant differences in different regions, and inaccurate manual-annotation may also bring more label noise. Therefore, in the production of SinoLC-1, we obtained the land-cover information from three GLC products, and only extracted the accurate traffic route data from the OSM to construct the training labels.

The cited references of this response are as follows:

Guzder-Williams, B., Mackres, E., Angel, S., Blei, A.M. and Lamson-Hall, P., 2023. Intra-urban land use maps for a global sample of cities from Sentinel-2 satellite imagery and computer vision. Computers, Environment and Urban Systems, 100, p.101917. https://doi.org/10.1016/j.compenvurbsys.2022.101917

Zhu, Q., Lei, Y., Sun, X., Guan, Q., Zhong, Y., Zhang, L. and Li, D. (2022). Knowledge-guided land pattern depiction for urban land use mapping: A case study of Chinese cities. Remote Sensing of Environment, 272, p.112916. https://doi.org/10.1016/j.rse.2022.112916

Audebert, N., Le Saux, B. and Lefèvre, S. (2017). Joint learning from earth observation and OpenStreetMap data to get faster better semantic maps. In Proceedings of the IEEE Conference on Computer Vision and Pattern Recognition Workshops 67-75.

**(4) It's possible to argue against the classification system's building category. Table 2 compares building to mining land in the NLRS, which is inappropriate because mining land refers to a mine site (see the NLRS land category determination rules published in 2019).**

**Moreover, optical images and even RGB images should have difficulty classifying forest swamps. The authors are suggested to submit mapping results for land cover types that are challenging to distinguish in medium resolution imagery in order to show the scientific significance and applicability of SinoLC-1.**

**Response:**

Thank you for the constructive comments. For the first comment, we carefully checked the document 'Detailed Rules for the Recognition of Land Classification in the Third National Land Survey (2019)' at the website of the Ministry of Natural Resources of the People's Republic of China (https://m.mnr.gov.cn/zt/td/dscqggtdc/zl/201906/P020190604539900543194.pdf). Indeed, the mining land is inappropriate to be sorted into 'building' type. According to the rules, the type of mining land in NLRS refers to ground production lands such as mining, quarrying, sand (sand) quarries, brick and tile kilns, as well as soil (stone) and tailings storage areas. Therefore, we revised the corresponding land-cover type relationship between the SinoLC-1 products and the 3$^{rd}$ NLRS shown in Table 2 of the manuscript (Table R1-2 of the response letter) and the statistical validation set collected from the third national land resource survey projects shown in Table 3 of the manuscript (Table R1-3 of the response letter). Furthermore, based on the revised classification system and relationship table of the land-cover types, we updated the statistical-level validation results shown in Figures 21−23 of the revised manuscript.

[revised manuscript text omitted]

**(5) Add legends to all maps to address the current difficulty of comparing different product qualities, such as Figure 13.**

**Response:**

Thanks for your constructive suggestions. We have supplemented the legends to all maps in Figures 8, 13, and 14 of the revised manuscript (Figures R1-7, R1-8, and R1-9). In the comparison of different products, we demonstrated the original land-cover types of different products and unified the color of similar land-cover types for better visual comparison.

[Figure]

**Figure R1-7. Demonstration of SinoLC-1: a 1-meter-resolution national-scale land-cover map of China.**

[Figure]

**Figure R1-8. Demonstration of the visual comparison for Changzhou City, Jiangsu Province. The VHR remote sensing image in the figure is from © Google Earth 2021.**

[Figure]

**Figure R1-9. Demonstrations of the visual comparison for four typical regions. The VHR remote sensing images in the figure are from © Google Earth 2021.**

**(6) The authors utilized current global-scale land cover products as mapping samples, but the quality of them in the Chinese region is uncertain. The quality of these products in the Chinese region is not always robust according to the text and figures in section 4.2 of the manuscript. Therefore, how do the authors account for these variables that might affect SinoLC-1's quality?**

**Response:**

Thank you for your constructive comment. We agree that using global-scale land-cover products as training data for SinoLC-1 may bring uncertainty. To be clearer and in accordance with your concerns, we would like to respond to this question from three aspects, including (1) Analyzing the sources of uncertainty in training labels; (2) Analyzing how we reduce the uncertainty during sample selection and network training; (3) Comprehensively analyzing the impact of uncertainty on the SinoLC-1 mapping results.

● (1/3) Unstable sample brought by the low-resolution, outdated, noisy training labels.

The manual annotation of labeled data is laborious and time-consuming, which challenges the efficiency, expenditure, and applied coverage of the VHR land-cover mapping. Based on this situation, the SinoLC-1 dataset was produced by using low-resolution land-cover products as training labels. According to the production of SinoLC-1, we sorted out the noise sources of labeled data into three main parts:

  (a) Spatial resolution mismatch

    There is a resolution gap between the 1-meter images and the 10-meter labels. The coarse label brought noisy samples to the fine edges and texture info of VHR images during the training process.

  (b) Temporal mismatch

    There is a temporal gap between the VHR images and the adopted 10-meter resolution GLC products. The land-cover maps produced at different time points brought noisy samples.

  (c) Product defects:

    Due to the defects of classifiers and the insufficient image quality, the using GLC products may have incorrect results, which brought labeling errors and noisy samples to the production of SinoLC-1.

Figure R1-10 shows the samples of these three main noise sources and the results of the SinoLC-

1 which are accurate and consistent with the VHR images.

[Figure]

**Figure R1-10. Demonstration of the main uncertainty and unstable samples exiting in the training labels**

- (2/3) Reliable training sample collection and network training process.

  To reduce the impact of uncertainty during the production of SinoLC-1, we conducted a reliable sample collection and network training process. Firstly, Figure R1-11 shows the details of the training sample collection process. The land-cover types of three 10-m global land-cover products were unified, and then they were intersected to generate the label-selected mask. In the selected mask, the pixels/areas, where their land-cover types were the same in the three GLC products, would be preserved as the stable labeled areas; otherwise, the pixels/areas would be set as unlabeled type and maintained void value. As an example of the selected training sample shown in Figure R1-12, we demonstrate three typical areas where the first area shows a preserved correct sample (three GLC products have the same type), the second area shows an inaccurate sample (partial samples are abandoned), and the third area shows an incorrect sample (the samples are completely abandoned). Based on this sample collection process, the stable parts of these GLC products were preserved and the uncertain parts were abandoned, which ensures the reliability of the training labels. Secondly, to address the noisy label issue, the low-to-high network (L2HNet) was designed with a weakly-supervised based Confident Area Selection

(CAS) module and a self-supervised loss function. As the network training process shown in Figure R1-13, the CAS module selects the high-confident samples from the coarse, outdated, and noisy labels based on the confidence probability of the prediction batches. For the loss calculation, the Cross-Entropy (CE) loss is only calculated on the selected confident area, and the vague area (with low confidence probability) is ignored in the CE loss calculation. Then, the self-supervised Dynamic Vague Area (DVA) loss is calculated between the confident area and the vague area by constraining the feature similarity of the same land-cover types. Based on these components, the L2HNet enables to learn reliable information from the coarse, outdated, noisy labels, and the capacity of L2HNet to utilize noisy labels for accurate large-scale VHR land mapping has been validated in numerous datasets (Li et al., 2022).

In general, by combining the reliable training sample collection and network training process, the impact of uncertainty in training labels could be reduced to a certain extent during the production of SinoLC-1.

[Figure]

**Figure R1-11. Demonstration of the training sample collection process**

[Figure]

**Figure R1-12. Demonstration of the selected training samples**

[Figure]

**Figure R1-13. Demonstration of the training process of the land-cover mapping network.**

● (3/3) Comprehensively analyzing the impact of uncertainty on mapping results.

To comprehensively analyze the uncertainty of three global land-cover products, which were used to generate the SinoLC-1, more rigorously, we added two widely used open-access validation datasets to assess the accuracy of five global-scale products (including three utilized 10-m products and other two 30-m products) across China. Figure R1-14 shows the

supplemented workflow added to comprehensively evaluate the accuracy and uncertainty of the SinoLC-1 and other land-cover products. We have fully evaluated their producer accuracy (P.A.), user accuracy (U.A.), overall accuracy (O.A.), and kappa coefficients for each land-cover type in China, which are presented in Table R1-8, Figure R1-19, and Figure R1-20. We also analyzed their potential impact on the production process of SinoLC-1 comprehensively.

[Figure]

**Figure R1-14. The supplemented workflow to evaluate the accuracy and uncertainty of the SinoLC-1 and other five global land-cover products**

Firstly, the utilized two open-access validation datasets are created based on multiple data sources and manual verification, reporting a stable quality and high independence. The detailed information on these validation sets is as follows:

(a) Validation set created by Liu et al. DOI: https://doi.org/10.5281/zenodo.3551995

Liu et al. (2019) created a global land-cover validation set by combining several existing reference datasets such as the GLCNMO2008 training dataset, VIIRS reference dataset, STEP reference dataset, Global cropland reference data, and so on to guarantee the confidence and objective of the validation samples. Furthermore, high-resolution imagery in Google Earth and time-series NDVI, NDSI values of each related point were integrated to derive the validation datasets.

(b) Validation set created by Zhao et al. DOI: https://doi.org/10.1080/01431161.2014.930202

Zhao et al. (2014) created a global land-cover validation set with a total of 38,664 sample units by interpreting Landsat images and MODIS EVI time series data, as well as high-resolution images from Google Earth, recording the quality of reference data, and interpreter confidence. Zhao et al. confirmed that the dataset had been carefully improved through several rounds of interpretation and verification by different image interpreters, and checked by one quality controller. Independent test interpretation indicated that the quality control correctness level reached 90% at level 1 land-cove type.

According to the description of the data providers, these validation sets contain two levels of land-cover types, and their spatial distribution and classification system are shown in Figure R1-15, Table R1-7, and Table R1-8.

The cited references of this response are as follows:

Zhao, Y., Gong, P., Yu, L., Hu, L., Li, X., Li, C., Zhang, H., Zheng, Y., Wang, J., Zhao, Y. and Cheng, Q. (2014). Towards a common validation sample set for global land-cover mapping. International Journal of Remote Sensing, 35(13), 4795-4814. https://doi.org/10.1080/01431161.2014.930202

Liu, L., Gao, Y., Zhang, X., Chen, X., & Xie, S. (2019). A Dataset of Global Land Cover Validation Samples (Version v1) [Data set]. Zenodo. https://doi.org/10.5281/zenodo.3551995

[Figure]

(a)  Validation set created by Liu et al.     (b) Validation set created by Zhao et al.

**Figure R1-15. Demonstration of two open-access validation set.**

**Table R1-4. The classification system of the validation set created by Liu et al.**

| Level 1 type | Level 2 type | Sample count | Total | Proportion (%) |
|---|---|---|---|---|
| Cropland | Rainfed cropland | 44 | | |
| | Herbaceous cover | 0 | 353 | 14.33% |
| | Irrigated cropland | 311 | | |
| Forest | Evergreen broadleaved forest | 123 | | |
| | Deciduous broadleaved forest | 303 | 542 | 22.01% |
| | Mixed leaf forest | 116 | | |
| Shrubland | Shrubland | 78 | 104 | 4.22% |
| | Evergreen shrubland | 26 | | |
| Grassland | Grassland | 360 | 360 | 14.62% |
| Wetlands | Wetlands | 17 | 17 | 0.69% |
| Impervious surfaces | Impervious surfaces | 71 | 71 | 2.88% |
| Bare areas | Sparse vegetation | 285 | | |
| | Bare areas | 329 | 641 | 26.03% |
| | Consolidated bare areas | 3 | | |
| | Unconsolidated bare areas | 24 | | |
| Water body | Water body | 37 | 37 | 1.50% |
| Permanent ice and snow | Permanent ice and snow | 338 | 338 | 13.72% |

**Table R1-5. The classification system of the validation set created by Zhao et al.**

| Level 1 type | Level 2 type | Sample count | Total | Proportion (%) |
|---|---|---|---|---|
| Crop | Rice | 3 | | |
| | Greenhouse | 1 | 353 | 16.98% |
| | Other | 349 | | |
| Forest | Broadleaf | 303 | | |
| | Needleleaf | 81 | 512 | 24.63% |
| | Mixed | 114 | | |
| | Orchard | 14 | | |
| Grass | Managed | 0 | 312 | 15.01% |
| | Nature | 312 | | |
| Shrub | Shrub | 103 | 103 | 4.95% |
| Wetland | Grass | 15 | 15 | 0.72% |
| | Silt | 0 | | |
| Water | Lake | 7 | | |
| | Pond | 19 | 33 | 1.59% |
| | River | 7 | | |
| | Sea | 0 | | |
| Impervious | High albedo | 19 | 52 | 2.50% |
| | Low albedo | 33 | | |
| Bare land | Saline-Alkali | 10 | | |
| | Sand | 138 | | |
| | Gravel | 303 | 619 | 29.77% |
| | Bare-cropland | 89 | | |
| | Dry river/lake bed | 2 | | |
| | other | 77 | | |
| Snow and Ice | Snow | 80 | 80 | 3.85% |
| | Ice | 0 | | |

Secondly, to comprehensively validate the accuracy and uncertainty of the SinoLC-1, we calculated the confusion matrix of SinoLC-1 and further validated its P.A., U.A., O.A., and kappa coefficient based on two open-access validation sets. As shown in Table R1-6 and Table R1-7, the

O.A. of the SinoLC-1 validated on the validation sets created by Liu et al. and Zhao et al. are 78.80% and 64.69%, respectively. The Kappa of the SinoLC-1 validated on the validation sets created by Liu et al. and Zhao et al. are 0.7394 and 0.5588, respectively. Furthermore, to illustrate more detailed assessment results, Figure R1-16 shows the corresponding confusion proportions for each considered land-cover type of the SinoLC-1 validated on two datasets.

**Table R1-6. Confusion matrix for the SinoLC-1 according to the validation set created by Liu et al.**

| Classification | TC | SL | GL | CL | IP | BL&SV | S&I | WT | WL | Total | P.A. (%) |
|---|---|---|---|---|---|---|---|---|---|---|---|
| Tree Cover | 421 | 5 | 80 | 32 | 0 | 2 | 1 | 1 | 0 | 542 | 77.68 |
| Shrubland | 7 | 34 | 32 | 2 | 0 | 27 | 2 | 0 | 0 | 104 | 32.69 |
| Grassland | 2 | 2 | 342 | 0 | 0 | 7 | 0 | 0 | 0 | 353 | 96.88 |
| Cropland | 5 | 1 | 3 | 316 | 29 | 3 | 0 | 3 | 0 | 360 | 87.78 |
| Impervious | 7 | 0 | 3 | 7 | 51 | 3 | 0 | 0 | 0 | 71 | 71.83 |
| Barren &Sparse veg. | 1 | 7 | 12 | 0 | 0 | 616 | 5 | 0 | 0 | 641 | 96.10 |
| Snow and ice | 1 | 0 | 78 | 0 | 0 | 121 | 136 | 2 | 0 | 338 | 40.24 |
| Water | 7 | 0 | 0 | 5 | 3 | 1 | 0 | 19 | 2 | 37 | 51.35 |
| Wetland | 0 | 0 | 1 | 4 | 0 | 1 | 2 | 3 | 6 | 17 | 35.29 |
| Total | 451 | 49 | 551 | 366 | 83 | 781 | 146 | 28 | 8 | 2463 | |
| U.A. (%) | 93.35 | 69.39 | 62.07 | 86.34 | 61.45 | 78.87 | 93.15 | 67.86 | 75.00 | | |
| O.A. (%) | | | | | 78.80 | | | | | | |
| Kappa | | | | | 0.7394 | | | | | | |

Note: TC=Tree cover; SL=Shrubland; GL=Grassland; CL=Cropland; IP=Impervious (Building and traffic route); BL&SV=Barren and sparse vegetation; S&I=Snow and ice; WT=Water; WL=Wetland

**Table R1-7. Confusion matrix for the SinoLC-1 according to the validation set created by Zhao et al.**

| Classification | TC | SL | GL | CL | IP | BL&SV | S&I | WT | WL | Total | P.A. (%) |
|---|---|---|---|---|---|---|---|---|---|---|---|
| Tree Cover | 376 | 13 | 60 | 35 | 6 | 7 | 1 | 5 | 9 | 512 | 73.44 |
| Shrubland | 10 | 34 | 33 | 3 | 0 | 21 | 2 | 0 | 0 | 103 | 33.01 |
| Grassland | 37 | 2 | 215 | 20 | 3 | 33 | 0 | 1 | 1 | 312 | 68.91 |
| Cropland | 83 | 1 | 36 | 191 | 18 | 21 | 0 | 2 | 1 | 353 | 54.11 |
| Impervious | 3 | 0 | 4 | 13 | 29 | 3 | 0 | 0 | 0 | 52 | 55.77 |
| Barren &Sparse veg. | 13 | 7 | 109 | 62 | 6 | 411 | 5 | 5 | 1 | 619 | 66.40 |
| Snow and ice | 1 | 0 | 5 | 0 | 0 | 8 | 65 | 1 | 0 | 80 | 81.25 |
| Water | 6 | 0 | 0 | 7 | 2 | 2 | 0 | 16 | 0 | 33 | 48.48 |
| Wetland | 0 | 0 | 2 | 0 | 1 | 2 | 2 | 0 | 8 | 15 | 53.33 |
| Total | 529 | 57 | 464 | 331 | 65 | 508 | 75 | 30 | 20 | 2079 | |
| U.A. (%) | 71.08 | 59.65 | 46.34 | 57.70 | 44.62 | 80.91 | 86.67 | 53.33 | 40.00 | | |
| O.A. (%) | | | | | 64.69 | | | | | | |
| Kappa | | | | | 0.5588 | | | | | | |

Note: TC=Tree cover; SL=Shrubland; GL=Grassland; CL=Cropland; IP=Impervious (Building and traffic route); BL&SV=Barren and sparse vegetation; S&I=Snow and ice; WT=Water; WL=Wetland

(a) Confusion proportions for land-cover type of the SinoLC-1 validated with the set created by Liu et al.

(b) Confusion proportions for land-cover type of the SinoLC-1 validated with the set created by Zhao et al.

**Figure R1-16. Confusion proportions of the validation results.**

Thirdly, to assess the uncertainty impact of three utilized 10-m land-cover products more rigorously and transparently, and to conduct a more complete comparison, we used these validation sets to validate the accuracy of five comparative land-cover datasets (including three utilized 10-m products and other two 30-m products). Figure R1-17 and Figure R1-18 show the spatial distribution of two validation sets among five comparative land-cover products in China. For clearer expression, we mark the validation set created by Liu et al. (2019) as S1 and mark the set created by Zhao et al. (2017) as S2.

The comparison results are shown in Table R1-8 and Figure R1-19. From the quantitative comparison, the SinoLC-1 has the second highest O.A. on two validation sets where the SinoLC-1 has a O.A. of 0.6469 with S1 (lower than the 10-meter ESA_GLC10) and has an O.A. of 0.7881 with S2 (lower than the 30-meter GLC_FCS30). Furthermore, we compared the U.A. of every considered type between the SionLC-1 and the other five products in Figure R1-20. From the results shown in Figure R1-20 (a), the SinoLC-1 has the second highest U.A. in types of 'Tree canopy', 'Shrubland', 'Grassland', and 'Wetland' compared to the other five products, and has the U.A. of 'Cropland' and 'Impervious surface' surpassing the average of other five products. From the results shown in Figure R1-20 (b), the SinoLC-1 has the highest U.A. in types of 'Shrubland' and 'Grassland', and has the U.A. of 'Snow and ice' and 'Wetland' surpassing the average of the other five products.

In general, by quantitatively comparing the SinoLC-1 product with five widely used land-cover products on two open-access validation datasets, the produced SinoLC-1 shows acceptable confusion proportion among land-cover types and has competitive accuracy among the other land-cover products across China. The quantitative results are used to explain the uncertainty of the three training data and further demonstrate that the production process of the SinoLC-1 reduced the impact of noise labels to a certain extent. The supplemented quantitative comparison and analysis were added in Section 4.2.2 of the revised manuscript.

[Figure]

**Figure R1-17. Demonstration of five comparison products and the validation set (S1) created by Liu et al.**

[Figure]

**Figure R1-18. Demonstration of five comparison products and the validation set (S2) created by Zhao et al.**

**Table R1-8. Quantitative comparison between the SionLC-1 and other five land-cover products.**

| Metric / Dataset | Validation set of Zhao et al. | | Validation set of Liu et al. | |
|---|---|---|---|---|
| | O. A. | Kappa | O. A. | Kappa |
| SinoLC-1 | 0.6469 | 0.5588 | 0.7881 | 0.7394 |
| ESA_GLC10 | 0.6646 | 0.5722 | 0.7356 | 0.6269 |
| FROM_GLC10 | 0.6411 | 0.5942 | 0.7538 | 0.6871 |
| ESRI_GLC10 | 0.6232 | 0.5210 | 0.6675 | 0.5972 |
| GlobaLand30 | 0.6209 | 0.5285 | 0.7694 | 0.7090 |
| GLC_FCS30 | 0.5778 | 0.4675 | 0.8684 | 0.8241 |

[Figure]

(a)   The validation results based on S1         (b)   The validation results based on S2

**Figure R1-19. The quantitative validation and comparison of the SinoLC-1 and other five products**

[Figure]

(a)   The U.A. comparison based on S1       (b)   The U.A. compassion based on S2

**Figure R1-20. The U.A. comparison of the SinoLC-1 and other five products**

**(7) It is challenging to automatically map forests, shrubs, grasslands, wetlands, and tundra using medium-resolution images. To help the reader comprehend the characteristics of various land cover types in Google images, it is advised that the authors change Figure 6 by adding VHR samples.**

**Response:**

Thanks for the constructive comment. We agree that some of the land-cover types are challenging to identify in medium-resolution images due to their low spatial details in the images. According to your suggestion, we have added the VHR samples captured from the 1.07-m Google Earth images for all the land-cover types of the SinoLC-1. As shown in Figure 6 of the revised manuscript (Figure R1-21 of the response letter), every land-cover type includes three VHR samples to help the readers comprehend their characteristics.

[Figure]

**Figure R1-21. Demonstration of the sample grid, VHR samples, and the national validation sample set. Left: the spatial distributions of the sample set (the legend is written in shorter forms). Right: the VHR samples of different land-cover types collected from 1.07-m resolution © Google Earth imagery all around China.**

**(8) The area discrepancies between the provincial land cover categories of SinoLC-1 and NLRS are compared in Figure 17. It is important to note that the value interval on the vertical axis is too big. For instance, in Henan province, each pitch of the vertical axis corresponds to a 5,000 km$^2$ gap. Thus, the area difference between the two results cannot be well reflected for land cover categories with small areas. It is suggested that the authors seek alternative comparison methods to make the area difference between all types of land cover clear.**

**Response:**

We appreciate this helpful feedback to increase the statistical comparison between the SinoLC-1 and 3$^{rd}$ NLRS. According to your feedback, we changed Figure 21 in the revised manuscript to directly illustrate the misestimation area between sinoLC-1 and 3rd NLRS under each land-cover type. To better demonstrate the differences between SinoLC-1 and 3$^{rd}$ NLRS, the overestimation (positive value) and underestimation (negative value) of SinoLC-1 are also reflected in Figure 21 of the manuscript (Figure R1-22 of the response letter). To make the area difference between all land-cover types clearly visible, we used breakpoints to illustrate the excessively large values of misestimation area to ensure that the land-cover types with large gaps can be reasonably displayed in the same vertical axis.

Furthermore, we have also taken your consideration in the comparison at the national scale. In the box chart shown in Figure 23 (a) of the manuscript (Figure R1-23 (a) of the response letter), we have changed the vertical axis from 'misestimation area (km$^2$)' into 'misestimation rate', because the coverage and misestimation area of different land-cover types has significant differences, which makes it difficult to reflect in the box chart with the vertical axis of 'area (km$^2$)'. Moreover, we supplemented Figure 23 (c) in the revised manuscript (Figure R1-23 (b) of the response letter) to show the overall misestimation rates of SinLC-1 in the whole China and evaluated the performance of Sinolc-1 from a statistical perspective which has a misestimation rate under 10% among all the land-cover types.

In addition, to demonstrate the spatial distribution of the misestimation rate for each land-cover type across China, and to provide more comparable information on the statistical assessment, we have collected the results and added the map of the misestimation rate for every land-cover type in Figure 22 of the revised manuscript (shown in Figure R1-24 of the response letter).

[Figure]

**Figure R1-22. Statistical comparison between SinoLC-1 and 3rd NLRS data for 31 provinces in China. The provinces in different geographical region are represented by dissimilar wireframe colors. In every subplot, the abscissa axis represents the land-cover types, and the vertical axis represents the misestimation area.**

[Figure]

(a) Overall misestimation rate of every land-cover type through 31 provinces in China

(b) National misestimation rate of every land-cover type across China

**Figure R1-23. Overall misestimation distributions in every land-cover type across China.**

[Figure]

**Figure R1-24. The misestimation rate of SinoLC-1 for 31 provinces in China. In every subplot, the statistical comparison between SinoLC-1 and 3rd NLRS data in every land-cover type is illustrated.**

**(9) It is advised that the authors use more inclusive language, primarily in section 4.2, where words like "worst" need to be changed.**

**Response:**

Thank you for the comments, we have carefully checked the whole manuscript. Primarily, in Section 4.2.1, we changed the word "the worst performance" to "limited performance". Then we inclusively analyzed the performance of different products, especially comprehensive descriptions of every comparative product in different land-cover types and demonstration areas. Furthermore, the revised manuscript has been carefully proofread by a language editor, and we have made numerous changes on the expressions to make sure the language is inclusive.

**(10) Checking the terms and some phrases is advised, e.g., "OBAI" in line 104 and "cropped" in line 199.**

**Response:**

Thank you for the corrections. We have corrected the grammar issue and checked the whole manuscript with the help of language editors. We corrected the "OBAI" in line 104 to "OBIA" and changed the expression of 'cropped ' to 'divided'.

---

## Author Response (AR2)

**Referee #1**

**General comments:**

**The issues have been addressed in the manuscript. My suggestion is Accept**

**Response:**

We appreciate your considerable comments and decision. We have carefully revised the manuscript and made technical corrections to further improve the manuscript's quality.
* * *
**Referee #3**

We thank the reviewer and editor for a thoughtful and thorough review of our manuscript (ESSD-2023-87) and for giving the decision on minor revisions. We have carefully addressed all the comments and revised the manuscript. The suggestions and comments are listed in **bold** type. The modified words or materials are marked as blue color in the revised manuscript. The item-by-item responses to all comments are listed below.

**General comments:**

**The manuscript takes on the very substantial challenge of developing a national land-scale land cover of China with 1-meter spatial resolution. The newly developed very-high-resolution (VHR) land cover data was established using a deep learning-based framework and open-access data including global land-cover (GLC) products, open street map (OSM), and Google Earth imagery. In general, the revised manuscript is well structured, and authors did a lot of work in addressing the two reviewer's comments, especially the data validation. However, the texts about validation and comparison still need to be improved. Some detailed suggestions and comments were provided below. Overall, this paper could be publishable after revision.**

**Response:**

We are grateful for your careful reading, and for giving us constructive comments. According to the suggestions and comments, we have carefully considered all of them and tried our best to improve the manuscript before it could be published especially the validation and comparison parts.

**Suggestions and comments:**

**(1) Line 187 and Table 1. In this manuscript, the authors defined a classification system, including 11 land cover types. Though you tried to match the classification definition between FROM_GLC10, ESA_GLC10, ESRI_GLC10 and SinoLC-1. However, the land cover class definition between the 10-m resolution is also different. For example, the tree cover in ESA_GLC10 includes any geographic area dominated by trees with a cover of 10% or more. Forest in FROM_GLC10 limits tree cover percentage classification to >15%, limits tree height classification to >3 m (Gong et al., 2012, IJRS). Trees in ESRI_GLC10 is defined as any significant clustering of tall (~15 feet or higher) dense vegetation, typically with a closed or dense canopy[1]. Other land use types should also have large differences. So, how do you consider such differences when selecting training and validation samples? Besides, a clear definition for each land use type in SinoLC-1 should be given in a table.**

**Response:**

We appreciate the questions and suggestions from the reviewer. We agree that a clear definition for each land-cover type in the SinoLC-1 product is important for the users and downstream applications. The item-by-item responses to this comment are listed below.

During the selection of training and validation samples, we considered three main issues and tried to resolve them by conducting a weakly supervised strategy in the mapping framework. Firstly, as the reviewer mentioned, there is a difference between the type definition of the utilized global 10-meter products. Secondly, there are resolution gaps between the imagery and the 10-meter products. Thirdly, there is a temporal mismatch between the three utilized 10-meter products. The above-mentioned problems can be all regarded as "noisy label issues", where the training labels have unreliable samples. Figure R1 shows a case of the label generation process that takes the intersection parts of three products. Although there are label noises caused by the mismatched type definition, spatial resolution, and temporal resolution, the reminded samples shown in Figure R1 (e) are still relatively reliable. Based on such intersection labels, the result of the SinoLC-1 shown in Figure R1 (f) is accurate and consists of the 1-meter imagery.
* * *
[1] https://www.arcgis.com/home/item.html?id=d3da5dd386d140cf93fc9ecbf8da5e31

For giving a clearer explanation to the readers, firstly, we added a description of type unification in line 20 to descript the differences in type definitions among the three 10-meter products. Secondly, we supplemented the clear definition of each land-cover type of the proposed SinoLC-1 in Table R1 (Table 2 of the revised manuscript). By combining the actual situation of the proposed SinoLC-1 product and the information from the China Ministry of Natural Resources, the definitions including detailed descriptions and examples were given to better explain each land-cover type.

[Figure]

(a) 1-m Google imagery        (b) 10-m ESA_GLC10

Area dominated by trees with a cover of 10% or more.

(c) 10-m ESRI_GLC10        (d) 10-m FROM_GLC10

Any significant clustering of tall (~15 feet or higher) dense vegetation      Area with tree cover percentage classification to >15%, height >3 m.

(e) Intersection label        (f) 1-m SinoLC-1 (Our)

Trees that generally have larger crowns and higher than 5-m.

**Figure R1. A case (Tree cover) of the label generation process of the SinoLC-1.**

**Table R1. The definition, value, and color of each land-cover type of the SinoLC-1**

| Land-cover type | Definition | Value | Color | |
|---|---|---|---|---|
| Tree cover | Areas covered by trees generally have larger crowns and are higher than 5 meters. It can be sparse arbors or clustered forests which include evergreen forests, mixed forests, artificial forests, bamboo groves, etc. | 2 | (0, 100, 0) | |
| Shrubland | Areas covered by clusters of shrubs with a height below 5 meters. | 3 | (255, 190, 35) | |
| Grassland | Areas covered by low herbaceous plants. It generally includes natural grasslands with a fractional vegetation coverage greater than 5, rangeland with tree canopy density less than 0.3 or shrub canopy density less than 0.4, urban's vacant land dominated by grass, and other artificial grasslands. | 4 | (233, 255, 190) | |
| Cropland | The arable land and human planted crops not at tree height including upland crops (e.g., wheat, corn, potatoes, and cotton) and irrigated crops (e.g., paddy filed, lotus root, and water spinach). | 5 | (255, 235, 175) | |
| Building | Human-made structures and homogenous impervious surfaces including industrial, residential, commercial areas, and construction sites. It is generally located in urban and rural areas with high human activities. | 6 | (255, 170, 0) | |
| Traffic route | Areas constructed according to certain technical standards and equipped with necessary transportation facilities, including railways, highways, urban/rural roads, and pipelines. | 1 | (255, 0, 0) | |
| Barren and sparse vegetation | Areas covered by sparse vegetation or bare land covered by sand, gravel, or rocks, including mountains without dense vegetation and snow cover, deserts, grasslands degraded by drought, and wasteland in urban/rural areas with sparse or no vegetation. | 7 | (180, 180, 180) | |
| Snow and ice | Areas covered by large-scale permanent snow or ice, including glaciers and permanent snowpack in mountain areas or high latitudes. | 8 | (240, 240, 240) | |
| Water | Areas covered by water for a long period, including oceans, naturally formed water bodies (e.g., lakes, rivers, and runoff), and artificially formed water bodies (e.g., reservoirs, canals, water conservancy facilities, ponds, and aquaculture farms). | 9 | (0, 100, 200) | |
| Wetland | Areas with perennial or seasonal water accumulation and vegetation growth. It includes forest/shrub/grass swamps, peatlands, mudflats, mangroves, and coastal/inland tidal flats. | 10 | (0, 150, 160) | |
| Moss and lichen | Surfaces or rocks attached by moss or tiny lichen plants. | 12 | (250, 230, 160) | |

**(2) The Google Earth (GE) images were used for land cover classification, but it is like a figure with R, G, and B bands rather than the satellite image with multiple optical bands. Though a deep machine learning method was used to extract the features, the classification process was conducted at the pixel level. Object-based methods may have more advantages than pixel-based methods for VHR image classification. Why didn't choose the object-based classification method? As it should be, the pixel-based method can also generate accurate land cover. But a post-processing step should be included because the pixel-based method would always generate many fragmented patches.**

**Response:**

Thank you for your thoughtful consideration and constructive comments. We would like to respond to them in three aspects:

Firstly, by considering the very large-scale land-cover mapping process (i.e., the entire China covering 9,600,000 km$^2$), we designed a deep learning framework with a weakly supervised strategy to deal with the resolution mismatched issue between the training labels and imagery. On one hand, deep learning technique has been adopted in large-scale land-cover mapping in many cases where most of the deep learning frameworks are end-to-end trainable with pixel-based classification head (Li et al., 2022; Liu et al., 2023). However, most of the Object-based Image analysis (OBIA) methods are integrated into geographic software (e.g., ArcGIS and eCognition) which require human intervention and hand-craft labeled data[2]. On the other hand, the proposed framework in our manuscript has high efficiency in training and inference which shortens the time cycle for nationwide land-over mapping. However, as shown in Figure R2, the OBIA method is generally time-consuming in object generation and the parameters for generating objects often need to be carefully selected according to the situation. In conclusion, by considering the attributes of training data, the long span of the mapping areas, various landforms throughout China, and the efficiency of land-cover mapping, we chose the deep learning framework with a regular pixel-based classification head in the production of SinoLC-1.
* * *
[2] https://gisgeography.com/obia-object-based-image-analysis-geobia/

[Figure]

**Figure R2. ArcGIS segmentation using the Segment Mean Shift algorithm.**

[Figure]

**Figure R3 Demonstration of the mapping and merging for producing SinoLC-1.**

Secondly, as the mapping and merging process shown in Figure R3, the large-scale images need to be cropped into many tiles (6000×6000 pixels) and batches (256×256 pixels) during the training and inference process. The edges adjacent to image batches usually require taking the predicted average or majority of two adjacent batches to improve the results on the edge. However, it is difficult to the object-based classification since there are differences in the object boundaries of adjacent batches, which bring batches' edges mismatched during the large-scale land-cover mapping process.

Thirdly, we agree that the post-processing step can remove the fragmented patches inferred by the

pixel-based classification methods. According to your suggestion, we have tried to conduct the post-processing in many regions by fusing multimodel results and enhancing typical land-cover types (Li et al., 2021). As shown in Figure R4, we used two-step post-processing to improve the final results, and most of the salt and pepper look in the classification result was removed. We will attempt to conduct the post-processing throughout the product and continuously update the published version in the future.

[Figure]

(a) HR image       (b) Original result

(c) Post-processing (step 1)   (d) Post-processing (step 2)

**Figure R4. An example of different steps of the post-processing results**

The cited references of this response are as follows:

Li, Z., Zhang, H., Lu, F., Xue, R., Yang, G., & Zhang, L. (2022). Breaking the resolution barrier: A low-to-high network for large-scale high-resolution land-cover mapping using low-resolution labels. ISPRS Journal of Photogrammetry and Remote Sensing, 192, 244-267. https://doi.org/10.1016/j.isprsjprs.2022.08.008

Liu, S., Wang, H., Hu, Y., Zhang, M., Zhu, Y., Wang, Z., ... & Wang, F. (2023). Land Use and Land Cover Mapping in China Using Multi-modal Fine-grained Dual Network. IEEE Transactions on Geoscience and Remote Sensing. 10.1109/TGRS.2023.3285912

Li, Z., Lu, F., Zhang, H., Yang, G., & Zhang, L. (2021, July). Change cross-detection based on label improvements and multi-model fusion for multi-temporal remote sensing images. In 2021 IEEE International Geoscience and Remote Sensing Symposium IGARSS (pp. 2054-2057). IEEE. 10.1109/IGARSS47720.2021.9553120

**(3) Line 350-351. "The tree canopies and dense vegetation are mainly in the southern part and the northeast border of China". Here should be tree cover (defined in Table 1) rather than tree canopies and dense vegetation. Many different words were used to describe the tree cover in the texts and figures. For example, tree cover is labeled as tree in Figure 4b, Figure 6, Trees in Figure 8, Forest in Figure 13, Tree canopy in Figure 14, respectively. It will confuse readers and should be consistent in the text and figure legend.**

**Response:**

Thanks for the comments and corrections. We have checked the whole manuscript and corrected all descriptions of land-cover types which include the legends of Figure 4b, Figure 6, Figure 8, Figure 13, Figure 14, Figure 15, Figure 21, and Figure 22.

**(4) Line 488. Northern China, including Beijing, Tianjin, Hebei, Shanxi, and Inner Mongolia, have the lowest O.A. (lower than 70%) because the longitude span of the region is very wide, and the landscapes are diverse and various. For Inner Mongolia, the explanation looks good. But it is unreasonable for Beijing, Tianjin, Hebei, and Shanxi. I think the reason can be found in the confusion matrix of these provinces, especially which land cover type has the lowest accuracy.**

**Response:**

Thank you for the constructive comments. To better explain and analyze the classification results in Northern China, we carefully checked the confusion matrix of Beijing, Tianjin, Hebei, and Shanxi and summarized the confused land-cover types which cause the low accuracy in these regions. For Beijing, most of the misclassified samples are (1) the confusion between "Tree cover" and "Grassland"; (2) the confusion between "Building" and "Traffic route". For Tianjin, most of the misclassified sample is the confusion among "Cropland", "Building", and "Traffic route". For Hebei, most of the misclassified samples are (1) the confusion between "Tree cover" and "Grassland"; (2) the confusion between "Cropland" and "Grassland". For Shanxi, most of the misclassified samples are (1) the confusion among "Tree cover", "Grassland", and "Cropland"; (2) the confusion between "Building" and "Traffic route"; (3) the confusion between "Cropland" and "Barren & sparse vegetation". Detailed explanations were added in Section 4.3 "Quantitative analysis and accuracy assessment" to analyze the reason causing low accuracy in Northern China.

**(5) Section 4.2.2, Quantitative comparison with other land-cover products. You did the same validation work as that in section 4.3.1 rather than data comparisons between SinoLC-1 and the other five land cover datasets. I suggest moving this section to section 4.3.1 or doing a data comparison like section 4.3.2.**

**Response:**

Thanks for your constructive suggestions. We agree that changing the section order of quantitative comparison can better improve the structure and readability of the manuscript. We have moved Section 4.2.2 "Quantitative comparison with other land-cover products" to Section 4.3 "Quantitative analysis and accuracy assessment" which can better support the analysis of Section 4.3.1 "Pixel-level sample validation".

**(6) Figure 22 shows that there are significant differences in some provinces between SinoLC-1 and NLRS data. For example, the differences in tree cover area of Chongqing and Hainan are larger than 40%. Some explanations for such large differences should be given. Moreover, it is also looks weird to combine the grassland and barren into one type for comparison.**

**Response:**

Thank you for your constructive comment. For the first suggestion, we carefully checked the qualitative results and the statistical results of the SinoLC-1 and NLRS data. The corresponding analysis, which aims at explaining the large differences and misestimation of the statistical comparison, was supplemented in Section 4.3.3 "Statistical-level validation". Especially, according to the concerns in the comments regarding the results of Hainan and Chongqing, we demonstrated the qualitative comparison and the statistical comparison of the SinoLC-1 shown in Figure R5 and Figure R6. From the statistical comparison results, the SinoLC-1 of Hainan and Chongqing has a high overestimation of "Tree cover" and an underestimation of "Cropland". From the qualitative comparison results, the SinoLC-1 of Hainan and Chongqing Provinces are more consistent with the 10-m ESA_GLC10 where the "Tree cover" occupies more large areas than the FROM_GLC10 and ESRI_GLC10. In general, Hainan and Chongqing Provinces have a high proportion of "Tree cover" in practice, and the labels generated for model training retain massive samples of "Tree cover" in these two areas, which led to the model overfitting and overestimating the types of "Tree cover" (underestimating the type of "Cropland").

For the second question, we combined the land-cover types of "Grassland" and "sparse vegetation" by considering that the type of "Barren and sparse vegetation" also includes the areas with low and sparse grass cover. Besides, we carefully checked the document 'Detailed Rules for the Recognition of Land Classification in the Third National Land Survey (2019)' at the website of the Ministry of Natural Resources of the People's Republic of China (https://m.mnr.gov.cn/zt/td/dscqggtdc/zl/201906/P020190604539900543194.pdf). In the classification system of NRLS data, there is no land-cover type used to describe bare land with sparse vegetation. However, as shown in Table R2, the land-cover type of "Grassland" includes many subcategories that descript sparse vegetation coverage areas. Furthermore, in the classification system of SinoLC-1, the "Barren and sparse vegetation" also includes the grasslands degraded by drought and wasteland in urban/rural areas with sparse or no vegetation. In general, based on the classification system of NLRS data and SinoLC-1, we combined the "Grassland" and "Barren and sparse vegetation" of the SinoLC-1 into one type for comparison with the "Grassland" of NLRS data. Especially in Northwest China (Figure R7) where there is a large-scale mixed distribution of "Grassland" and "Barren and sparse vegetation", the strategy can be a suitable way to evaluate the overall performance of SinoLC-1.

[Figure]

Figure R5. The qualitative and statistical results of the SionLC-1 in Hainan Province.

[Figure]

(a) Comparison between the SinoLC-1 and NLRS

(b) FROM_GLC10

(c) ESA_GLC10

(d) ESRI_GLC10

(e) SinoLC-1

**Figure R6. The qualitative and statistical results of the SionLC-1 in Chongqing Province.**

[Figure]

**Figure R7. Demonstration of SinoLC-1 land-cover product in Northwest China.**

**Table R2. The definition of "Grassland" in the NLRS data.**

| Land-cover type of NLRS data | The definition of NLRS data |
|---|---|
| Grassland | 1. Grasslands mainly composed of natural herbaceous plants, used for grazing or mowing, including grasslands where grazing prohibition measures are implemented. |
| | 2. A grassland artificially planted with grass. |
| | 3. Grassland improved by irrigation, drainage, fertilization, loosening, and replanting. |
| | 4. The surface layer of soil with a canopy density of less than 0.1. |
| | 5. Land used for scientific research, experimentation, and demonstration of herbaceous plants. |
| | 6. Due to factors such as engineering needs and improving the living environment, migration, and relocation have caused natural growth of herbaceous plants in villages, farmland, and other areas. |
| | 7. Land outside the scope of land acquisition for railways and highways, or the ditches without land acquisition, artificially planted and fixed for greening and beautifying the environment with herbaceous plants. |
| | 8. Land outside urban or villages which is artificially planted with herbaceous plants for greening and beautifying the environment. |
| | 9. Land used for cultivating herbaceous plant seeds. |
| | 10. In the grassland, supporting facilities that directly serve animal husbandry, such as land for storing forage and feed, drinking water for humans and animals, medicinal baths, shearing points, fire prevention, etc |
| | 11. Due to natural disasters causing damage to the arable land, naturally growing herbaceous land which is difficult to immediately restore cultivated through simple reclamation measures. |
| | 12. The mixed growth of herbaceous plants, trees, and shrubs cannot be distinguished, and the land is mainly composed of herbaceous plants. (Among them, the tree canopy density is less than 0.1 and the shrub coverage is less than 40%). |
| | 13. Land for planting commercial turf. |

**(7) The newly developed high-resolution land cover data will be very useful in land planning and management. It can also be used as the reference data for land cover classification at a relatively coarse resolution. Thus, it is important to know the acquisition time. Figure 24 shows the image capture time and the number of image tiles in different years, but I did not find the acquisition time file on the zenodo.**

**Response:**

Thanks for the comment. We agree that image capture time is very important to land planning and management when users utilize the proposed land-cover product. We have supplemented the acquisition time file (organized as a tiff file) on the Zenodo website (https://zenodo.org/record/8214467) and added other materials to improve the data availability and integrity. Besides, according to the data update, we also released the new version of "User Guide v2.4" on the Zenodo website (https://zenodo.org/record/8214871).